# FedMuon: Accelerating Federated Learning with Matrix Orthogonalization

## Abstract

The core bottleneck of Federated Learning (FL) lies in the communication rounds. That is, how to achieve more effective local updates is crucial for reducing communication rounds. Existing FL methods still primarily use element-wise local optimizers (Adam/SGD), neglecting the geometric structure of the weight matrices. This often leads to the amplification of pathological directions in the weights during local updates, leading deterioration in the condition number and slow convergence. Therefore, we introduce the Muon optimizer in local (named `Local Muon`), which has matrix orthogonalization to optimize matrix-structured parameters. Experimental results show that, in IID setting, `Local Muon` significantly accelerates the convergence of FL and reduces communication rounds compared to Local SGD and Local AdamW. However, in non-IID setting, independent matrix orthogonalization based on the local distributions of each client induces strong client drift. Applying Muon in non-IID FL poses significant challenges: (1) client preconditioner leading to client drift; (2) moment reinitialization. To address these challenges, we propose a novel Federated Muon optimizer (`FedMuon`), which incorporates two key techniques: (1) momentum aggregation, where clients use the aggregated momentum for local initialization; (2) local-global alignment, where the local gradients are aligned with the global update direction to significantly reduce client drift. Theoretically, we prove that `FedMuon` achieves a linear speedup convergence rate of $\mathcal{O}(\sqrt{(L\Delta\sigma_l^2)/(SKR)} + (L\Delta)/R)$ without the heterogeneity assumption, where $S$ is the number of participating clients per round, $K$ is the number of local iterations, and $R$ is the total number of communication rounds. Empirically, we validate the effectiveness of `FedMuon` on language and vision models. Compared to several baselines, `FedMuon` significantly reduces communication rounds and improves test accuracy. The code is available in `https://anonymous.4open.science/r/FedMuon-935D`.

## 1 Introduction

With the rapid growth of data and rising concerns over user privacy, traditional centralized training paradigms have become inadequate. Federated Learning (FL) McMahan et al. (2017) offers a scalable and privacy-preserving framework that enables collaborative model training across decentralized clients without sharing raw data (Liu et al., 2024). As data becomes increasingly siloed, FL is a practical solution for large-scale distributed deep learning. However, data heterogeneity and limited communication rounds create significant bottlenecks in FL. Recent studies reveal that the Hessian matrix in neural networks exhibits an approximate block-diagonal structure with several dense sub-blocks (Collobert, 2004; Zhang et al., 2024), as shown in Figure 1. Understanding parameter matrix structures is crucial for effective federated aggregation, yet this perspective has been largely overlooked in the federated learning literature. Currently, when clients use element-wise optimizers (such as AdamW/SGD) for multi-step updates on their local data, the weight matrices may gradually become ill-conditioned (see Figure 5), causing the update directions to either cancel out or amplify after aggregation. As a result, in each communication round clients struggle to obtain effective updates, and the global model converges slowly.

Recent advancements in the Muon optimizer offer a novel solution to this challenge. The Muon optimizer (Jordan et al.) has recently demonstrated that orthogonal normalization of weight update matrices can significantly accelerate neural network training (see Figure 2). By conditioning the

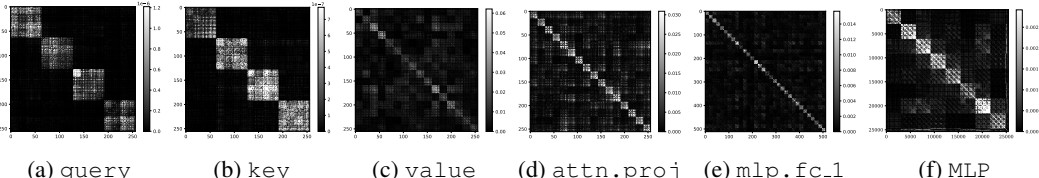

(a) `query`    (b) `key`    (c) `value`    (d) `attn.proj`    (e) `mlp.fc_1`    (f) `MLP`

Figure 1: (a–f):Block-wise Hessian structure of Transformer parameters and MLP (Zhang et al., 2024).

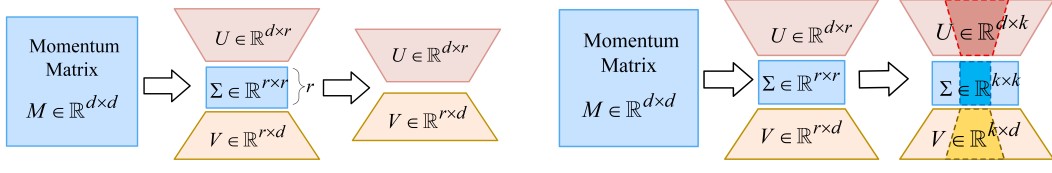

(a) Matrix orthogonalizatio with SVD      (b) Matrix compression using SVD

Figure 2: (a) shows SVD-based matrix orthogonalization; (b) applies SVD to the momentum matrix $M \in \mathbb{R}^{d \times d}$, i.e., $M \approx U \Sigma V^{\top}$, and keeps the top-$k$ singular vectors to obtain $U \in \mathbb{R}^{d \times k}$ and $V \in \mathbb{R}^{k \times d}$.

weight updates to produce consistent changes in the hidden states, orthogonal normalization updates lead to faster convergence, improved training stability, and better hyperparameter transferability across different model scales (Bernstein & Newhouse, 2024; Large et al., 2024; Pethick et al., 2025). Moonshot AI (Liu et al., 2025) found that, when training a 16B model, Muon achieved nearly twice the computational efficiency compared to AdamW (Loshchilov et al., 2017). Similarly, Essential AI (Shah et al., 2025) observed significant improvements with Muon in large-batch training. Both GLM 4.5 and K2 are trained with the Muon optimizer (Liu et al., 2025). These features suggest that using Muon for local training in FL (`Local Muon`) could accelerate local training and reduce communication rounds.

We have also validated the effectiveness of `Local Muon` in FL in IID setting. `Local Muon` significantly outperforms Local SGD and Local AdamW (see Figure 4). `Local Muon` accelerates local convergence and reduces the number of communication rounds required to reach the same level of precision, with faster local loss decrease, smoother training curves, and faster global model convergence (see Figure 4). *However, in non-IID setting, although the local losses of each client still decrease rapidly, the global model after aggregation becomes significantly unstable or even fails to converge (see Figure 4).* We identify the reasons why the Muon optimizer fails in the case of non-IID federated learning from two complementary perspectives.

> **(Challenge 1)** *Client preconditioner leading to client drift: In non-IID FL, Muon's client-specific preconditioner scales gradients from local data distribution, causing misalignment in aggregation.*

> **(Challenge 2)** *Moment reinitialization: reinitializing the moment of Muon from scratch in every round hinders the convergence.*

These challenges motivate us to develop a novel **Federated Muon** optimizer, `FedMuon`, the first FL optimizer that explicitly accounts for the structure of update matrices. `FedMuon` addresses the impact of non-IID data through two key mechanisms: (1) **local-global alignment**, where the current local gradients are aligned with the global update to significantly reduce cross-client inconsistency; (2) **momentum aggregation**, where clients initialize using the aggregated momentum.

**Our contributions** are summarized as follows:

- **Introducing Muon into Federated Learning.** We are the first to design a federated optimizer that explicitly considers the structure of parameter matrices, introducing the matrix orthogonalization method (i.e., `Muon`) into federated learning. Extensive experiments demonstrate its superiority. However, in highly non-IID settings, severe client drift arises. We analyze this issue from two perspectives: (1) **client preconditioner leading to client drift**, (2) **moment reinitialization**.

- **We propose `FedMuon`, a principled FL algorithm with Matrix Orthogonalization.** To address above challenges, `FedMuon` introduce the two mechanisms, **local-global alignment** and

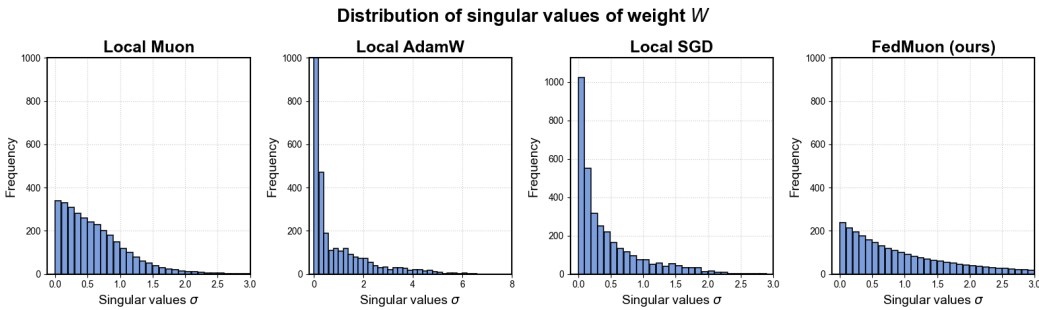

Figure 3: Singular value distributions of Local Muon, Local AdamW, Local SGD, and FedMuon. Local SGD/AdamW are more ill-conditioned with heavier tails and larger singular values, while FedMuon has a more balanced spectrum and a smaller condition number (where the condition number is defined as the ratio between the largest and smallest singular values).

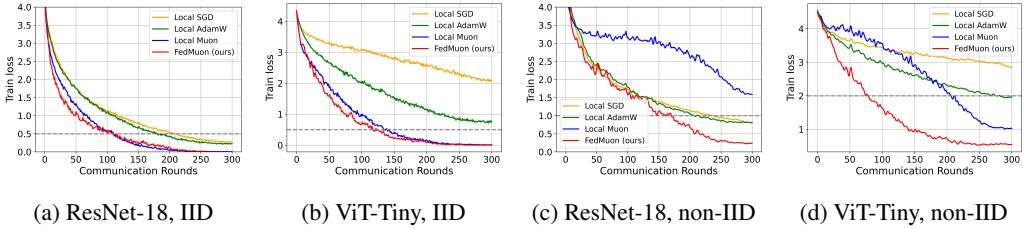

(a) ResNet-18, IID    (b) ViT-Tiny, IID    (c) ResNet-18, non-IID    (d) ViT-Tiny, non-IID

Figure 4: Performance of Local SGD, Local AdamW and Local Muon, we carefully tune the learning rate.

**momentum aggregation**. Inspired by the Hessian structure, we also design a communication-efficient aggregation strategy that communicates the SVD compression of momentum.

- **Theoretical guarantees with improved convergence.** FedMuon achieves a linear convergence rate of $\mathcal{O}(\sqrt{(L\Delta\sigma_l^2)/(SKR)}+(L\Delta)/R)$ without the widely used data heterogeneity assumption. Due to the local-global alignment, its convergence speed is unaffected by data heterogeneity.

## 2 RELATED WORK

• **Optimizers in non-IID Federated Learning.** Data heterogeneity across clients is a fundamental challenge in FL. A range of algorithms have been proposed to mitigate the adverse effects of non-i.i.d. data distributions. For example, FedProx (Li et al., 2020a) introduces a proximal term to restrict local updates; SCAFFOLD (Karimireddy et al., 2020b) applies control variates to correct client drift; and FedCM (Xu et al., 2021) leverages client momentum to stabilize updates. FedOpt (Reddi et al., 2020) incorporates server-side adaptivity using Adam. More recently, Sun et al. (2023) proposed FedLADA to only aggregate the second-moment estimate of Adam to overcome client drift. **Novelty.** Prior correction methods (e.g., SCAFFOLD, FEDCM) assume local SGD and overlook other

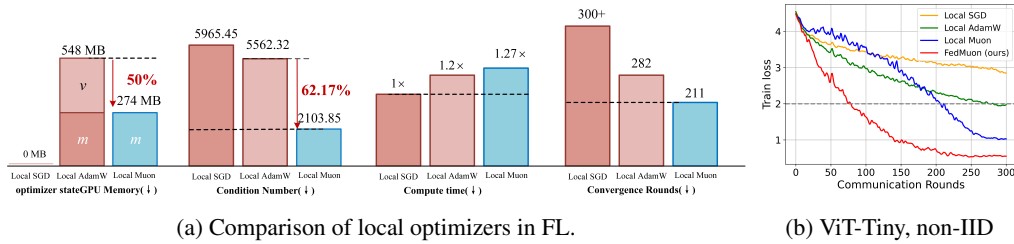

(a) Comparison of local optimizers in FL.      (b) ViT-Tiny, non-IID

Figure 5: (a) Analysis on ViT-Tiny with CIFAR-100, showing optimizer state memory, condition number, computation time, and convergence rounds. Local Muon achieves lower memory cost, lower the condition number, and faster convergence. (b) Training loss curves of ViT-Tiny under non-IID.

optimizers. Directly applying correction methods such as SCAFFOLD or FedCM into Muon optimizer becomes ineffective. We propose **local–global alignment**, injecting a global direction into local updates to curb client drift with advanced optimizers (every local optimizers), while using *half the communication* of SCAFFOLD.

• **Optimizers in Centralized Settings.** Although widely used optimizers such as SGD, Adam (Kingma & Ba, 2014), and AdamW (Loshchilov et al., 2017) are effective in many deep learning settings, they generally treat inherently structured parameters (e.g., matrices) as flattened vectors during optimization. In contrast, recent work has increasingly focused on structure-aware optimizers that make explicit use of the underlying parameter geometry. Examples include Adafactor (Duchi et al., 2011), LAMB (Chen et al., 2023), and Adam-mini (Zhang et al., 2024), which exploit matrix- or layer-level structure to reduce memory footprint. Shampoo (Gupta et al., 2018) further targets matrix and tensor parameters and can be interpreted as an efficient approximation to AdaGrad's full-matrix preconditioner (Duchi et al., 2011). More recently, SOAP (Vyas et al., 2024) integrates the ideas of Adam with Shampoo's matrix-aware design. The Muon optimizer (Jordan et al.) extends this line of work by orthogonalizing weight-update matrices, yielding substantially faster and more stable neural network training.

• **Our contributions.** (1) FedMuon can be viewed as the first federated extension of the Muon optimizer. Unlike standard local Muon, which applies matrix orthogonalization independently on each client, FedMuon augments Muon with a **local–global alignment** to correct the client-drift induced by heterogeneous data and matrix orthogonalization. (2) FedMuon bridges structured optimizers and classical FL methods (Table 21), and we prove that matrix orthogonalization accelerates the convergence of federated learning algorithms. (3) We design a federated framework that is applicable to all matrix-structured optimizers (Muon / Shampoo / LAMB / Soap, etc.), which specifically addresses the problem of preconditioner drift through local–global alignment and momentum aggregation (Table 22).

## 3 FL PROBLEM SETUP

FL aims to optimize model parameters with local clients, i.e., minimizing the following problem:

$$f(\boldsymbol{x}) = \frac{1}{N} \sum_{i=1}^{N} \left( f_i(\boldsymbol{x}) := \mathbb{E}_{\xi_i \sim \mathcal{D}_i} \left[ F_i\left(\boldsymbol{x}; \xi_i\right) \right] \right). \tag{1}$$

The function $f_i$ represents the loss function on client $i$. $\mathbb{E}_{\xi_i \sim \mathcal{D}_i}[\cdot]$ denotes the conditional expectation with respect to the sample $\xi_i$. $\xi_i$ is drawn from distribution $\mathcal{D}_i$ in client $i$. $N$ is the number of clients.

## 4 CHALLENGES OF MUON IN FL

### 4.1 THE MUON OPTIMIZER

**Motivation** Most parameters in neural networks are inherently matrix-valued (e.g., in linear layers or the Q/K/V components of attention mechanisms). However, conventional optimization algorithms such as SGD and AdamW treat these parameters as vectors, effectively flattening them during updates and thereby neglecting their matrix structure. Muon is specifically designed to address this limitation by operating Matrix Orthogonalization directly on update matrix.

**The Muon Optimizer** Muon has recently been proposed as an optimization method for training neural network weights that can be represented as matrices. At iteration $t$, given the current weight $\mathbf{W}_{t-1}$, momentum $\beta$, learning rate $\eta_t$, and the objective $F(\mathbf{W})$, the update rules for the Muon optimizer are:

$$\begin{aligned}
\mathbf{M}_t &= \beta \mathbf{M}_{t-1} + \nabla F(\mathbf{W}_{t-1}); \\
\mathbf{O}_t &= \text{Newton-Schulz-5}(\mathbf{M}_t); \\
\mathbf{W}_t &= \mathbf{W}_{t-1} - \eta_t \mathbf{O}_t.
\end{aligned} \tag{2}$$

Here, $\mathbf{M}_t$ represents the momentum of the gradient at iteration $t$, initialized as a zero matrix when $t = 0$. In Eq.(2), a Newton–Schulz iteration is employed to approximate the solution of $(\mathbf{M}_t \mathbf{M}_t^\top)^{-1/2} \mathbf{M}_t$. Let $\mathbf{U}\boldsymbol{\Sigma}\mathbf{V}^\top = \mathbf{M}_t$ be the singular value decomposition (SVD) of $\mathbf{M}_t$. Then, we have $(\mathbf{M}_t \mathbf{M}_t^\top)^{-1/2} \mathbf{M}_t = \mathbf{U}\mathbf{V}^\top$, which orthogonalizes $\mathbf{M}_t$ (see Figure 2(a)). Intuitively, this orthogonalization ensures that the update matrices remain isomorphic, preventing the weights from

learning solely along a few dominant directions. All matrix orthogonalization operations in this paper are computed using five Newton-Schulz iterations, resulting in about 5% higher computation time compared to AdamW (Jordan et al.). In Table 13, we report our computational time overhead.

### 4.2 CHALLENGES OF MUON IN FL

Despite the widespread use of Muon in centralized deep learning, its adaptation to federated settings remains largely unexplored. In this subsection, we analyze two fundamental challenges that hinder its effectiveness in FL settings.

> **(Challenge 1)** *In non-IID FL, Muon's client-specific preconditioner scales gradients from the client's local data distribution, causing misalignment and cancellation in aggregation.*

**Challenge Analysis:** The matrix orthogonalization in Muon can be viewed as applying a client-specific linear preconditioner $P_i$ to each client's gradient (which can be approximated by Newton-Schulz), transforming the update direction from $g_i$ to $Pg_i$. In the case of non-IID, the gradients $\{g_i\}$ are distributed across their respective dominant subspaces, and the $P_i$ are independently estimated from the local data geometry of each client. This leads to direction mismatch and correlation/amplification: the global update is approximated as $\sum_i \tilde{g}_i = \sum_i P_i g_i$. When the $\{P_i\}$ apply different "rotations/scalings" to the gradient subspaces across clients, the sign and magnitude of $\langle \tilde{g}_i, \tilde{g}_j \rangle$ fluctuate significantly, making it prone to direction cancellation (weakening the norm and making step size ineffective) or phase misalignment (leading to oscillations as it crosses stable regions). These mechanisms together result in the phenomenon of **local–global inconsistency**: the convergence shown on the client side (local loss decreases rapidly) does not translate into global progress (global loss/accuracy stagnates or degrades).

> **(Challenge 2)** *Moment reinitialization: reinitializing the moment of Muon from scratch in every round hinders the convergence rate.*

**Challenge Analysis:** In FL, the Muon optimizer state is reinitialized to zero at the beginning of each round, i.e., $M_i^{r,0} \leftarrow 0$. This reset erases temporal memory across rounds, preventing the accumulation of momentum and thereby slowing convergence. Moreover, accumulating momentum from zero exacerbates client drift.

---

**Algorithm 1** `FedMuon` Algorithm

---

1: **Initial** model $\boldsymbol{x}^0$, $\beta = 0.98$, the number of all clients $N$, each round selected clients $S$.
2: **for** $r = 1, \ldots, R$ **do**
3:    **for** each selected client $i \in \{1, \ldots, S\}$ in parallel **do**
4:       $\boldsymbol{x}_i^{r,0} \leftarrow \boldsymbol{x}^r$, $\boxed{\boldsymbol{M}_i^{r,0} \leftarrow \bar{\boldsymbol{M}}^r}$;
5:       **for** $k = 1, \ldots, K$ **do**
6:          $\boldsymbol{G}_i^{r,k} \leftarrow \nabla f_i(\boldsymbol{x}_i^{r,k}; \xi_i)$; $\boldsymbol{M}_i^{r,k} = \beta \boldsymbol{M}_i^{r,k-1} + \boldsymbol{G}_i^{r,k}$;
7:          $\boldsymbol{U}_i^{r,k}\boldsymbol{V}_i^{r,k\top} = $ Newton-Schulz-5$(\boldsymbol{M}_i^{r,k})$; $\boxed{\boldsymbol{x}_i^{r,k+1} = \boldsymbol{x}_i^{r,k} - \eta[(1-\alpha)\boldsymbol{U}_i^{r,k}\boldsymbol{V}_i^{r,k\top} + \alpha\boldsymbol{\Delta}_G^r]}$;
8:       **end for**
9:       Communicate $(\boldsymbol{x}_i^{r,K} - \boldsymbol{x}_i^{r,0}, \boldsymbol{M}_i^{r,K} \approx U\Sigma V^\top)$ to Server;
10:    **end for**
11:    $\boldsymbol{\Delta}_G^{r+1} = -\frac{1}{SK\eta}\sum_{i=1}^S(\boldsymbol{x}_i^{r,K} - \boldsymbol{x}_i^{r,0})$; $\boldsymbol{x}^{r+1} = \boldsymbol{x}^r + \frac{1}{S}\sum_{i=1}^S(\boldsymbol{x}_i^{r,K} - \boldsymbol{x}_i^{r,0})$;
12:    $\boxed{\bar{\boldsymbol{M}}^{r+1} = \frac{1}{S}\sum_{i=1}^S \boldsymbol{M}_i^{r,K}}$; Communicate $(\boldsymbol{x}^{r+1}, \bar{\boldsymbol{M}}^{r+1}, \boldsymbol{\Delta}_G^{r+1})$ to Clients.
13: **end for**

---

## 5 OUR ALGORITHM: FEDMUON

To robustly leverage matrix orthogonalization in FL, we propose `FedMuon`, with two core mechanisms for the non-IID regime.

### 5.1 MECHANISM I: LOCAL–GLOBAL ALIGNMENT

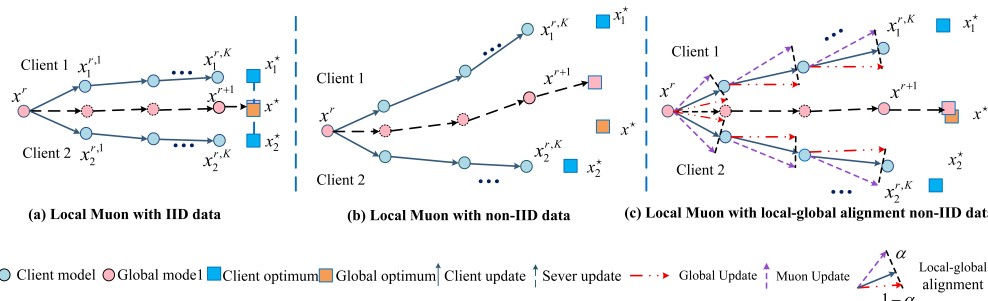

Figure 6: An illustration of `FedMuon`, which corrects client drift through local-global alignment.

---

**(Q1)** *How to overcome local–global inconsistency in Local Muon?*

To address **Challenge 1**, we incorporate local-global alignment into the local update rule:

$$\boldsymbol{x}_i^{r,k+1} = \boldsymbol{x}_i^{r,k} - \eta\big[(1-\alpha)\boldsymbol{U}_i^{r,k}\boldsymbol{V}_i^{r,k\top} + \alpha\boldsymbol{\Delta}_G^r\big], \tag{3}$$

where $\boldsymbol{\Delta}_G^r = -\frac{1}{SK\eta}\sum_{i=1}^S(\boldsymbol{x}_i^{r,K} - \boldsymbol{x}_i^{r,0})$ is the estimated global update. $\alpha$ is the trade-off coefficient between local and global updates. As shown in **Figure 6**, this alignment reduces the divergence of local models and improves global consistency. We also validate its effectiveness in the following experiments (see **Table 5** below). All matrix orthogonalization operations and SVD operations in this paper are computed using five Newton-Schulz iterations, resulting in about 5% higher computation time compared to AdamW (Jordan et al.). In Table 13, we report our computational time overhead.

### 5.2 MECHANISM II: MOMENTUM AGGREGATION

**(Q2)** *How to initialize momentum of Muon in local?*

To achieve better initialization of the momentum $\boldsymbol{M}$ in local, we aggregate local momentum $\boldsymbol{M}_i^{r,K}$ and transmit the aggregated result $\bar{\boldsymbol{M}}$ back to the clients. This strategy partially mitigates the client drift caused by reinitializing momentum from zero, and better aligns local updates with the global update direction (see **Table 5** below).

**(Q3)** *How to efficiently communicate momentum matrices?*

**Momentum Compression via SVD.** Directly communicating the full momentum matrix $M$ would introduce prohibitive communication overhead. To reduce the cost, we compress $M$ using singular value decomposition (SVD): $M = U\Sigma V^\top$, where $U$ and $V$ are orthogonal matrices and $\Sigma$ is the diagonal matrix of singular values. Instead of transmitting the full decomposition, we retain only the top-$k$ singular values (with $k$ set to 5% of the matrix rank), yielding a low-rank approximation (see Figure 2): $M \approx U_k\Sigma_k V_k^\top$. This significantly reduces the communication cost 95%. We refer to this variant as `FedMuon_SVD`. In the following experiments, we show that this approach achieves performance comparable to `FedMuon` (see Table 7). The communication cost of each algorithm is reported in Table 13. The communication overhead of FedMuon increases by only 5%. Here we consider only the upload-side communication cost, because client download bandwidth is typically more than 100× faster than upload and can therefore be ignored in practice.

## 6 THEORETICAL ANALYSIS

In this part, we give the convergence theoretical analysis of our proposed `FedMuon` algorithm. Firstly we state some standard assumptions for the non-convex function $f$.

**Assumption 1** (Smoothness). *The non-convex $f_i$ is a $L$-smooth function for all $i \in [m]$, i.e., $\|\nabla f_i(\boldsymbol{x}) - \nabla f_i(\boldsymbol{y})\| \le L\|\boldsymbol{x} - \boldsymbol{y}\|$, for all $\boldsymbol{x}, \boldsymbol{y} \in \mathbb{R}^d$.*

**Assumption 2** (Bounded Stochastic Gradient). *$\boldsymbol{g}_i^r = \nabla f_i(\boldsymbol{x}_i^r, \xi_i^r)$ computed by using a sampled mini-batch data $\xi_i^r$ in the local client $i$ is an unbiased estimator of $\nabla f_i$ with bounded variance, i.e., $\mathbb{E}_{\xi_i^r}[\boldsymbol{g}_i^r] = \nabla f_i(\boldsymbol{x}_i^r)$ and $\mathbb{E}_{\xi_i^r}\|g_i^r - \nabla f_i(\boldsymbol{x}_i^r)\|^2 \le \sigma_l^2$, for all $\boldsymbol{x}_i^r \in \mathbb{R}^d$.*

These assumptions are standard in FL optimization literature (Fan et al., 2024; Sun et al., 2023).

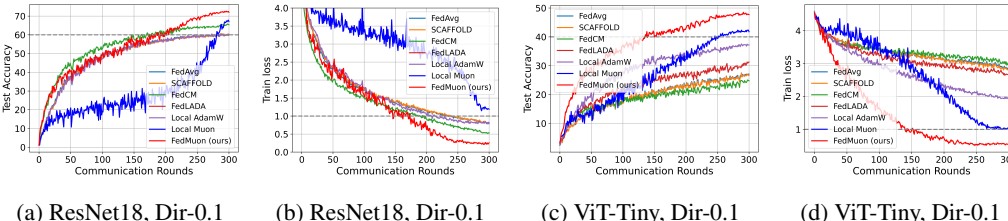

(a) ResNet18, Dir-0.1  (b) ResNet18, Dir-0.1  (c) ViT-Tiny, Dir-0.1  (d) ViT-Tiny, Dir-0.1

Figure 7: Training loss and Test acc curves on CIFAR-100 using ResNet-18 and ViT-Tiny in Dir-0.1.

Table 1: Test accuracy, training loss of each method on CIFAR-100 using **ResNet-18** and **ViT-Tiny** over 300 communication rounds under Dir-0.6 and Dir-0.1 (100 clients, 10% participation, batch size 50, $K = 50$).

| Method | ResNet-18 (Dir-0.6) | | ResNet-18 (Dir-0.1) | | ViT-Tiny (Dir-0.6) | | ViT-Tiny (Dir-0.1) | |
|---|---|---|---|---|---|---|---|---|
| | Test Acc | Loss | Test Acc | Loss | Test Acc | Loss | Test Acc | Loss |
| FedAvg | $64.08_{\pm 0.18}$ | 0.376 | $60.25_{\pm 0.20}$ | 0.767 | $32.36_{\pm 0.08}$ | 2.350 | $27.14_{\pm 0.12}$ | 2.867 |
| FedProx | $63.12_{\pm 0.15}$ | 0.458 | $59.66_{\pm 0.28}$ | 0.812 | $31.51_{\pm 0.12}$ | 2.425 | $26.84_{\pm 0.15}$ | 2.875 |
| FedDyn | $66.12_{\pm 0.28}$ | 0.352 | $63.01_{\pm 0.28}$ | 0.615 | $33.25_{\pm 0.22}$ | 2.125 | $27.66_{\pm 0.18}$ | 2.723 |
| Mime | $67.34_{\pm 0.21}$ | 0.312 | $63.37_{\pm 0.18}$ | 0.604 | $34.12_{\pm 0.14}$ | 2.101 | $27.76_{\pm 0.22}$ | 2.702 |
| FedAdam | $67.23_{\pm 0.18}$ | 0.332 | $63.61_{\pm 0.21}$ | 0.512 | $34.32_{\pm 0.32}$ | 1.965 | $28.50_{\pm 0.11}$ | 2.425 |
| SCAFFOLD | $65.01_{\pm 0.19}$ | 0.365 | $59.37_{\pm 0.16}$ | 0.814 | $32.17_{\pm 0.12}$ | 2.295 | $27.31_{\pm 0.11}$ | 2.855 |
| FedCM | $70.42_{\pm 0.11}$ | 0.282 | $66.73_{\pm 0.14}$ | 0.639 | $26.33_{\pm 0.12}$ | 2.681 | $23.18_{\pm 0.12}$ | 3.038 |
| FedLADA | $65.07_{\pm 0.17}$ | 0.671 | $57.78_{\pm 0.18}$ | 0.498 | $38.33_{\pm 0.12}$ | 2.121 | $31.50_{\pm 0.12}$ | 2.678 |
| Local AdamW | $62.84_{\pm 0.08}$ | 0.363 | $58.97_{\pm 0.10}$ | 0.794 | $40.47_{\pm 0.09}$ | 1.026 | $37.86_{\pm 0.11}$ | 1.954 |
| Local Muon | $71.66_{\pm 0.15}$ | 0.395 | $66.71_{\pm 0.15}$ | 1.504 | $46.69_{\pm 0.12}$ | 0.201 | $40.53_{\pm 0.12}$ | 1.432 |
| FedMuon | $\mathbf{74.12_{\pm 0.15}}$ | 0.001 | $\mathbf{73.05_{\pm 0.15}}$ | 0.246 | $\mathbf{50.22_{\pm 0.12}}$ | 0.162 | $\mathbf{48.18_{\pm 0.12}}$ | 0.556 |

**Theorem 1** (Convergence for non-convex functions). *Under Assumptions 1, 2, if we take $g^0 = 0, \beta_1 = 0, \lambda = 0$ then* FedMuon *converges as follows*

$$\frac{1}{R} \sum_{r=0}^{R-1} \mathbb{E}\left[\|\nabla f(\boldsymbol{x}^r)\|^2\right] \lesssim \mathcal{O}\left(\frac{L\Delta}{R} + \sqrt{\frac{L\Delta \; \sigma_l^2}{R \; SK}}\right). \tag{4}$$

*Here $G_0 := \frac{1}{N} \sum_{i=1}^{N} \left\|\nabla f_i\left(\boldsymbol{x}^0\right)\right\|^2, \Delta = f\left(\boldsymbol{x}^0\right) - f^\star$, $S$ is the number of participating clients per round, $K$ is the number of local iterations, and $R$ is the total number of communication rounds, $\sigma$ is lower bound on singular values, $d$ is the total dimensionality of the parameter.*

The detailed proof is provided in the **Appendix**. The convergence rate of FedMuon is faster than than that of Local Muon and Local SGD, $\mathcal{O}\left(\frac{L\Delta}{R} + \sqrt{\frac{L\Delta \sigma_l^2 + \sigma_g^2}{R \; SK}}\right)$. Notably, our result does not rely on data heterogeneity **Assumption**. This improvement stems from the suppression of local drift achieved by the proposed local–global alignment mechanism. The effectiveness of this design is further validated in the **ablation study** (Table 5). The data heterogeneity **Assumption** is standard in federated learning. With our global–local alignment, we mitigate data heterogeneity and no longer rely on this assumption, achieving faster convergence than existing methods, as confirmed by both theory and experiments.

# 7 EXPERIMENTS

**Datasets.** We evaluate FedMuon on both vision and language tasks. (*i*) For image classification, we use CIFAR-100 (Krizhevsky et al., 2009), and Tiny ImageNet (Le & Yang, 2015). (*ii*) For NLP tasks, we adopt benchmark datasets from the GLUE benchmark, including SST-2 (Socher et al., 2013), QQP (Socher et al., 2013), and OpenWebText dataset. To simulate data heterogeneity across clients, we follow the Dirichlet partitioning scheme (Hsu et al., 2019), where a Dir-0.6 corresponds to a low heterogeneity and Dir-0.1 implies high heterogeneity.

**Model Architectures.** We explore a variety of model types: (*i*) ResNet-18 (He et al., 2016) as a representative convolutional neural network (CNN), (*ii*) Swin Transformer (Liu et al., 2021) and ViT-Tiny (Dosovitskiy et al., 2020) for Vision Transformers, and (*iii*) RoBERTa-Base (Liu et al.,

2019) and GPT-2 Radford et al. (2019) for large-scale language model.

**Baselines.** We compare our method against state-of-the-art FL algorithms: FedAvg (Local SGD) (McMahan et al., 2017), SCAFFOLD (Karimireddy et al., 2020b), FedCM (Xu et al., 2021), FedLADA (Sun et al., 2023), Local AdamW and Local Muon, FedProx Li et al. (2020b), Fed-Dyn Acar et al. (2021), Mime Karimireddy et al. (2020a), and FedAdam Reddi et al. (2020). In the Appendix (Table 15), we compare additional FL algorithms designed to address data heterogeneity.

**Hyperparameter Settings.** For FedAvg, SCAFFOLD, FedCM, FedProx, FedDyn, Mime, and FedAdam, the $lr$ is selected from $\{10^{-2}, 3 \times 10^{-2}, 5 \times 10^{-2}, 10^{-1}, 3 \times 10^{-1}\}$, with a weight decay of 0.001. For FedLADA, Local AdamW, the $lr$ is selected from $\{10^{-4}, 3 \times 10^{-4}, 5 \times 10^{-4}, 8 \times 10^{-4}, 10^{-3}\}$, with weight decay 0.01 or 0.001, $\beta_1 = 0.9$, $\beta_2 = 0.999$. We apply cosine learning rate decay, and set FedMuon to $\alpha = 0.5$, weight decay 0.01. We set the learning rate of FedMuon and Local Muon to be $3 \times 10^{-2}, 2 \times 10^{-2}, 3 \times 10^{-3}$. Additional hyperparameter configurations are detailed in the **Appendix** (Table 10, Table 12). We release all code, configuration files to ensure full reproducibility. All results are averaged over 5 runs with std reported with seeds 42, 43, 44, 45, 46.

## 7.1 RESULTS ON CONVOLUTIONAL NEURAL NETWORKS AND TRANSFORMER

**Training on CIFAR-100 with ResNet-18.** Table 1 and **Figure 7** present the test accuracy and training loss on CIFAR-100 using ResNet-18. FedMuon achieves the best performance under both Dir-0.6 and Dir-0.1 settings, reaching a top accuracy of **74.12%** and **73.05%**, respectively. It also attains the lowest training loss (**0.001** and **0.246**), demonstrating faster and more stable convergence. Compared to other adaptive baselines such as Local AdamW, FedMuon shows superior generalization under data heterogeneity, confirming its effectiveness in CNNs. In our experiments, Muon provides immediate speedups under IID data, but under non-IID data Local Muon initially converges slowly due to mismatched client preconditioners and exacerbated client drift (Challenge 1). FedMuon mitigates this issue, achieving fast and stable convergence, and also yields clear speedups in IID settings (Table 16).

**Training on CIFAR-100 with ViT-Tiny.** Table 1 and **Figure 7** show FedMuon achieves the best performance across both heterogeneity levels, with **50.22%** (Dir-0.6) and **48.18%** (Dir-0.1), and the lowest training loss (**0.162** and **0.556**), confirming its efficient convergence. These results validate that FedMuon is particularly effective for federated vision Transformers under non-i.i.d. conditions. The small dataset CIFAR100 is difficult to support the performance of ViT, resulting in lower accuracy. Therefore, we continued to test on the pretrained model.

**Fine-tuning Results on Swin Transformer.** Table 2 reports results on Swin Transformer under Dir-0.1 with LoRA. FedMuon achieves the highest test accuracy on both CIFAR-100 (**84.88%**) and Tiny ImageNet (**84.95%**), while also attaining the lowest training loss, reflecting faster convergence. FedMuon consistently outperforms baselines (including Local AdamW and Local Muon), demonstrating its effectiveness in fine-tuning Vision Transformer models under non-IID data.

**Fine-tuning Results on LLMs.** Table 18 summarizes results on the GLUE benchmark using RoBERTa-Base with LoRA, 20 clients, 20% participation, batch size 16, $K = 50$, rank=16. FedMuon achieves the highest accuracy of GLUE outperforming strong baselines such as FedAvg and Local Muon. It is particularly strong on challenging tasks like **RTE** and **QQP**, exceeding the next best methods by **+1.65%** and **+1.59%**, respectively. In the appendix, we additionally report results under the setting with 4 clients, 100% client participation, and Dir-0.8 data partitioning (see Table 17).

Table 2: Comparison of test accuracy and training loss for **Swin Transformer** under Dir-0.1 with 100 rounds (100 clients, 5% participation, batch size 16, $K = 50$).

| Method | CIFAR-100 | | Tiny ImageNet | |
|---|---|---|---|---|
| | Test Acc | Loss | Test Acc | Loss |
| FedAvg | $80.02_{\pm 0.28}$ | 0.588 | $80.38_{\pm 0.22}$ | 0.826 |
| FedProx | $81.21_{\pm 0.13}$ | 0.521 | $81.86_{\pm 0.12}$ | 0.885 |
| FedDyn | $81.67_{\pm 0.15}$ | 0.501 | $82.48_{\pm 0.18}$ | 0.641 |
| Mime | $82.21_{\pm 0.11}$ | 0.562 | $82.56_{\pm 0.14}$ | 0.655 |
| FedAdam | $82.56_{\pm 0.15}$ | 0.545 | $82.21_{\pm 0.11}$ | 0.685 |
| SCAFFOLD | $81.30_{\pm 0.18}$ | 0.514 | $82.41_{\pm 0.18}$ | 0.650 |
| FedCM | $82.38_{\pm 0.11}$ | 0.565 | $83.18_{\pm 0.14}$ | 0.522 |
| FedLADA | $74.64_{\pm 0.15}$ | 0.598 | $70.94_{\pm 0.19}$ | 0.944 |
| Local AdamW | $83.35_{\pm 0.16}$ | 0.381 | $80.26_{\pm 0.12}$ | 0.686 |
| Local Muon | $79.73_{\pm 0.18}$ | 0.396 | $80.24_{\pm 0.10}$ | 0.734 |
| FedMuon | $\mathbf{84.88_{\pm 0.17}}$ | **0.123** | $\mathbf{84.95_{\pm 0.12}}$ | **0.394** |

Table 3: Test accuracy (%) using RoBERTa-Base with LoRA across GLUE tasks over 100 communication rounds under Dirichlet-0.5 partition. (20 clients, 20% participation, batch size 16, $K = 50$)

| Method (Dir-0.5) | CoLA | RTE | SST-2 | QQP | MRPC | QNLI | MNLI |
|---|---|---|---|---|---|---|---|
| FedAvg | $51.00_{\pm0.26}$ | $51.99_{\pm0.24}$ | $93.04_{\pm0.16}$ | $81.75_{\pm0.11}$ | $88.24_{\pm0.18}$ | $89.36_{\pm0.15}$ | $81.72_{\pm0.25}$ |
| FedProx | $53.11_{\pm0.14}$ | $53.25_{\pm0.21}$ | $92.26_{\pm0.18}$ | $81.15_{\pm0.11}$ | $87.36_{\pm0.12}$ | $88.12_{\pm0.14}$ | $81.41_{\pm0.21}$ |
| FedDyn | $53.21_{\pm0.28}$ | $52.22_{\pm0.30}$ | $92.36_{\pm0.21}$ | $81.35_{\pm0.21}$ | $87.89_{\pm0.11}$ | $89.12_{\pm0.21}$ | $82.18_{\pm0.21}$ |
| Mime | $52.15_{\pm0.17}$ | $51.62_{\pm0.21}$ | $92.21_{\pm0.28}$ | $80.26_{\pm0.18}$ | $88.04_{\pm0.12}$ | $89.11_{\pm0.21}$ | $82.51_{\pm0.20}$ |
| FedAdam | $53.21_{\pm0.28}$ | $52.52_{\pm0.31}$ | $92.36_{\pm0.25}$ | $82.22_{\pm0.28}$ | $88.12_{\pm0.34}$ | $88.01_{\pm0.23}$ | $82.66_{\pm0.22}$ |
| SCAFFOLD | $52.15_{\pm0.17}$ | $50.65_{\pm0.20}$ | $93.28_{\pm0.16}$ | $80.26_{\pm0.18}$ | $88.35_{\pm0.16}$ | $89.32_{\pm0.24}$ | $82.11_{\pm0.20}$ |
| FedCM | $53.21_{\pm0.28}$ | $52.22_{\pm0.30}$ | $92.56_{\pm0.25}$ | $81.22_{\pm0.28}$ | $88.56_{\pm0.13}$ | $89.02_{\pm0.23}$ | $82.12_{\pm0.27}$ |
| FedLADA | $54.66_{\pm0.17}$ | $57.02_{\pm0.08}$ | $93.88_{\pm0.16}$ | $81.56_{\pm0.20}$ | $89.01_{\pm0.28}$ | $89.86_{\pm0.29}$ | $82.44_{\pm0.17}$ |
| Local AdamW | $55.38_{\pm0.12}$ | $59.57_{\pm0.25}$ | $93.81_{\pm0.19}$ | $81.51_{\pm0.05}$ | $88.73_{\pm0.23}$ | $89.55_{\pm0.15}$ | $82.86_{\pm0.26}$ |
| Local Muon | $55.54_{\pm0.05}$ | $64.93_{\pm0.17}$ | $93.58_{\pm0.27}$ | $83.06_{\pm0.11}$ | $88.95_{\pm0.13}$ | $90.52_{\pm0.27}$ | $84.63_{\pm0.10}$ |
| FedMuon(ours) | $\mathbf{56.78_{\pm0.11}}$ | $\mathbf{66.58_{\pm0.29}}$ | $\mathbf{93.54_{\pm0.25}}$ | $\mathbf{84.65_{\pm0.16}}$ | $\mathbf{88.21_{\pm0.07}}$ | $\mathbf{90.24_{\pm0.13}}$ | $\mathbf{85.21_{\pm0.18}}$ |

Table 4: Test accuracy of each method on CIFAR-100 using **ViT-Tiny**, **ViT-Small**, **ViT-Base** and **ViT-Large** over 300 communication rounds under Dir-0.1 (100 clients, 10% participation, batch size 50, $K = 50$), and train loss of each method on OpenWebText data using **GPT-2 Small**, **GPT-2 Medium**, **GPT-2 Large** and **GPT-2 XL** over 300 communication rounds (20 clients, 20% participation, batch size 16, $K = 100$).

| Method | CIFAR-100 (Test Acc, %) | | | | OpenWebText (Train Loss) | | | |
|---|---|---|---|---|---|---|---|---|
| | ViT-Tiny | ViT-Small | ViT-Base | ViT-Large | GPT-2 S | GPT-2 M | GPT-2 L | GPT-2 XL |
| FedAvg | 27.14 | 29.52 | 31.15 | 33.56 | 4.25 | 4.12 | 4.01 | 3.91 |
| FedProx | 26.84 | 28.63 | 31.05 | 33.25 | 4.33 | 4.21 | 4.15 | 4.05 |
| FedDyn | 27.31 | 30.24 | 32.85 | 34.58 | 4.12 | 4.01 | 3.95 | 3.82 |
| Mime | 27.66 | 31.23 | 33.11 | 35.34 | 4.10 | 4.02 | 3.89 | 3.78 |
| FedAdam | 28.50 | 33.15 | 33.15 | 33.15 | 4.02 | 3.95 | 3.82 | 3.75 |
| SCAFFOLD | 27.31 | 30.24 | 32.85 | 34.58 | 4.12 | 4.01 | 3.95 | 3.82 |
| FedCM | 23.18 | 25.15 | 27.88 | 29.01 | 4.32 | 4.21 | 4.02 | 3.91 |
| FedLADA | 31.50 | 33.15 | 33.15 | 33.15 | 3.56 | 3.45 | 3.33 | 3.24 |
| Local AdamW | 37.86 | 37.86 | 37.86 | 37.86 | 3.44 | 3.35 | 3.27 | 3.15 |
| Local Muon | 40.53 | 42.34 | 45.26 | 46.54 | 3.33 | 3.21 | 3.09 | 2.98 |
| FedMuon(ours) | **48.18** | **50.52** | **53.63** | **56.24** | **3.12** | **2.98** | **2.85** | **2.74** |

Table 4 compares FedMuon with a range of federated optimizers on both vision and language benchmarks. On CIFAR-100, FedMuon consistently achieves the highest test accuracy across all ViT scales, improving from 27.14% to 48.18% on ViT-Tiny and from 33.56% to 56.24% on ViT-Large compared to FedAvg, and further outperforming Local AdamW and Local Muon by a large margin. On C4 language modeling with GPT-2, FedMuon attains the lowest training loss for all model sizes, reducing the loss from 4.25 to 3.12 on GPT-2 Small and from 3.91 to 2.74 on GPT-2 XL. These results indicate that FedMuon scales effectively to larger Transformer models and consistently improves optimization efficiency over strong baselines in both vision and language tasks.

## 7.2 ABLATION STUDY

**Impact of $\Delta_G$ and $\bar{m}$.** As shown in Table 5 left, we conduct an ablation study of FedMuon. FedMuon incorporates momentum averaging $\bar{m}$ and global update differences $\Delta_G$. The results clearly indicate that Local Muon consistently outperforms both SGD and AdamW, demonstrating its superior ability to handle non-IID FL. Moreover, our strategy consistently improves the performance of other optimizers as well.

**Impact of $\Delta_G$ and $\bar{m}$ on other optimizers.** As shown in Table 5 right, we compare different local optimizers with $\Delta_G$ and $\bar{m}$. The results demonstrate that Local Muon consistently achieves the best performance, significantly outperforming SGD and AdamW, thereby highlighting its effectiveness in mitigating data heterogeneity. Further results on additional optimizers in Table 22.

**Accelerationof Matrix Orthogonalization on Federated Learning.** See in Table 21.Matrix orthogonalization also provides acceleration benefits for other federated learning algorithms.

Table 5: Ablation study of `FedMuon` on CIFAR-100 (Dir-0.1, 300 rounds). Left: effect of removing components. Right: effect of different local optimizers.

| Variant | ResNet-18 | ViT-Tiny | Variant | ResNet-18 | ViT-Tiny |
|---------|-----------|----------|---------|-----------|----------|
| A1: w/o $\bar{m}$ | $69.12_{\pm 0.18}$ | $43.67_{\pm 0.19}$ | Local SGD + $\bar{m}$+$\Delta_G$ | $66.28_{\pm 0.17}$ | $32.56_{\pm 0.11}$ |
| A2: w/o $\Delta_G$ | $68.05_{\pm 0.10}$ | $44.56_{\pm 0.16}$ | Local AdamW + $\bar{m}$+$\Delta_G$ | $64.25_{\pm 0.12}$ | $41.26_{\pm 0.17}$ |
| A3: `FedMuon` | $\mathbf{73.05_{\pm 0.15}}$ | $\mathbf{48.18_{\pm 0.12}}$ | Local Muon + $\bar{m}$+$\Delta_G$ | $\mathbf{73.05_{\pm 0.15}}$ | $\mathbf{48.18_{\pm 0.12}}$ |

Table 6: Impact of $\alpha$ and $\beta$ on `FedMuon` using ViT-Tiny and ResNet-18 on CIFAR-100 (Dir-0.1).

| Model | $\alpha$ | | | | | $\beta$ | | | | |
|-------|------|------|------|------|------|------|------|------|------|------|
| | 0.00 | 0.25 | **0.50** | 0.75 | 0.90 | 0.80 | 0.90 | 0.95 | **0.98** | 0.99 |
| **ResNet-18** | 68.05 | 69.89 | **73.01** | 72.12 | 67.56 | 68.22 | 70.56 | 71.23 | **73.01** | 72.66 |
| **ViT-Tiny** | 44.56 | 46.28 | **48.18** | 47.59 | 46.23 | 44.86 | 45.23 | 46.59 | **48.18** | 47.56 |

Table 7: Ablation of momentum aggregation strategies in `FedMuon` on CIFAR-100 under Dir-0.1. **CommCost** denotes communication cost per round (MB), and **CompCost** denotes computation time per round (s).

| Aggregation | ResNet-18 | | | ViT-Tiny | | |
|-------------|-----------|----------|----------|----------|----------|----------|
| | Test Acc | CommCost | CompCost | Test Acc | CommCost | CompCost |
| NoAgg | 69.12 | 46.8 MB (1×) | 6.23 s | 43.67 | 22.8 MB (1×) | 5.14 s |
| Agg-$m$ | 73.05 | 93.6 MB (2×) | 6.44 s | 48.18 | 45.6 MB (2×) | 5.21 s |
| Agg-$m$-SVD | 72.56 | 49.2 MB (1.05×) | 6.48 s | 47.66 | 23.9 MB (1.05×) | 5.25 s |

**Impact of $\alpha$.** **Table 6** evaluates the effect of the local-global alignment parameter $\alpha$ in `FedMuon`. As shown by **Theorem 1**, incorporating global update direction helps suppress client drift and accelerates convergence. We observe that $\alpha = 0.5$ yields the best performance, striking a balance between local adaptivity and global consistency, in line with our theoretical insight.

**Impact of $\beta$.** **Table 6** verifies the effectiveness of *local momentum accumulation*. When the momentum parameter $\beta$ is too small, the aggregated global momentum is quickly diluted. Conversely, an overly large $\beta$ slows local gradient accumulation and delays responsiveness to new data. These results suggest that $\beta$ should balance global momentum preservation with timely adaptation to client updates. We observe that $\beta = 0.98$ yields the best performance.

**Impact of Momentum Aggregation Strategy.** **Table 7** shows Momentum Aggregation Strategy, Agg-$m$-SVD (`FedMuon_SVD`), achieves the best balance between accuracy and communication cost. While Agg-$m$ improves performance, it introduces excessive communication (2×). In contrast, Agg-$m$-SVD attains similar benefits with only 1.05× communication cost.

## 8 CONCLUSION

In this work, we proposed `FedMuon`, a structure-aware federated optimizer for training large-scale Transformer and vision models. `FedMuon` addresses core challenges of non-IID. Federated learning—client drift, unstable optimizer states, and inefficient communication—by coupling *matrix-orthogonalized* local updates with *local-globall alignment* and *cross-round momentum aggregation*, complemented by low-rank state sharing. We provided non-convex convergence analysis clarifying how alignment and orthogonalization jointly control the bias introduced by multi-step local training, and we documented strong empirical gains across vision and language tasks, particularly on Transformer architectures. These results highlight that treating optimizer updates as matrices (rather than flat vectors) offers a principled route to reliable and efficient FL. We believe `FedMuon` opens a pathway for adapting modern, structure-aware optimizers to federated settings and inspires future extensions to related methods such as LAMB (Chen et al., 2023) or Lion (Chen et al., 2023). Beyond federated learning, the principles of `FedMuon` can be directly applied to large-scale distributed training and parameter-efficient fine-tuning of foundation models, where communication efficiency and stable optimization are equally critical.

## 9 ETHICS STATEMENT

This work adheres to the ICLR Code of Ethics. Our study involves no human subjects or animal experimentation. All experiments are conducted on public, license-compliant academic benchmarks under non-IID federated partitions; no personally identifiable information is collected or processed. Data usage follows the original dataset terms, and we apply standard safeguards to avoid amplifying social or demographic biases (e.g., consistent splits, shared hyperparameter budgets, and reporting of variance across seeds). The method—FEDMUON, which aggregates cross-round momentum and performs matrix-orthogonalized local updates—does not require access to raw user data beyond standard benchmark usage, and introduces no additional privacy risks beyond those present in conventional federated optimization. We will release code and configurations to support transparent verification.

## 10 REPRODUCIBILITY STATEMENT

We make every effort to ensure reproducibility. The paper specifies training steps, model configurations (e.g., ResNet/ViT for vision and RoBERTa-style encoders for NLP), non-IID partition protocols, client sampling, and hardware details. Unless noted otherwise, each configuration is repeated with five independent seeds $\{42, 43, 44, 45, 46\}$; we report mean $\pm$ standard deviation and provide per-run logs/curves. Implementation details for FEDMUON (orthogonalized updates, global–local alignment, cross-round momentum aggregation, and low-rank SVD compression) are described in algorithmic form with all tunables exposed. An anonymous repository includes source code, configuration files, data-partition scripts, and instructions to exactly reproduce the main tables and figures.

## 11 LLM USAGE

Large Language Models (LLMs) were used solely for language editing (grammar, phrasing, and clarity) of the manuscript text. LLMs were *not* involved in research ideation, methodological design, theoretical analysis, dataset preparation, implementation, or result selection. The authors are fully responsible for the scientific content and verify that any LLM-assisted passages comply with ethical guidelines and do not constitute plagiarism or scientific misconduct.

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

**LIST OF APPENDIX**

---

**Algorithm 2** `FedMuon` Algorithm (analysis variant)

---

1: **Initial** model $\boldsymbol{x}^0$, $\beta_1 = 0.98$, the number of all clients $N$, each round selected clients $S$, weight decay $\lambda$.
2: **for** $r = 1, \ldots, R$ **do**
3:     **for** each selected client $i \in \{1, \ldots, S\}$ in parallel **do**
4:         $\boldsymbol{x}_i^{r,0} \leftarrow \boldsymbol{x}^r$, $\boldsymbol{M}_i^{r,0} \leftarrow \bar{\boldsymbol{M}}^r$;
5:         **for** $k = 1, \ldots, K$ **do**
6:             $\boldsymbol{G}_i^{r,k} \leftarrow \nabla f_i(\boldsymbol{x}_i^{r,k}; \xi_i)$;
7:             $\boldsymbol{U}_i^{r,k}, \boldsymbol{\Sigma}_i^{r,k}, \boldsymbol{V}_i^{r,k} = \text{SVD}(\boldsymbol{G}_i^{r,k})$;
8:             $\boldsymbol{x}_i^{r,k+1} = \boldsymbol{x}_i^{r,k} - \eta\big[(1-\alpha)\boldsymbol{U}_i^{r,k}\boldsymbol{V}_i^{r,k^\top} + \alpha\boldsymbol{\Delta}_G^r\big]$;
9:         **end for**
10:         Communicate $(\boldsymbol{x}_i^{r,K} - \boldsymbol{x}_i^{r,0}, \boldsymbol{M}_i^{r,K})$ to Server;
11:     **end for**
12:     $\boldsymbol{\Delta}_G^r = \frac{-1}{SK\eta}\sum_{i=1}^S (\boldsymbol{x}_i^{r,K} - \boldsymbol{x}_i^{r,0})$;
13:     $\boldsymbol{x}^{r+1} = \boldsymbol{x}^r + \frac{1}{S}\sum_{i=1}^S (\boldsymbol{x}_i^{r,K} - \boldsymbol{x}_i^{r,0})$;
14:     $\bar{\boldsymbol{M}}^{r+1} = \frac{1}{S}\sum_{i=1}^S \boldsymbol{M}_i^{r,K}$;
15:     Communicate $(\boldsymbol{x}^{r+1}, \bar{\boldsymbol{M}}^{r+1}, \boldsymbol{\Delta}_G^{r+1})$ to Clients.
16: **end for**

---

## A    APPENDIX A: PROOF OF THEOREM 1 AND CONVERGENCE ANALYSIS

### A.1    FEDMUON ALGORITHM

To simplify the analysis, we consider the iterative rules as in Algorithm 2, where we let $\beta_1 = 0$. The local update takes the following rule:

$$\boldsymbol{x}_i^{r,k+1} = \boldsymbol{x}_i^{r,k} - \eta_t\big[(1-\alpha)\boldsymbol{U}_i^{r,k}\boldsymbol{V}_i^{r,k^\top} + \alpha\boldsymbol{\Delta}_G^r\big].$$

**Simplified setting for theoretical analysis.** Our primary focus in this paper is to investigate how the *matrix orthogonalization mechanism* accelerates convergence in federated learning. Introducing an additional *local momentum term* (e.g., $\beta_1 > 0$) would bring in temporal dependencies across iterations, making the theoretical convergence analysis substantially more complex without offering additional conceptual insights into the effect of orthogonalization itself.

Therefore, for analytical tractability, we consider a simplified variant where we set $\beta_1 = 0$ in the local update rule:

$$\boldsymbol{x}_i^{r,k+1} = \boldsymbol{x}_i^{r,k} - \eta_t\big[(1-\alpha)\,\boldsymbol{U}_i^{r,k}\boldsymbol{V}_i^{r,k^\top} + \alpha\,\boldsymbol{\Delta}_G^r\big].$$

This simplification isolates the impact of low-rank orthogonalization and global gradient mixing, allowing us to derive clean convergence bounds that clearly reveal how orthogonalization improves communication efficiency and stability.

Importantly, we empirically verify that this simplified version performs **on par** with the full algorithm using local momentum. The empirical results demonstrate that setting $\beta_1 = 0$ does not materially affect convergence speed or final accuracy, thereby justifying the use of this simplified formulation for theoretical analysis.

### A.2    ASSUMPTION

**Assumption A.1** (Smoothness). *The non-convex $f_i$ is one $L$-smooth function for all $i \in [m]$, i.e., $\|\nabla f_i(\boldsymbol{x}) - \nabla f_i(\boldsymbol{y})\| \leq L\|\boldsymbol{x} - \boldsymbol{y}\|$, for all $\boldsymbol{x}, \boldsymbol{y} \in \mathbb{R}^d$.*

**Assumption A.2** (Bounded Stochastic Gradient). *$\boldsymbol{g}_i^r = \nabla f_i(\boldsymbol{x}_i^r, \xi_i^r)$ computed by using a sampled mini-batch $\xi_i^r$ in client $i$ is an unbiased estimator of $\nabla f_i$ with bounded variance: $\mathbb{E}_{\xi_i^r}[\boldsymbol{g}_i^r] = \nabla f_i(\boldsymbol{x}_i^r)$ and $\mathbb{E}_{\xi_i^r}\|g_i^r - \nabla f_i(\boldsymbol{x}_i^r)\|^2 \leq \sigma_l^2$.*

In this section, we give the theoretical analysis of our proposed uon algorithm. Firstly we state some standard assumptions for the non-convex function $F$.

### A.3    MAIN LEMMAS

**Lemma 1.** *Suppose $\{X_1, \cdots, X_\tau\} \subset \mathbb{R}^d$ be random variables that are potentially dependent. If their marginal means and variances satisfy $\mathbb{E}[X_i] = \mu_i$ and $\mathbb{E}[\|X_i - \mu_i\|^2] \leq \sigma^2$, then it holds that*

$$\mathbb{E}\left[\left\|\sum_{i=1}^{\tau} X_i\right\|^2\right] \leq \left\|\sum_{i=1}^{\tau} \mu_i\right\|^2 + \tau^2 \sigma^2.$$

*If they are correlated in the Markov way such that $\mathbb{E}[X_i \mid X_{i-1}, \cdots X_1] = \mu_i$ and $\mathbb{E}\left[\|X_i - \mu_i\|^2 \mid \mu_i\right] \leq \sigma^2$, i.e., the variables $\{X_i - \mu_i\}$ form a martingale. Then the following tighter bound holds:*

$$\mathbb{E}\left[\left\|\sum_{i=1}^{\tau} X_i\right\|^2\right] \leq 2\mathbb{E}\left[\left\|\sum_{i=1}^{\tau} \mu_i\right\|^2\right] + 2\tau\sigma^2$$

.

**Lemma 2.** *Given vectors $v_1, \cdots, v_N \in \mathbb{R}^d$ and $\bar{v} = \frac{1}{N}\sum_{i=1}^{N} v_i$, if we sample $\mathcal{S} \subset \{1, \cdots, N\}$ uniformly randomly such that $|\mathcal{S}| = S$, then it holds that*

$$\mathbb{E}\left[\left\|\frac{1}{S}\sum_{i\in\mathcal{S}} v_i\right\|^2\right] = \|\bar{v}\|^2 + \frac{N-S}{S(N-1)}\frac{1}{N}\sum_{i=1}^{N}\|v_i - \bar{v}\|^2.$$

*Proof.* Letting $\mathbb{I}\{i \in \mathcal{S}\}$ be the indicator for the event $i \in \mathcal{S}_r$, we prove this lemma by direct calculation as follows:

$$\mathbb{E}\left[\left\|\frac{1}{S}\sum_{i\in\mathcal{S}} v_i\right\|^2\right] = \mathbb{E}\left[\left\|\frac{1}{S}\sum_{i=1}^{N} v_i\mathbb{I}\{i \in \mathcal{S}\}\right\|^2\right]$$

$$= \frac{1}{S^2}\mathbb{E}\left[\left(\sum_i \|v_i\|^2\,\mathbb{I}\{i \in \mathcal{S}\} + 2\sum_{i<j} v_i^\top v_j \mathbb{I}\{i,j \in \mathcal{S}\}\right)\right]$$

$$= \frac{1}{SN}\sum_{i=1}^{N}\|v_i\|^2 + \frac{1}{S^2}\frac{S(S-1)}{N(N-1)}2\sum_{i<j} v_i^\top v_j$$

$$= \frac{1}{SN}\sum_{i=1}^{N}\|v_i\|^2 + \frac{1}{S^2}\frac{S(S-1)}{N(N-1)}\left(\left\|\sum_{i=1}^{N} v_i\right\|^2 - \sum_{i=1}^{N}\|v_i\|^2\right)$$

$$= \frac{N-S}{S(N-1)}\frac{1}{N}\sum_{i=1}^{N}\|v_i\|^2 + \frac{N(S-1)}{S(N-1)}\|\bar{v}\|^2$$

$$= \frac{N-S}{S(N-1)}\frac{1}{N}\sum_{i=1}^{N}\|v_i - \bar{v}\|^2 + \|\bar{v}\|^2.$$

$\square$

### A.4    BASIC ASSUMPTIONS AND NOTATIONS

Let $\mathcal{F}^0 = \emptyset$ and $\mathcal{F}_i^{r,k} := \sigma\left(\left\{x_i^{r,j}\right\}_{0 \leq j \leq k} \cup \mathcal{F}^r\right)$ and $\mathcal{F}^{r+1} := \sigma\left(\cup_i \mathcal{F}_i^{r,K}\right)$ for all $r \geq 0$ where $\sigma(\cdot)$ indicates the $\sigma$-algebra. Let $\mathbb{E}_r[\cdot] := \overline{\mathbb{E}}[\cdot \mid \mathcal{F}^r]$ be the expectation, conditioned on the filtration $\mathcal{F}^r$, with respect to the random variables $\left\{\mathcal{S}^r, \left\{\xi_i^{r,k}\right\}_{1 \leq i \leq N, 0 \leq k < K}\right\}$ in the $r$-th iteration. We also use $\mathbb{E}[\cdot]$ to denote the global expectation over all randomness in algorithms. Through out the proofs, we use $\sum_i$ to represent the sum over $i \in \{1, \ldots, N\}$, while $\sum_{i \in \mathcal{S}^r}$ denotes the sum over $i \in \mathcal{S}^r$.

Similarly, we use $\sum_k$ to represent the sum of $k \in \{0, \dots, K-1\}$. For all $r \geq 0$, we define the following auxiliary variables to facilitate proofs:

$$\mathcal{E}_r := \mathbb{E}\left[\left\|\nabla f\left(x^r\right) - g^{r+1}\right\|^2\right]$$

$$U_r := \frac{1}{NK} \sum_i \sum_k \mathbb{E}\left[\left\|x_i^{r,k} - x^r\right\|\right]^2$$

$$\zeta_i^{r,k} := \mathbb{E}\left[x_i^{r,k+1} - x_i^{r,k} \mid \mathcal{F}_i^{r,k}\right]$$

$$\Xi_r := \frac{1}{N} \sum_{i=1}^N \mathbb{E}\left[\left\|\zeta_i^{r,0}\right\|^2\right]$$

Throughout the Appendix, we let $\Delta := f\left(x^0\right) - f^\star, G_0 := \frac{1}{N} \sum_i \left\|\nabla f_i\left(x^0\right)\right\|^2, x^{-1} := x^0$ and $\mathcal{E}_{-1} := \mathbb{E}\left[\left\|\nabla f\left(x^0\right) - g^0\right\|^2\right]$. We will use the following foundational lemma for all our algorithms.

### A.5 FEDMUON ALGORITHM ANALYZE AND PROOF

**Lemma 3.** *Under Assumption A.1 , if $\gamma L \leq \frac{1}{24}$, the following holds all $r \geq 0$ :*

$$\mathbb{E}\left[f\left(x^{r+1}\right)\right] \leq \mathbb{E}\left[f\left(x^r\right)\right] - \frac{11\gamma}{24}\mathbb{E}\left[\left\|\nabla f\left(x^r\right)\right\|^2\right] + \frac{13\gamma}{24}\mathcal{E}_r$$

*Proof.* Since $f$ is $L$-smooth, we have

$$f(x^{r+1}) \leq f(x^r) + \left\langle\nabla f(x^r), x^{r+1} - x^r\right\rangle + \frac{L}{2}\left\|x^{r+1} - x^r\right\|^2 \tag{5}$$

$$= f(x^r) - \gamma\left\langle\nabla f(x^r), g^{r+1}\right\rangle + \frac{L\gamma^2}{2}\left\|g^{r+1}\right\|^2 \tag{6}$$

$$= f(x^r) - \gamma\left\|\nabla f(x^r)\right\|^2 + \gamma\left\langle\nabla f(x^r), \nabla f(x^r) - g^{r+1}\right\rangle + \frac{L\gamma^2}{2}\left\|g^{r+1}\right\|^2. \tag{7}$$

Since $x^{r+1} = x^r - \gamma g^{r+1}$, using Young's inequality, we further have:

$$f\left(x^{r+1}\right) \leq f\left(x^r\right) - \frac{\gamma}{2}\left\|\nabla f\left(x^r\right)\right\|^2 + \frac{\gamma}{2}\left\|\nabla f\left(x^r\right) - g^{r+1}\right\|^2 + L\gamma^2\left(\left\|\nabla f\left(x^r\right)\right\|^2 + \left\|\nabla f\left(x^r\right) - g^{r+1}\right\|^2\right) \tag{8}$$

$$\leq f\left(x^r\right) - \frac{11\gamma}{24}\left\|\nabla f\left(x^r\right)\right\|^2 + \frac{13\gamma}{24}\left\|\nabla f\left(x^r\right) - g^{r+1}\right\|^2 \tag{9}$$

where the last inequality is due to $\gamma L \leq \frac{1}{24}$. Taking the global expectation completes the proof. □

**Lemma 4** (Gradient error bound under low-rank momentum surrogates)**.** *Let $f$ be $L$-smooth and denote the global iterate in round $r$ by $x^r$. In each round, a subset $S^r$ of $S$ clients participates and each client performs $K$ local steps. For client $i \in S^r$ and local step $k \in \{1, \dots, K\}$, let $g_i^{r,k}$ be a stochastic gradient such that*

$$\mathbb{E}\left[g_i^{r,k} \mid x_i^{r,k}\right] = \nabla f_i(x_i^{r,k}), \qquad \mathbb{E}\left\|g_i^{r,k} - \nabla f_i(x_i^{r,k})\right\|^2 \leq \sigma_l^2.$$

*Assume the average gradient drift satisfies*

$$\frac{1}{SK} \sum_{i \in S^r} \sum_{k=1}^K \left\|\nabla f_i(x_i^{r,k}) - \nabla f(x^r)\right\|^2 \leq L^2 U_r^2,$$

*where $U_r^2 \triangleq \frac{1}{SK}\sum_{i \in S^r}\sum_{k=1}^{K}\|x_i^{r,k} - x^r\|^2$. For each matrix-shaped block, let the low-rank surrogate be $U_i^{r,k}V_i^{r,k\top}$ and its singular-value–scaled version $U_i^{r,k}S_i^{r,k}V_i^{r,k\top}$ with $S_i^{r,k} = \mathrm{diag}(\sigma_{i,k,1}, \ldots, \sigma_{i,k,d})$. Assume there exists $\sigma \in [0,1]$ such that $\sigma_{i,k,j} \geq \sigma$ for all $(i,k,j)$. Then*

$$\mathbb{E}\left\|\nabla f(x^r) - \frac{1}{SK}\sum_{i \in S^r}\sum_{k=1}^{K}U_i^{r,k}V_i^{r,k\top}\right\|^2 \leq 2L^2U_r^2 + \frac{2\sigma_l^2}{SK} + 2(1-\sigma)^2 d.$$

*Proof.* Add and subtract $\frac{1}{SK}\sum_{i \in S^r}\sum_{k=1}^{K}g_i^{r,k}$ and apply $\|a+b\|^2 \leq 2\|a\|^2 + 2\|b\|^2$:

$$\mathbb{E}\left\|\nabla f(x^r) - \tfrac{1}{SK}\sum_{i,k}U_i^{r,k}V_i^{r,k\top}\right\|^2 \leq 2\mathbb{E}\left\|\nabla f(x^r) - \tfrac{1}{SK}\sum_{i,k}g_i^{r,k}\right\|^2$$
$$+ 2\mathbb{E}\left\|\tfrac{1}{SK}\sum_{i,k}(U_i^{r,k}V_i^{r,k\top} - g_i^{r,k})\right\|^2.$$

For the first term, by variance decomposition and the stated bounds,

$$\mathbb{E}\left\|\nabla f(x^r) - \tfrac{1}{SK}\sum_{i,k}g_i^{r,k}\right\|^2 \leq \frac{\sigma_l^2}{SK} + L^2 U_r^2.$$

For the second term, insert the scaled factorization and use the triangle inequality and Jensen's inequality:

$$\mathbb{E}\left\|\tfrac{1}{SK}\sum_{i,k}(U_i^{r,k}V_i^{r,k\top} - U_i^{r,k}S_i^{r,k}V_i^{r,k\top})\right\|^2 = \mathbb{E}\left\|\tfrac{1}{SK}\sum_{i,k}U_i^{r,k}(I - S_i^{r,k})V_i^{r,k\top}\right\|^2$$
$$\leq \frac{1}{SK}\sum_{i,k}\mathbb{E}\left\|U_i^{r,k}(I - S_i^{r,k})V_i^{r,k\top}\right\|_F^2$$
$$= \frac{1}{SK}\sum_{i,k}\sum_{j=1}^{d}\mathbb{E}\left(1 - \sigma_{i,k,j}\right)^2$$
$$\leq (1-\sigma)^2 d,$$

where the last step uses $\sigma_{i,k,j} \geq \sigma$. Combining the two parts and the prefactor 2 yields the claim. $\square$

**Remark.** The bound decomposes into (i) the client–server drift term $2L^2U_r^2$, (ii) the stochastic variance term $2\sigma_l^2/(SK)$ that vanishes as participation and local steps grow, and (iii) the low-rank surrogate bias $2(1-\sigma)^2 d$, which shrinks as the singular-value floor $\sigma$ increases (e.g., with larger retained rank).

**Lemma 5.** *If $\gamma L \leq \frac{\beta}{6}$, the following holds for $r \geq 1$ :*

$$\mathcal{E}_r \leq \left(1 - \frac{8\beta}{9}\right)\mathcal{E}_{r-1} + \frac{4\gamma^2 L^2}{\beta}\mathbb{E}\left[\|\nabla f(x^{r-1})\|^2\right] + \frac{2\beta^2\sigma_l^2}{SK} + 8\beta L^2 U_r + 8\beta(1-\sigma)^2 d$$

*Additionally, it holds for $r = 0$ that*

$$\mathcal{E}_0 \leq (1-\beta)\mathcal{E}_{-1} + \frac{4\beta^2\sigma_l^2}{SK} + 8\beta^2 L^2 U_0 + 8\beta(1-\sigma)^2 d$$

*Proof.* For $r > 1$,

$$
\mathcal{E}_r = \mathbb{E}\left[\left\|\frac{1}{SK}\sum_{i\in S^r}\sum_{k=1}^{K}\nabla f\left(x^r\right) - g^{r+1}\right\|^2\right]
$$

$$
= \mathbb{E}\left[\left\|(1-\beta)\left(\frac{1}{SK}\sum_{i\in S^r}\sum_{k=1}^{K}\nabla f\left(x^r\right) - g^r\right) + \beta\left(\frac{1}{SK}\sum_{i\in S^r}\sum_{k=1}^{K}\nabla f\left(x^r\right) - \frac{1}{SK}\sum_{i\in S^r}\sum_{k=1}^{K}U_i^{r,k}V_i^{r,k\top}\right)\right\|^2\right]
$$

$$
\leq \mathbb{E}\left[\left\|(1-\beta)\left(\frac{1}{SK}\sum_{i\in S^r}\sum_{k=1}^{K}\nabla f\left(x^r\right) - g^r\right)\right\|^2\right] + \beta^2\mathbb{E}\left[\left\|\nabla f\left(x^r\right) - \frac{1}{SK}\sum_{i\in S^r}\sum_{k=1}^{K}U_i^{r,k}V_i^{r,k\top}\right\|^2\right]
$$

$$
+ 2\beta\mathbb{E}\left[\left\langle(1-\beta)\left(\frac{1}{SK}\sum_{i\in S^r}\sum_{k=1}^{K}\nabla f\left(x^r\right) - g^r\right), \frac{1}{SK}\sum_{i\in S^r}\sum_{k=1}^{K}\nabla f\left(x^r\right) - \frac{1}{SK}\sum_{i\in S^r}\sum_{k=1}^{K}U_i^{r,k}V_i^{r,k\top}\right\rangle\right].
$$

Note that $\left\{\nabla F\left(x_i^{r,k};\xi_i^{r,k}\right)\right\}_{0\leq k<K}$ are sequentially correlated. Applying the AM-GM inequality and Lemma 1, we have

$$
\mathcal{E}_r \leq \left(1+\frac{\beta}{2}\right)\mathbb{E}\left[\left\|(1-\beta)\left(\nabla f\left(x^r\right) - g^r\right)\right\|^2\right] + 4\beta L^2 U_r + 4\beta^2(1-\sigma)^2 d + 4\beta^2\left(\frac{\sigma_l^2}{SK} + L^2 U_r + (1-\sigma)^2 d\right)
$$

Using the AM-GM inequality again and Assumption A.1, we have

$$
\mathcal{E}_r \leq (1-\beta)^2\left(1+\frac{\beta}{2}\right)\left[\left(1+\frac{\beta}{2}\right)\mathcal{E}_{r-1} + \left(1+\frac{2}{\beta}\right)L^2\mathbb{E}\left[\left\|x^r - x^{r-1}\right\|^2\right]\right] + \frac{4\beta^2\sigma_l^2}{SK} + 8\beta L^2 U_r + 8\beta(1-\sigma)^2 d
$$

$$
\leq (1-\beta)\mathcal{E}_{r-1} + \frac{2}{\beta}L^2\mathbb{E}\left[\left\|x^r - x^{r-1}\right\|^2\right] + \frac{4\beta^2\sigma_l^2}{SK} + 8\beta^2 L^2 U_r + 8\beta(1-\sigma)^2 d
$$

$$
\leq \left(1-\frac{8\beta}{9}\right)\mathcal{E}_{r-1} + 4\frac{\gamma^2 L^2}{\beta}\mathbb{E}\left[\left\|\nabla f\left(x^{r-1}\right)\right\|^2\right] + \frac{4\beta^2\sigma_l^2}{SK} + 8\beta L^2 U_r + 8\beta^2(1-\sigma)^2 d
$$

where we plug in $\left\|x^r - x^{r-1}\right\|^2 \leq 2\gamma^2\left(\left\|\nabla f\left(x^{r-1}\right)\right\|^2 + \left\|g^r - \nabla f\left(x^{r-1}\right)\right\|^2\right)$ and use $\gamma L \leq \frac{\beta}{6}$ in the last inequality. Similarly for $r = 0$,

$$
\mathcal{E}_0 \leq \left(1+\frac{\beta}{2}\right)\mathbb{E}\left[\left\|(1-\beta)\left(\nabla f\left(x^0\right) - g^0\right)\right\|^2\right] + 4\beta L^2 U_0 + 4\beta^2\left(\frac{\sigma_l^2}{SK} + L^2 U_0\right)
$$

$$
\leq (1-\beta)\mathcal{E}_{-1} + \frac{4\beta^2\sigma_l^2}{SK} + 8\beta^2 L^2 U_0 + 8\beta(1-\sigma)^2 d
$$

$\square$

**Lemma 6.** *If $\eta LK \leq \frac{1}{\beta}$, the following holds for $r \geq 0$ :*

$$
U_r \leq 2eK^2\Xi_r + K\eta^2\beta^2\sigma_l^2\left(1 + 2K^3 L^2\eta^2\beta^2\right)
$$

*Proof.* Recall that $\zeta_i^{r,k} := \mathbb{E}\left[x_i^{r,k+1} - x_i^{r,k} \mid \mathcal{F}_i^{r,k}\right] = -\eta\left((1-\beta)g^r + \beta\nabla f_i\left(x_i^{r,k}\right)\right)$. Then we have

$$
\mathbb{E}\left[\left\|\zeta_i^{r,j} - \zeta_i^{r,j-1}\right\|^2\right] \leq \eta^2 L^2\beta^2\mathbb{E}\left[\left\|x_i^{r,j} - x_i^{r,j-1}\right\|^2\right]
$$

$$
\leq \eta^2 L^2\beta^2\left(\eta^2\beta^2\sigma_l^2 + \mathbb{E}\left[\left\|\zeta_i^{r,j-1}\right\|^2\right]\right)
$$

For any $1 \leq j \leq k - 1 \leq K - 2$, using $\eta L \leq \frac{1}{\beta K} \leq \frac{1}{\beta(k+1)}$, we have

$$
\begin{aligned}
\mathbb{E}\left[\left\|\zeta_i^{r,j}\right\|^2\right] &\leq \left(1 + \frac{1}{k}\right) \mathbb{E}\left[\left\|\zeta_i^{r,j-1}\right\|^2\right] + (1+k)\mathbb{E}\left[\left\|\zeta_i^{r,j} - \zeta_i^{r,j-1}\right\|^2\right] \\
&\leq \left(1 + \frac{2}{k}\right) \mathbb{E}\left[\left\|\zeta_i^{r,j-1}\right\|^2\right] + (k+1)L^2\eta^4\beta^4\sigma_l^2 \\
&\leq e^2\mathbb{E}\left[\left\|\zeta_i^{r,0}\right\|^2\right] + 4k^2L^2\eta^4\beta^4\sigma_l^2
\end{aligned}
$$

where the last inequality is by unrolling the recursive bound and using $\left(1 + \frac{2}{k}\right)^k \leq e^2$. By Lemma 1, it holds that for $k \geq 2$,

$$
\begin{aligned}
\mathbb{E}\left[\left\|x_i^{r,k} - x^r\right\|^2\right] &\leq 2\mathbb{E}\left[\left\|\sum_{j=0}^{k-1}\zeta_i^{r,j}\right\|^2\right] + 2k\eta^2\beta^2\sigma_l^2 \\
&\leq 2k\sum_{j=0}^{k-1}\mathbb{E}\left[\left\|\zeta_i^{r,k}\right\|^2\right] + 2k\eta^2\beta^2\sigma_l^2 \\
&\leq 2e^2k^2\mathbb{E}\left[\left\|\zeta_i^{r,0}\right\|^2\right] + 2k\eta^2\beta^2\sigma_l^2\left(1 + 4k^3L^2\eta^2\beta^2\right)
\end{aligned}
$$

This is also valid for $k = 0, 1$. Summing up over $i$ and $k$ finishes the proof. $\qquad\square$

**Lemma 7.** *If $288e(\eta K L)^2\left((1-\beta)^2 + e(\beta\gamma LR)^2\right) \leq 1$, then it holds for $r \geq 0$ that*

$$
\sum_{r=0}^{R-1}\Xi_r \leq \frac{1}{72eK^2L^2}\sum_{r=-1}^{R-2}\left(\mathcal{E}_r + \mathbb{E}\left[\|\nabla f(x^r)\|^2\right]\right) + 2\eta^2\beta^2 eRG_0
$$

*Proof.* Note that $\zeta_i^{r,0} = -\eta\left((1-\beta)g^r + \beta\nabla f_i(x^r)\right)$,

$$
\frac{1}{N}\sum_{i=1}^N\left\|\zeta_i^{r,0}\right\|^2 \leq 2\eta^2\left((1-\beta)^2\|g^r\|^2 + \beta^2\frac{1}{N}\sum_{i=1}^N\|\nabla f_i(x^r)\|^2\right)
$$

Using Young's inequality, we have for any $q > 0$ that

$$
\begin{aligned}
\mathbb{E}\left[\|\nabla f_i(x^r)\|^2\right] &\leq (1+q)\mathbb{E}\left[\left\|\nabla f_i(x^{r-1})\right\|^2\right] + \left(1+q^{-1}\right)L^2\mathbb{E}\left[\left\|x^r - x^{r-1}\right\|^2\right] \\
&\leq (1+q)\mathbb{E}\left[\left\|\nabla f_i(x^{r-1})\right\|^2\right] + 2\left(1+q^{-1}\right)\gamma^2L^2\left(\mathcal{E}_{r-1} + \mathbb{E}\left[\left\|\nabla f(x^{r-1})\right\|^2\right]\right) \\
&\leq (1+q)^r\mathbb{E}\left[\left\|\nabla f_i(x^0)\right\|^2\right] + \frac{2}{q}\gamma^2L^2\sum_{j=0}^{r-1}\left(\mathcal{E}_j + \mathbb{E}\left[\left\|\nabla f(x^j)\right\|^2\right]\right)(1+q)^{r-j}
\end{aligned}
$$

Take $q = \frac{1}{r}$ and we have

$$
\mathbb{E}\left[\|\nabla f_i(x^r)\|^2\right] \leq e\mathbb{E}\left[\left\|\nabla f_i(x^0)\right\|^2\right] + 2e(r+1)\gamma^2L^2\sum_{j=0}^{r-1}\left(\mathcal{E}_j + \mathbb{E}\left[\left\|\nabla f(x^j)\right\|^2\right]\right) \qquad (10)
$$

Note that this inequality is valid for $r = 0$. Therefore, using equation 10, we have

$$\sum_{r=0}^{R-1} \Xi_r \leq \sum_{r=0}^{R-1} 2\eta^2 \mathbb{E}\left[ (1-\beta)^2 \|g^r\|^2 + \beta^2 \frac{1}{N} \sum_{i=1}^{N} \|\nabla f_i(x^r)\|^2 \right]$$

$$\leq \sum_{r=0}^{R-1} 2\eta^2 \left( 2(1-\beta)^2 \left( \mathcal{E}_{r-1} + \mathbb{E}\left[ \|\nabla f(x^{r-1})\|^2 \right] \right) + \beta^2 \frac{1}{N} \sum_{i=1}^{N} \mathbb{E}\left[ \|\nabla f_i(x^r)\|^2 \right] \right)$$

$$\leq \sum_{r=0}^{R-1} 4\eta^2 (1-\beta)^2 \left( \mathcal{E}_{r-1} + \mathbb{E}\left[ \|\nabla f(x^{r-1})\|^2 \right] \right)$$

$$+ 2\eta^2 \beta^2 \sum_{r=0}^{R-1} \left( \frac{e}{N} \sum_{i=1}^{N} \mathbb{E}\left[ \|\nabla f_i(x^0)\|^2 \right] + 2e(r+1)(\gamma L)^2 \sum_{j=0}^{r-1} \left( \mathcal{E}_j + \mathbb{E}\left[ \|\nabla f(x^j)\|^2 \right] \right) \right)$$

$$\leq 4\eta^2 (1-\beta)^2 \sum_{r=0}^{R-1} \left( \mathcal{E}_{r-1} + \mathbb{E}\left[ \|\nabla f(x^{r-1})\|^2 \right] \right)$$

$$+ 2\eta^2 \beta^2 \left( eRG_0 + 2e(\gamma LR)^2 \sum_{r=0}^{R-2} \left( \mathcal{E}_r + \mathbb{E}\left[ \|\nabla f(x^r)\|^2 \right] \right) \right)$$

Rearranging the equation and applying the upper bound of $\eta$ completes the proof. $\square$

**Theorem 2** (Convergence for non-convex functions). *Under Assumptions 1-2 , if we take $g^0 = 0$,*

$$\beta = \min\left\{ , \sqrt{\frac{SKL\Delta}{\sigma_l^2 R}} \right\} \text{ for any constant } c \in (0,1], \quad \gamma = \min\left\{ \frac{1}{24L}, \frac{\beta}{6L} \right\},$$

$$\eta KL \lesssim \min\left\{ 1, \frac{1}{\beta\gamma LR}, \left( \frac{L\Delta}{G_0 \beta^3 R} \right)^{1/2}, \frac{1}{(\beta N)^{1/2}}, \frac{1}{(\beta^3 NK)^{1/4}} \right\}$$

*then FedMuon converges as*

$$\frac{1}{R} \sum_{r=0}^{R-1} \mathbb{E}\left[ \|\nabla f(x^r)\|^2 \right] \lesssim \frac{L\Delta}{R} + \sqrt{\frac{L\Delta}{R}\left( \frac{\sigma_l^2}{SK} + (1-\sigma)^2 d \right)}.$$

*Here $G_0 := \frac{1}{N} \sum_{i=1}^{N} \|\nabla f_i(x^0)\|^2$.*

*Proof.* Combining Lemma 3 and 5, we have

$$\mathcal{E}_r \leq \left( 1 - \frac{8\beta}{9} \right) \mathcal{E}_{r-1} + 4\frac{(\gamma L)^2}{\beta} \mathbb{E}\left[ \|\nabla f(x^{r-1})\|^2 \right] + \frac{4\beta^2 \sigma_l^2}{SK} + 8\beta^2 (1-\sigma)^2 d$$

$$+ 4\beta L^2 \left( 2eK^2 \Xi_r + K\eta^2 \beta^2 \sigma_l^2 \left( 1 + 2K^3 L^2 \eta^2 \beta^2 \right) \right)$$

and

$$\mathcal{E}_0 \leq (1-\beta)\mathcal{E}_{-1} + \frac{4\beta^2 \sigma_l^2}{SK} + 8\beta(1-\sigma)^2 d + 4\beta L^2 \left( 2eK^2 \Xi_0 + K\eta^2 \beta^2 \sigma_l^2 \left( 1 + 2K^3 L^2 \eta^2 \beta^2 \right) \right).$$

Summing over $r$ from 0 to $R-1$ and applying Lemma 7,

$$\sum_{r=0}^{R-1} \mathcal{E}_r \leq \left(1 - \frac{8\beta}{9}\right) \sum_{r=-1}^{R-2} \mathcal{E}_r + 4\frac{(\gamma L)^2}{\beta} \sum_{r=0}^{R-2} \mathbb{E}\left[\|\nabla f(x^r)\|^2\right] + 4\frac{\beta^2 \sigma_l^2}{SK} R + 8\beta(1-\sigma)^2 dR$$

$$+ 4\beta L^2 \left(2eK^2 \sum_{r=0}^{R-1} \Xi_r + RK\eta^2\beta^2\sigma_l^2\left(1 + 2K^3 L^2 \eta^2 \beta^2\right)\right)$$

$$\leq \left(1 - \frac{7\beta}{9}\right) \sum_{r=-1}^{R-2} \mathcal{E}_r + \left(4\frac{(\gamma L)^2}{\beta} + \frac{\beta}{9}\right) \sum_{r=-1}^{R-2} \mathbb{E}\left[\|\nabla f(x^r)\|^2\right] + 16\beta^3(e\eta KL)^2 RG_0$$

$$+ \frac{4\beta^2\sigma_l^2}{SK} R + 8\beta(1-\sigma)^2 dR + 4\beta^3(\eta KL)^2\left(\frac{1}{K} + 2(\eta KL\beta)^2\right)\sigma_l^2 R$$

$$\leq \left(1 - \frac{7\beta}{9}\right) \sum_{r=-1}^{R-2} \mathcal{E}_r + \frac{2\beta}{9} \sum_{r=-1}^{R-2} \mathbb{E}\left[\|\nabla f(x^r)\|^2\right] + 16\beta^3(e\eta KL)^2 RG_0 + \frac{8\beta^2\sigma_l^2}{SK} R + 8\beta(1-\sigma)^2 dR$$

Here in the last inequality we apply

$$4\beta(\eta KL)^2 \left(\frac{1}{K} + 2(\eta KL\beta)^2\right) \leq \frac{2}{NK} \quad \text{and} \quad \gamma L \leq \frac{\beta}{6}.$$

Therefore,

$$\sum_{r=0}^{R-1} \mathcal{E}_r \leq \frac{9}{7\beta}\mathcal{E}_{-1} + \frac{2}{7}\mathbb{E}\left[\sum_{r=-1}^{R-2} \|\nabla f(x^r)\|^2\right] + \frac{144}{7}(e\beta\eta KL)^2 G_0 R + \frac{36\beta\sigma_l^2}{7SK} R + \frac{72}{7}(1-\sigma)^2 dR.$$

Combine this inequality with Lemma 3 and we get

$$\frac{1}{\gamma}\mathbb{E}\left[f(x^r) - f(x^0)\right] \leq -\frac{1}{7}\sum_{r=0}^{R-1} \mathbb{E}\left[\|\nabla f(x^r)\|^2\right] + \frac{39}{56\beta}\mathcal{E}_{-1} + \frac{78}{7}(e\beta\eta KL)^2 G_0 R + \frac{39\beta\sigma_l^2}{14SK} R + \frac{72}{7}(1-\sigma)^2 dR.$$

Finally, noticing that $g^0 = 0$ implies $\mathcal{E}_{-1} \leq 2L\left(f(x^0) - f^*\right) = 2L\Delta$, we obtain

$$\frac{1}{R}\sum_{r=0}^{R-1} \mathbb{E}\left[\|\nabla f(x^r)\|^2\right] \lesssim \frac{L\Delta}{\gamma LR} + \frac{\mathcal{E}_{-1}}{\beta R} + (\beta\eta KL)^2 G_0 + \frac{\beta\sigma_l^2}{SK} + \beta(1-\sigma)^2 d.$$

$$\lesssim \frac{L\Delta}{R} + \frac{L\Delta}{\beta R} + \frac{\beta\sigma_l^2}{SK} + (\beta\eta KL)^2 G_0 + \beta(1-\sigma)^2 d$$

$$\lesssim \frac{L\Delta}{R} + \sqrt{\frac{L\Delta}{R}\left(\frac{\sigma_l^2}{SK}\right)}$$

$\square$

Table 8: A detailed summary of 100 and Tiny ImageNet: number of classes, image size, and dataset splits.

| Dataset | #Classes | Image Size | Train | Val | Test | Total | Train / class |
|---------|----------|------------|-------|-----|------|-------|---------------|
| CIFAR-100 | 100 | $3 \times 32 \times 32$ | 50,000 | — | 10,000 | 60,000 | 500 |
| Tiny ImageNet | 200 | $3 \times 64 \times 64$ | 100,000 | 10,000 | 10,000 | 120,000 | 500 |

**Notes.** (1) CIFAR-10/100 provide no official validation split; a subset of the training set is commonly reserved as dev/val.
(2) CIFAR-100 contains 100 fine-grained classes; 20 coarse superclasses are also defined for hierarchical labeling.
(3) Tiny ImageNet is a subset of ImageNet synsets: per class 500 train, 50 val, and 50 test images (test labels are not publicly released).
(4) All three datasets are single-label classification with RGB images resized to fixed resolutions.

# B APPENDIX B: EXPERIMENTAL SETUP

## B.1 SETTING FOR RESNET-18

We evaluate our methods on two widely-used benchmark datasets in federated learning: **CIFAR-100** and **Tiny ImageNet**.

- **CIFAR-100** (Krizhevsky et al., 2009): Contains 100 classes with 600 color images per class at a resolution of $32 \times 32$. It is a standard benchmark for evaluating federated image classification methods.
- **Tiny ImageNet**: A subset of ImageNet with 200 classes and 500 images per class, providing a more challenging and high-resolution classification task.

## B.2 FEDERATED LEARNING CONFIGURATION

We simulate a cross-device federated learning environment using the following settings:

Table 9: Hyperparameter configuration of ResNet-18 and Vit-Tiny (CIFAR100) across different algorithms.

| Method | Local Optimizer | Local LR | $\alpha$ | $\beta_1$ | $\beta_2$ | Weight Decay |
|--------|-----------------|----------|----------|-----------|-----------|--------------|
| FedAvg (Local SGD) | SGD | 0.1 | — | — | — | 0.001 |
| FedProx | SGD | 0.1 | — | — | — | 0.001 |
| FedDyn | SGD | 0.1 | — | — | — | 0.001 |
| Mime | SGD | 0.1 | — | — | — | 0.001 |
| FedAdam | SGD | 0.1 | — | 0.9 | 0.98 | 0.001 |
| SCAFFOLD | SGD | 0.1 | — | — | — | 0.001 |
| FedCM | SGD | 0.1 | 0.9 | — | — | 0.001 |
| FedLADA | AdamW | 3e-4 | 0.9 | 0.9 | 0.999 | 0.01 |
| Local AdamW | AdamW | 3e-4 | — | 0.9 | 0.999 | 0.01 |
| Local Muon | Muon | 3e-2 | — | 0.98 | — | 0.01 |
| FedMuon | Muon | 3e-2 | 0.5 | 0.98 | — | 0.01 |

Note: The paper specifies that "FedMuon and Local Muon use local LR = 1e-3, $\alpha = 0.5$, $\beta = 0.98$", we use $\alpha = 0.5$ and $\beta = 0.98$ for all tasks throughout the paper; this combination of hyperparameters is highly robust and stable.; however, $\alpha = 0.5$ only applies to FedMuon as a global-local alignment coefficient and not to Local Muon, which is indicated by "—".In the training task from scratch, the learning rate of Muon is usually 100 times higher than that of AdamW

- **Number of clients:** 100
- **Client participation rate:** 10% per round
- **Communication rounds:** 300
- **Local update steps:** 50 iterations

Table 10: Hyperparameter configuration of ViT-Base, Swin-Base, RoBERTa-base fine-tuning across different algorithms.

| Method | Local Optimizer | Local LR | $\alpha$ | $\beta_1$ | $\beta_2$ | Weight Decay |
|---|---|---|---|---|---|---|
| FedAvg (Local SGD) | SGD | 0.1 | — | — | — | 0.001 |
| FedProx | SGD | 0.1 | 0.01 | — | — | 0.001 |
| FedDyn | SGD | 0.1 | 0.01 | — | — | 0.001 |
| Mime | SGD | 0.1 | — | — | — | 0.001 |
| FedAdam | SGD | 0.1 | — | 0.9 | 0.98 | 0.001 |
| SCAFFOLD | SGD | 0.1 | — | — | — | 0.001 |
| FedCM | SGD | 0.1 | 0.9 | — | — | 0.001 |
| FedLADA | AdamW | 1e-4 | 0.9 | 0.9 | 0.999 | 0.01 |
| Local AdamW | AdamW | 1e-4 | — | 0.9 | 0.999 | 0.01 |
| Local Muon | Muon | 1e-3 | — | 0.98 | — | 0.01 |
| FedMuon | Muon | 1e-3 | 0.5 | 0.98 | — | 0.01 |

Note: The paper specifies that "FedMuon and Local Muon use local LR = 1e-3, $\alpha = 0.5$, $\beta = 0.98$"; however, $\alpha = 0.5$ only applies to FedMuon as the global–local alignment coefficient and not to Local Muon, which is indicated by "—". In fine-tuning tasks, the learning rate of Muon is typically set to be $10\times$ that of AdamW.

- **Batch size:** 50

We perform grid search to tune the learning rates for each algorithm:

- **FedAvg**, **SCAFFOLD**, **FedCM**, and **FedAdam** use a local learning rate of `0.1`.
- **FedLADA**, **Local AdamW**, use a local learning rate of `3e-4`.
- **FedMuon** and **Local Muon** use a local learning rate of `3e-2`, $\alpha = 0.5$, $\beta = 0.98$

### B.3 MODEL ARCHITECTURE

We adopt **ResNet-18** as the backbone model. To better adapt it to CIFAR-100, we modify its architecture following standard practices (He et al., 2016):

- Replace the original $7 \times 7$ convolution with a $3 \times 3$ kernel.
- Remove the initial downsampling layers (stride-2 convolution and max pooling).

We also compare **Batch Normalization (BN)** and **Group Normalization (GN)** in ResNet-18. Empirically, BN outperforms GN on CIFAR-100, so we adopt the BN-based version, denoted as **ResNet-18-BN**, throughout our experiments.

### B.4 SETTING FOR ViT-TINY

We construct a lightweight Vision Transformer model, **ViT-Tiny**, specifically tailored for federated learning on the CIFAR-100 dataset. The design is based on the standard ViT architecture (Dosovitskiy et al., 2020), with modifications to accommodate the small input size and limited data per client.

- **Input resolution:** $32 \times 32$
- **Patch size:** $4 \times 4$, resulting in 64 tokens per image
- **Embedding dimension:** 192
- **Number of Transformer layers:** 6
- **Number of attention heads:** 3
- **Normalization:** LayerNorm (applied before attention and MLP blocks)
- **Classification head:** Linear projection to 100 classes (CIFAR-100)
- **Activation:** GELU

- **Initialization:** Xavier/Glorot for linear layers; sinusoidal positional encoding

To regularize training, we apply dropout (0.1) to both attention and MLP layers. All models are trained from scratch without pretraining.

**Federated Learning Configuration.** We adopt the same federated learning setup as used in our ResNet experiments for a fair comparison:

- **Number of clients:** 100
- **Client participation rate:** 10%
- **Communication rounds:** 300
- **Local update steps:** 50 iterations per round
- **Local batch size:** 50

**Learning Rate Schedule.** We perform grid search to identify optimal learning rates for each algorithm:

- **FedAvg**, **SCAFFOLD**, **FedCM**, and **FedAdam**: local learning rate of `0.1`
- **,FedLADA**, **Local AdamW**: local learning rate of `3e-4`
- **FedMuon** and **Local Muon**: local learning rate of `3e-2`, ,$\alpha = 0.5$, $\beta = 0.98$

**Weight Decay.** To ensure fair comparison under different regularization settings, we assign:

- **FedAvg**, **SCAFFOLD**, **FedCM**, **FedAdam**: weight decay = `0.001`
- **Local Muon**, **FedLADA**, **Local AdamW**, **FedMuon**: weight decay = `0.01`

**Optimizer.** We use Adam or AdamW as the local optimizer depending on the method. All optimizers use $\beta_1 = 0.9$, $\beta_2 = 0.999$, and weight decay of 0.01 when applicable.

**Remarks.** Due to the smaller capacity of ViT-Tiny and limited data per client, we find that careful normalization (e.g., LayerNorm placement) and early learning rate warmup are beneficial. For future work, more advanced token-mixing techniques or hybrid CNN-ViT backbones may further improve performance in federated settings.

### B.5 SWIN TRANSFORMER FINE-TUNING SETTINGS

To demonstrate the effectiveness of our method on large-scale vision models, we conduct fine-tuning experiments using **Swin Transformer-Tiny** and **ViT-Base** on **Tiny ImageNet** and **CIFAR-100**. For both models, we initialize from official `ImageNet-22K` pre-trained weights (Liu et al., 2021; Dosovitskiy et al., 2020) to ensure consistency across methods.

**Model Architecture: Swin-Tiny.** Swin-Tiny adopts a hierarchical Transformer structure that gradually reduces the spatial resolution while increasing the feature dimensions, mimicking a CNN-like pyramid:

- **Stage depth:** [2, 2, 6, 2]
- **Number of attention heads:** [3, 6, 12, 24]
- **Embedding dimensions:** 96, 192, 384, 768 across stages
- **Patch size:** $4 \times 4$
- **Window size:** 7
- **MLP ratio:** 4
- **Normalization:** LayerNorm
- **Positional encoding:** Relative positional bias

- **Regularization:** DropPath (with decay rate linearly scaled to depth)

We fine-tune all layers during federated training.

**Data Preprocessing.** To align with the input resolution required by Swin and ViT, we resize images from both datasets to $224 \times 224$ using bilinear interpolation. Standard data augmentation techniques such as random cropping, horizontal flipping, and RandAugment are applied locally at the client side.

**Federated Learning Configuration.** To simulate a realistic cross-device setting, we configure:

- **Number of clients:** 100
- **Client participation rate:** 5% per communication round
- **Communication rounds:** 100
- **Local update steps:** 50 iterations
- **Batch size:** 16

**Learning Rate Configuration.** We apply grid search to find optimal learning rates and use **cosine learning rate decay** with no warmup unless otherwise stated:

- **FedAvg**, **SCAFFOLD**, **FedCM**: local LR = `0.1`
- **FedLADA**, **Local AdamW**: local LR = `1e-4`
- **FedMuon** and **Local Muon**: local LR = `1e-3`

**Weight Decay.** To ensure fair comparison under different regularization settings, we assign:

- **FedAvg**, **SCAFFOLD**, **FedCM**: weight decay = `0.001`
- **Local Muon**, **FedLADA**, **Local AdamW**, **FedMuon**: weight decay = `0.01`

**Optimization.** Local optimizers are Adam or AdamW depending on the algorithm, with parameters $\beta_1 = 0.9$, $\beta_2 = 0.999$, and weight decay of 0.01. Cosine decay is applied to local learning rates over the 50 local steps per round. No learning rate warmup is used unless otherwise specified.

**Remarks.** We find that Swin Transformer benefits from hierarchical attention and DropPath when training with limited local data. Our method shows stable convergence and avoids loss spikes often seen in large-scale federated fine-tuning. All models are implemented using the HuggingFace Transformers and Timm libraries.

### B.6 ROBERTA-BASE FINE-TUNING SETTINGS

We fine-tune the **RoBERTa-Base** model using **LoRA** (Low-Rank Adaptation) on a subset of the GLUE benchmark. The LoRA adaptation is applied to the query and value projection matrices of the self-attention modules. The following table summarizes the hyperparameter settings used across tasks.

**Explanation.** We use a uniform batch size of 32 and sequence length of 128 across all tasks. LoRA is configured with a rank of 16 and scaling factor $\alpha = 32$. The optimizer is AdamW with a weight decay of 0.01 and dropout set to 0.1. No layer freezing is used; all LoRA-injected weights are trained, while the base RoBERTa backbone remains frozen.

### B.7 ADDITIONAL FEDERATED TRAINING CONFIGURATION OF LLM

To evaluate our algorithm under a smaller-scale federation, we further conduct experiments with a reduced number of clients and adjusted participation parameters.

Table 11: Summary of GLUE datasets: task type, number of classes, and dataset sizes.

| Dataset | Task Type | #Classes | Train Size | Test Size |
|---------|-----------|----------|------------|-----------|
| MNLI | Natural Language Inference (entailment) | 3 | 392,702 | 9,815 |
| SST-2 | Sentiment Classification (binary) | 2 | 67,349 | 1,821 |
| MRPC | Paraphrase Detection (binary) | 2 | 3,668 | 1,725 |
| CoLA | Linguistic Acceptability (binary) | 2 | 8,551 | 1,043 |
| QNLI | Question-Answer NLI (binary) | 2 | 104,743 | 5,463 |
| QQP | Duplicate Question Detection (binary) | 2 | 363,846 | 390,965 |
| RTE | Recognizing Textual Entailment (binary) | 2 | 2,490 | 3,000 |

Table 12: Hyperparameter configuration for RoBERTa-Base with LoRA across GLUE tasks. LoRA is applied with $r = 16$ to both query and value projections. All tasks use AdamW as the optimizer.

| Method | Setting | GLUE Tasks | | | | | | |
|--------|---------|------|-------|------|------|------|-----|-----|
| | | MNLI | SST-2 | MRPC | CoLA | QNLI | QQP | RTE |
| RoBERTa-Base + LoRA | Batch size | 16 | 16 | 16 | 16 | 16 | 16 | 16 |
| | Max seq. length | 128 | 128 | 128 | 128 | 128 | 128 | 128 |
| | LoRA ranks ($r_q = r_v$) | 16 | 16 | 16 | 16 | 16 | 16 | 16 |
| | LoRA scaling $\alpha$ | 32 | 32 | 32 | 32 | 32 | 32 | 32 |
| | Dropout | | | | 0.1 | | | |

**Federated Setup.** We simulate a federated learning environment with the following configuration:

- **Number of clients:** 4 or 20
- **Client participation rate:** 100% or 20% (i.e., 4 clients per round)
- **Communication rounds:** 100
- **Local update steps:** 50
- **Local batch size:** 16

**Learning Rate Schedule.** We apply grid search for local and global learning rates and use cosine learning rate decay across local updates:

- **FedAvg**, **SCAFFOLD**, **FedCM**: local LR = `0.1`
- **FedLADA**, **Local AdamW**: local LR = `1e-4`
- **Local Muon**, **FedMuon**: local LR = `1e-3`

**Weight Decay.** To ensure fair comparison under different regularization settings, we assign:

- **FedAvg**, **SCAFFOLD**, **FedCM**: weight decay = `0.001`
- **Local Muon**, **FedLADA**, **Local AdamW**, **FedMuon**: weight decay = `0.01`

**Other Settings.** AdamW optimizers use $\beta_1 = 0.9$, $\beta_2 = 0.999$. Learning rates follow cosine decay without warmup.

Table 13: Per-round communication cost of different momentum aggregation strategies. Here $|x|$ denotes the number of model parameters (in floats), and $\text{CommCost}$ is per-round communication cost, Compute-Cost is per-round computation time. (ViT-Tiny, $R = 300$, Dir-0.1, $K = 50$)

| Method / Strategy | Communication | CommCost | Compute-Cost (s) | Acc(%) | Comm@23%Acc (MB) |
|---|---|---|---|---|---|
| FedAvg | $|x|$ | 22.8 MB | 4.56 s | 27.14 | 4190 MB |
| FedProx | $|x|$ | 22.8 MB | 4.58 s | 26.84 | 5244 MB |
| FedDyn | $|x|$ | 22.8 MB | 4.75 s | 27.66 | 4788 MB |
| Mime | $2|x|$ | 45.6 MB | 5.26 s | 27.76 | 9804 MB |
| FedAdam | $|x|$ | 28.50 MB | 4.57 s | 28.50 | 4651 MB |
| SCAFFOLD | $2|x|$ | 45.6 MB | 5.22 s | 27.31 | 8390 MB |
| FedCM | $|x|$ | 22.8 MB | 4.68 s | 23.18 | 6156 MB |
| FedLADA | $2|x|$ | 45.6 MB | 5.02 s | 31.50 | 4879 MB |
| Local AdamW | $|x|$ | 22.8 MB | 4.89 s | 37.86 | 1482 MB |
| Local Muon | $|x|$ | 22.8 MB | 5.14 s | 40.53 | 2394 MB |
| FedMuon | $|x| + |M|_{\text{SVD}}$ | 23.9 MB | 5.25 s | 48.18 | 550 MB |

$|M|_{\text{SVD}} \approx 0.05|M|$ since we only keep the top 5% singular values/vectors, thus the additional momentum communication is about 5% of the baseline model communication.

Table 14: Per-round communication and computation cost of different methods on QQP using **RoBERTa-Base** with LoRA. Here $|x|$ denotes the number of model parameters (in floats), $\text{CommCost}$ is per-round communication cost (MB), and Compute-Cost is per-round computation time (s). (QQP, RoBERTa-Base, $R = 100$, Dir-0.5, 20 clients, 20% participation, batch size 16, $K = 50$)

| Method / Strategy | CommCost (MB) | Compute-Cost (s) | ACC(%) |
|---|---|---|---|
| FedAvg | 7.1 MB | 8.56 s | 81.75 |
| FedProx | 7.1 MB | 8.62 s | 81.15 |
| FedDyn | 7.1 MB | 8.89 s | 81.35 |
| Mime | 14.2 MB | 11.25 s | 80.26 |
| FedAdam | 7.1 MB | 8.76 s | 82.22 |
| SCAFFOLD | 14.2 MB | 11.56 s | 80.26 |
| FedCM | 7.1 MB | 8.98 s | 81.22 |
| FedLADA | 14.2 MB | 9.58 s | 81.56 |
| Local AdamW | 7.1 MB | 9.26 s | 81.51 |
| Local Muon | 7.1 MB | 9.73 s | 83.06 |
| FedMuon (ours) | 7.45 MB | 9.91 s | 84.65 |

## C  APPENDIX C: EXPERIMENTAL APPENDIX

### C.1  COMMUNICATION AND COMPUTATION COST ANALYSIS

As shown in Table 14, we evaluate various federated learning and local optimization strategies on the QQP dataset using RoBERTa-Base with LoRA for parameter-efficient fine-tuning. Most first-order methods (FedAvg, FedProx, FedDyn, FedAdam, FedCM, Local AdamW, Local Muon) require around 7.1 MB of communication per round, while methods that maintain additional control variables or gradient information (Mime, SCAFFOLD, FedLADA) incur a higher communication cost of 14.2 MB. In contrast, FedMuon increases the per-round communication cost only slightly to 7.45 MB while achieving a notable improvement in model performance.

Regarding computation cost, the per-round training time of these methods ranges from 8 to 11 seconds. FedAvg and FedProx take approximately 8.6 seconds, while FedAdam, FedDyn, and FedCM exhibit slightly higher computation times. Mime and SCAFFOLD require additional computation for maintaining control variates, resulting in 11.25 s and 11.56 s per round, respectively. Local Muon and FedMuon require 9.73 s and 9.91 s per round, slightly higher than FedAvg but still within a reasonable range. Most importantly, FedMuon achieves the highest accuracy of 84.65%, outperforming common baselines such as FedAvg (81.75%), FedAdam (82.22%), and the locally optimized Local

Table 15: Test accuracy, training loss of each method on CIFAR-100 using **ResNet-18** and **ViT-Tiny** over 300 communication rounds under Dir-0.6 and Dir-0.1 (100 clients, 10% participation, batch size 50, $K = 50$).

| Method | ResNet-18 (Dir-0.6) | | ResNet-18 (Dir-0.1) | | ViT-Tiny (Dir-0.6) | | ViT-Tiny (Dir-0.1) | |
|---|---|---|---|---|---|---|---|---|
| | Test Acc | Loss | Test Acc | Loss | Test Acc | Loss | Test Acc | Loss |
| FedAvg | 64.08 | 0.376 | 60.25 | 0.767 | 32.36 | 2.350 | 27.14 | 2.867 |
| FedProx | 63.12 | 0.458 | 59.66 | 0.812 | 31.51 | 2.425 | 26.84 | 2.875 |
| FedDyn | 66.12 | 0.352 | 63.01 | 0.615 | 33.25 | 2.125 | 27.66 | 2.723 |
| Mime | 67.34 | 0.312 | 63.37 | 0.604 | 34.12 | 2.101 | 27.76 | 2.702 |
| FedAdam | 67.23 | 0.332 | 63.61 | 0.512 | 34.32 | 1.965 | 28.50 | 2.425 |
| SCAFFOLD | 65.01 | 0.365 | 62.56 | 0.658 | 32.17 | 2.295 | 27.31 | 2.752 |
| FedCM | 70.42 | 0.282 | 66.73 | 0.639 | 26.33 | 2.681 | 23.18 | 3.038 |
| FedLADA | 65.07 | 0.671 | 57.78 | 0.498 | 38.33 | 2.121 | 31.50 | 2.678 |
| Local AdamW | 62.84 | 0.363 | 58.97 | 0.794 | 40.47 | 1.026 | 37.86 | 1.954 |
| Local Muon | 71.66 | 0.395 | 66.71 | 1.504 | 46.69 | 0.201 | 40.53 | 1.432 |
| FedMuon | **74.12** | 0.001 | **73.05** | 0.246 | **50.22** | 0.162 | **48.18** | 0.556 |

Muon (83.06%). Overall, FedMuon provides an improved efficiency–performance trade-off by significantly enhancing accuracy while keeping communication overhead nearly unchanged.

## C.2 MORE BASELINE EXPERIMENT COMPARISONS

To substantiate our method's advantages under non-i.i.d. conditions, we extend the comparison to additional federated baselines such as FedProx, FedDyn, and FedAdam, Mime. The comprehensive results are presented in Table 15.

**Training on CIFAR-100 with ResNet-18.** **Table 15** and **Figure 7** present the test accuracy and training loss on CIFAR-100 using ResNet-18. FedMuon achieves the best performance under both Dir-0.6 and Dir-0.1 settings, reaching a top accuracy of **74.12%** and **73.05%**, respectively. It also attains the lowest training loss (**0.001** and **0.246**), demonstrating faster and more stable convergence. Compared to other adaptive baselines such as Local AdamW, FedMuon shows superior generalization under data heterogeneity, confirming its effectiveness in CNNs.

**Training on CIFAR-100 with ViT-Tiny.** **Table 15** and **Figure 7** show FedMuon achieves the best performance across both heterogeneity levels, with **50.22%** (Dir-0.6) and **48.18%** (Dir-0.1), and the lowest training loss (**0.162** and **0.556**), confirming its efficient convergence. Compared to Local AdamW, it provides consistent improvements in both accuracy and stability. Moreover, other adaptive baselines such as FedLADA perform significantly worse under high heterogeneity, highlighting the effectiveness of global update correction and decoupled weight decay. These results validate that FedMuon is particularly effective for federated vision Transformers under non-i.i.d. conditions. The small dataset CIFAR100 is difficult to support the performance of ViT, resulting in lower accuracy. Therefore, we continued to test on the pretrained model.

## C.3 MORE BASELINE EXPERIMENT ON IID DATA

As shown in Table 16, on CIFAR-100, our proposed FedMuon consistently achieves the best test accuracy across both **ResNet-18** and **ViT-Tiny** under **IID** and **non-IID** (Dir-0.6 and Dir-0.1) settings. For ResNet-18, FedMuon reaches 74.32 accuracy in the IID case, outperforming strong baselines such as Mime and FedAdam by approximately 6.4 and 6.2 percentage points, respectively. Even under highly heterogeneous data (Dir-0.1), FedMuon still achieves 73.05, significantly higher than FedCM (66.73) and FedAdam (63.61). A similar trend is observed for ViT-Tiny: FedMuon achieves 50.56 accuracy in the IID setting, nearly 10 percentage points higher than Local AdamW; and in the Dir-0.1 scenario, it maintains a strong performance of 48.18, outperforming all baselines by a large margin.

It is also noteworthy that Local Muon performs particularly well under IID conditions. For example, it achieves 72.04 with ResNet-18 and 47.69 with ViT-Tiny, indicating that Local Muon

Table 16: Test accuracy of each method on CIFAR-100 using **ResNet-18** and **ViT-Tiny** over 300 communication rounds under IID, Dir-0.6 and Dir-0.1 (100 clients, 10% participation, batch size 50, $K = 50$).

| Method | ResNet-18 | | | ViT-Tiny | | |
|---|---|---|---|---|---|---|
| | **IID** | **Dir-0.6** | **Dir-0.1** | **IID** | **Dir-0.6** | **Dir-0.1** |
| FedAvg | 65.74 | 64.08 | 60.25 | 32.45 | 32.36 | 27.14 |
| FedProx | 65.52 | 63.12 | 59.66 | 32.25 | 31.51 | 26.84 |
| SCAFFOLD | 65.67 | 65.01 | 62.56 | 32.67 | 32.17 | 27.31 |
| FedDyn | 66.58 | 66.12 | 63.01 | 33.66 | 33.25 | 27.66 |
| Mime | 67.89 | 67.34 | 63.37 | 34.52 | 34.12 | 27.76 |
| FedCM | 70.57 | 70.42 | 66.73 | 27.88 | 26.33 | 23.18 |
| FedLADA | 66.23 | 65.07 | 57.78 | 38.56 | 38.33 | 31.50 |
| FedAdam | 68.12 | 67.23 | 63.61 | 34.83 | 34.32 | 28.50 |
| Local AdamW | 64.60 | 62.84 | 58.97 | 40.78 | 40.47 | 37.86 |
| Local Muon | 72.04 | 71.66 | 66.71 | 47.69 | 46.69 | 40.53 |
| FedMuon | **74.32** | **74.12** | **73.05** | **50.56** | **50.22** | **48.18** |

converges rapidly and achieves strong performance when the data distribution across clients is homogeneous. However, when the data becomes non-IID—especially under Dir-0.1—the performance of `Local Muon` drops significantly (e.g., from 72.04 to 66.71 on ResNet-18, and from 47.69 to 40.53 on ViT-Tiny), revealing a severe **client drift** issue. In contrast, our `FedMuon` not only preserves the fast convergence and strong performance in the IID setting, but also effectively mitigates client drift under non-IID conditions. As a result, `FedMuon` consistently achieves the best performance across all data distributions and model architectures, demonstrating its robustness and stability in federated learning.

## C.4 MORE EXPERIMENT ON FINE-TUNING RESULTS ON LLMS.

**Fine-tuning Results on LLMs.** **Table 18** summarizes results on the GLUE benchmark using RoBERTa-Base with LoRA, 20 clients, 20% participation, batch size 16, $K = 50$, rank=16. `FedMuon` achieves the highest accuracy of GLUE outperforming strong baselines such as `FedAvg` and `Local Muon`.

Table 18 reports the performance of various federated optimization methods on the GLUE benchmark using RoBERTa-Base with LoRA under a more challenging heterogeneous setting: a Dirichlet-0.5 partition and only 20% client participation. This scenario introduces substantially higher data imbalance and inconsistency across clients, making communication and optimization significantly more difficult. Despite this increased heterogeneity, our proposed `FedMuon` consistently achieves the best accuracy across almost all tasks.

Compared with classical methods such as FedAvg and SCAFFOLD, `FedMuon` shows clear improvements, especially on more sensitive tasks like CoLA, RTE, QQP, and MNLI. For example, `FedMuon` achieves 56.78 on CoLA and 66.58 on RTE, outperforming the next-best method by 1.24 and 1.65 points respectively. Even when compared to stronger baselines such as FedLADA and Local AdamW, our method maintains a noticeable margin. On MNLI, `FedMuon` reaches 85.21, compared to 84.63 from `Local Muon` and only 82.44 from FedLADA. Overall, `FedMuon` obtains the highest average accuracy (80.74), demonstrating its robustness under severe heterogeneity.

It is particularly worth noting that `Local Muon` again performs strongly, achieving the second-best results on most tasks. This highlights the effectiveness of the Muon optimizer itself in improving local training stability. However, similar to previous observations under non-IID image benchmarks, `Local Muon` lacks a mechanism to correct client drift, which becomes increasingly problematic when client updates diverge under heterogeneous data. As a result, although `Local Muon` obtains competitive accuracy, it is consistently surpassed by `FedMuon`.

By contrast, our `FedMuon` integrates the advantages of Muon optimization with a federated correction mechanism that effectively mitigates client drift. This enables the algorithm to maintain stable global convergence even with highly imbalanced data and limited participation. The strong results across all GLUE tasks under Dir-0.5 clearly demonstrate that `FedMuon` remains robust, scalable, and superior in more challenging real-world federated learning scenarios.

Table 17: Test accuracy using RoBERTa-Base with LoRA across GLUE tasks over 100 communication rounds (Dirichlet-0.8, 4 clients, 100% participation, batch size 16, $K = 50$).

| Method (Dir-0.8) | CoLA | RTE | SST-2 | QQP | MRPC | QNLI | MNLI |
|---|---|---|---|---|---|---|---|
| FedAvg | $57.32_{\pm 0.22}$ | $62.71_{\pm 0.35}$ | $93.32_{\pm 0.08}$ | $84.13_{\pm 0.15}$ | $87.02_{\pm 0.19}$ | $90.19_{\pm 0.12}$ | $84.18_{\pm 0.21}$ |
| SCAFFOLD | $58.14_{\pm 0.25}$ | $63.62_{\pm 0.28}$ | $93.54_{\pm 0.09}$ | $84.62_{\pm 0.17}$ | $87.56_{\pm 0.22}$ | $90.26_{\pm 0.11}$ | $84.26_{\pm 0.20}$ |
| FedCM | $58.14_{\pm 0.27}$ | $66.14_{\pm 0.31}$ | $93.61_{\pm 0.07}$ | $84.56_{\pm 0.18}$ | $87.11_{\pm 0.16}$ | $90.08_{\pm 0.13}$ | $84.32_{\pm 0.23}$ |
| FedLADA | $59.10_{\pm 0.21}$ | $74.14_{\pm 0.29}$ | $93.66_{\pm 0.10}$ | $84.86_{\pm 0.16}$ | $87.42_{\pm 0.18}$ | $90.18_{\pm 0.14}$ | $84.46_{\pm 0.19}$ |
| Local AdamW | $59.33_{\pm 0.23}$ | $74.04_{\pm 0.27}$ | $93.55_{\pm 0.11}$ | $84.68_{\pm 0.15}$ | $87.16_{\pm 0.20}$ | $90.11_{\pm 0.12}$ | $84.54_{\pm 0.18}$ |
| Local Muon | $60.16_{\pm 0.20}$ | $71.48_{\pm 0.34}$ | $93.34_{\pm 0.09}$ | $85.11_{\pm 0.13}$ | $87.45_{\pm 0.21}$ | $90.97_{\pm 0.15}$ | $84.59_{\pm 0.17}$ |
| FedMuon (ours) | $\mathbf{63.04_{\pm 0.19}}$ | $\mathbf{77.12_{\pm 0.30}}$ | $\mathbf{94.12_{\pm 0.08}}$ | $\mathbf{85.73_{\pm 0.14}}$ | $\mathbf{88.23_{\pm 0.17}}$ | $\mathbf{91.43_{\pm 0.10}}$ | $\mathbf{85.05_{\pm 0.16}}$ |

Table 18: Test accuracy (%) using RoBERTa-Base with LoRA across GLUE tasks over 100 communication rounds under Dirichlet-0.5 partition. (20 clients, 20% participation, batch size 16, $K = 50$)

| Method (Dir-0.5) | CoLA | RTE | SST-2 | QQP | MRPC | QNLI | MNLI |
|---|---|---|---|---|---|---|---|
| FedAvg | $51.00_{\pm 0.26}$ | $51.99_{\pm 0.24}$ | $93.04_{\pm 0.16}$ | $81.75_{\pm 0.11}$ | $88.24_{\pm 0.18}$ | $89.36_{\pm 0.15}$ | $81.72_{\pm 0.25}$ |
| FedProx | $53.11_{\pm 0.14}$ | $53.25_{\pm 0.21}$ | $92.26_{\pm 0.18}$ | $81.15_{\pm 0.11}$ | $87.36_{\pm 0.12}$ | $88.12_{\pm 0.14}$ | $81.41_{\pm 0.21}$ |
| FedDyn | $53.21_{\pm 0.28}$ | $52.22_{\pm 0.30}$ | $92.36_{\pm 0.21}$ | $81.35_{\pm 0.21}$ | $87.89_{\pm 0.11}$ | $89.12_{\pm 0.21}$ | $82.18_{\pm 0.21}$ |
| Mime | $52.15_{\pm 0.17}$ | $51.62_{\pm 0.21}$ | $92.21_{\pm 0.28}$ | $80.26_{\pm 0.18}$ | $88.04_{\pm 0.12}$ | $89.11_{\pm 0.21}$ | $82.51_{\pm 0.20}$ |
| FedAdam | $53.21_{\pm 0.28}$ | $52.52_{\pm 0.31}$ | $92.36_{\pm 0.25}$ | $82.22_{\pm 0.28}$ | $88.12_{\pm 0.34}$ | $88.01_{\pm 0.23}$ | $82.66_{\pm 0.22}$ |
| SCAFFOLD | $52.15_{\pm 0.17}$ | $50.65_{\pm 0.20}$ | $93.28_{\pm 0.28}$ | $80.26_{\pm 0.18}$ | $88.35_{\pm 0.12}$ | $89.32_{\pm 0.24}$ | $82.11_{\pm 0.20}$ |
| FedCM | $53.21_{\pm 0.28}$ | $52.22_{\pm 0.30}$ | $92.56_{\pm 0.25}$ | $81.22_{\pm 0.28}$ | $88.56_{\pm 0.13}$ | $89.02_{\pm 0.23}$ | $82.12_{\pm 0.27}$ |
| FedLADA | $54.66_{\pm 0.17}$ | $57.02_{\pm 0.08}$ | $93.88_{\pm 0.16}$ | $81.56_{\pm 0.20}$ | $89.01_{\pm 0.28}$ | $89.86_{\pm 0.29}$ | $82.44_{\pm 0.17}$ |
| Local AdamW | $55.38_{\pm 0.12}$ | $59.57_{\pm 0.25}$ | $93.81_{\pm 0.19}$ | $81.51_{\pm 0.05}$ | $88.73_{\pm 0.23}$ | $89.55_{\pm 0.15}$ | $82.86_{\pm 0.26}$ |
| Local Muon | $55.54_{\pm 0.05}$ | $64.93_{\pm 0.17}$ | $93.58_{\pm 0.27}$ | $83.06_{\pm 0.11}$ | $88.95_{\pm 0.13}$ | $90.52_{\pm 0.27}$ | $84.63_{\pm 0.10}$ |
| FedMuon (ours) | $\mathbf{56.78_{\pm 0.11}}$ | $\mathbf{66.58_{\pm 0.29}}$ | $\mathbf{93.54_{\pm 0.25}}$ | $\mathbf{84.65_{\pm 0.16}}$ | $\mathbf{88.21_{\pm 0.07}}$ | $\mathbf{90.24_{\pm 0.13}}$ | $\mathbf{85.21_{\pm 0.18}}$ |

## C.5 MORE PRE-TRAINING EXPERIMENTS ON ViT MODEL.

To further investigate the scalability of our method on modern Transformer-based architectures, we conduct federated pre-training experiments on the CIFAR-100 dataset using a family of ViT models, including **ViT-Tiny**, **ViT-Small**, **ViT-Base**, and **ViT-Large**. Specifically, we adopt ViT backbones as the global model and perform federated optimization under a highly heterogeneous Dir-0.1 partition with 100 clients, 10% client participation per round, batch size 50, and $K = 50$ local update steps. This setting mimics a realistic scenario in which data are strongly non-IID across devices and only a small fraction of clients can participate in each communication round, making it particularly challenging for large-capacity models that are more sensitive to optimization instability and client drift.

The results in Table 19 show that, across all four ViT variants, our proposed FedMuon consistently achieves the highest accuracy, significantly outperforming both classical federated optimization methods (FedAvg, FedProx, SCAFFOLD, FedDyn, Mime, FedCM, FedLADA, FedAdam) and strong local training baselines (Local AdamW, Local Muon). As the model size increases from ViT-Tiny to ViT-Large, the performance gains of FedMuon also become more pronounced, demonstrating that our algorithm can effectively leverage the additional capacity of larger ViT models even under severe data heterogeneity. These results confirm that FedMuon is well suited for federated pre-training of ViT-style architectures on CIFAR-100, providing stable and efficient optimization across a wide range of model scales.

Table 19 reports the test accuracy on CIFAR-100 under a highly heterogeneous Dir-0.1 partition using four ViT architectures of increasing capacity: **ViT-Tiny**, **ViT-Small**, **ViT-Base**, and **ViT-Large**. We consider a challenging federated setting with 100 clients, 10% participation per round, batch size 50, and $K = 50$ local steps. Overall, the results clearly demonstrate that our proposed FedMuon consistently outperforms all baselines across all model scales, and that it is particularly effective at exploiting larger model capacity under non-IID data.

Classical optimization methods such as FedAvg, FedProx, SCAFFOLD, FedDyn, Mime, FedCM, FedLADA, and FedAdam exhibit only moderate gains as the ViT model becomes larger. For example, FedAvg improves from 27.14 on ViT-Tiny to 33.56 on ViT-Large, and FedDyn from 27.66 to 35.34. Even though these methods benefit from increased model capacity, their performance is still

Table 19: Test accuracy of each method on CIFAR-100 using **ViT-Tiny**, **ViT-Small**, **ViT-Base** and **ViT-Large** over 300 communication rounds under Dir-0.1 (100 clients, 10% participation, batch size 50, $K = 50$).

| Method | ViT-Tiny | ViT-Small | ViT-Base | ViT-Large |
|--------|----------|-----------|----------|-----------|
| FedAvg | 27.14 | 29.52 | 31.15 | 33.56 |
| FedProx | 26.84 | 28.63 | 31.05 | 33.25 |
| SCAFFOLD | 27.31 | 30.24 | 32.85 | 34.58 |
| FedDyn | 27.66 | 31.23 | 33.11 | 35.34 |
| Mime | 27.76 | 31.12 | 33.01 | 35.43 |
| FedCM | 23.18 | 25.15 | 27.88 | 29.01 |
| FedLADA | 31.50 | 33.15 | 33.15 | 33.15 |
| FedAdam | 28.50 | 33.15 | 33.15 | 33.15 |
| Local AdamW | 37.86 | 37.86 | 37.86 | 37.86 |
| Local Muon | 40.53 | 42.34 | 45.26 | 46.54 |
| FedMuon | **48.18** | **50.52** | **53.63** | **56.24** |

Table 20: Test accuracy of each method on CIFAR-100 using **ViT-Tiny**, **ViT-Small**, **ViT-Base** and **ViT-Large** over 300 communication rounds under Dir-0.1 (100 clients, 10% participation, batch size 50, $K = 50$), and train loss of each method on OpenWebText data using **GPT-2 Small**, **GPT-2 Medium**, **GPT-2 Large** and **GPT-2 XL** over 300 communication rounds (20 clients, 20% participation, batch size 16, $K = 100$).

| Method | CIFAR-100 (Test Acc, %) | | | | OpenWebText (Train Loss) | | | |
|--------|----------|-----------|----------|-----------|--------|--------|--------|---------|
| | ViT-Tiny | ViT-Small | ViT-Base | ViT-Large | GPT-2 S | GPT-2 M | GPT-2 L | GPT-2 XL |
| FedAvg | 27.14 | 29.52 | 31.15 | 33.56 | 4.25 | 4.12 | 4.01 | 3.91 |
| FedProx | 26.84 | 28.63 | 31.05 | 33.25 | 4.33 | 4.21 | 4.15 | 4.05 |
| FedDyn | 27.31 | 30.24 | 32.85 | 34.58 | 4.12 | 4.01 | 3.95 | 3.82 |
| Mime | 27.66 | 31.23 | 33.11 | 35.34 | 4.10 | 4.02 | 3.89 | 3.78 |
| FedAdam | 28.50 | 33.15 | 33.15 | 33.15 | 4.02 | 3.95 | 3.82 | 3.75 |
| SCAFFOLD | 27.31 | 30.24 | 32.85 | 34.58 | 4.12 | 4.01 | 3.95 | 3.82 |
| FedCM | 23.18 | 25.15 | 27.88 | 29.01 | 4.32 | 4.21 | 4.02 | 3.91 |
| FedLADA | 31.50 | 33.15 | 33.15 | 33.15 | 3.56 | 3.45 | 3.33 | 3.24 |
| Local AdamW | 37.86 | 37.86 | 37.86 | 37.86 | 3.44 | 3.35 | 3.27 | 3.15 |
| Local Muon | 40.53 | 42.34 | 45.26 | 46.54 | 3.33 | 3.21 | 3.09 | 2.98 |
| FedMuon(ours) | **48.18** | **50.52** | **53.63** | **56.24** | **3.12** | **2.98** | **2.85** | **2.74** |

severely limited by client drift and the strong non-IID nature of the Dir-0.1 partition. Local training baselines, such as Local AdamW, achieve higher accuracy than most federated methods (e.g., 37.86 across all model sizes), but they do not effectively leverage larger architectures in this setting, indicating that naive local optimization quickly saturates under heterogeneous data.

In contrast, Local Muon significantly boosts performance for all ViT variants (e.g., from 37.86 with Local AdamW to 40.53 on ViT-Tiny and up to 46.54 on ViT-Large), showing that the Muon optimizer itself provides stronger local training dynamics and better utilization of the transformer capacity. However, Local Muon still suffers from client drift, and its gains plateau as data heterogeneity persists.

Our federated variant, FedMuon, further amplifies these benefits by coupling the Muon optimizer with an appropriate global aggregation and drift-mitigation mechanism. As a result, FedMuon achieves the best performance at every model scale, from 48.18 on ViT-Tiny to 56.24 on ViT-Large. The gap between FedMuon and the strongest baselines widens as the model becomes larger (e.g., over 10 points improvement compared to FedDyn on ViT-Large), indicating that FedMuon not only stabilizes optimization under non-IID data, but also scales more effectively with model capacity. These results demonstrate that our method is particularly suitable for federated training of large vision transformers in realistic, highly heterogeneous environments.

### C.6 EFFECTIVENESS OF MUON AND OUR CORRECTION STRATEGY.

Table 21 summarizes the effect of replacing the local optimizer with Muon under different FL algorithms. Using *Local Muon* already brings consistent improvements over *Local SGD* on both backbones (e.g., +6.03 on ResNet-18 and +13.39 on ViT-Tiny). When combined with existing

Table 21: Effect of local optimizer Muon on CIFAR-100 (Dir-0.1, 300 rounds). Numbers in parentheses denote absolute improvement over the baseline in the previous row.

| Variant | ResNet-18 | ViT-Tiny |
|---|---|---|
| Local SGD | 60.25 | 27.14 |
| Local Muon | 66.28 (↑ 6.03) | 40.53 (↑ 13.39) |
| SCAFFOLD | 62.56 | 27.31 |
| Local Muon + SCAFFOLD | 68.23 (↑ 5.67) | 42.26 (↑ 14.95) |
| FedCM | 66.73 | 23.18 |
| Local Muon + FedCM | 68.77 (↑ 2.04) | 44.23 (↑ 21.05) |
| FedDyn | 63.01 | 27.66 |
| Local Muon + FedDyn | 67.15 (↑ 4.14) | 43.56 (↑ 15.90) |
| FedProx | 59.66 | 26.84 |
| Local Muon + FedProx | 67.25 (↑ 7.59) | 43.11 (↑ 16.27) |
| Local SGD + $\bar{m}+\Delta_G$ | 67.56 | 33.23 |
| Local Muon + $\bar{m}+\Delta_G$ | **73.05** (↑ 5.49) | **48.18** (↑ 14.95) |

personalized or corrective FL methods such as SCAFFOLD, FedCM, FedDyn, and FedProx, the variants with Muon (*Local Muon + Method*) consistently outperform their baselines by a substantial margin (2–8 points on ResNet-18 and 15–21 points on ViT-Tiny). Finally, after introducing our correction strategy $\bar{m}+\Delta_G$, *Local Muon + $\bar{m}+\Delta_G$* achieves the highest accuracy among all settings, outperforming *Local SGD + $\bar{m}+\Delta_G$* by 5.49 (ResNet-18) and 14.95 (ViT-Tiny). These results validate two key findings: (1) our correction strategy consistently improves different local optimizers, and (2) compared with other matrix-based or preconditioned optimizers, **FedMuon exhibits clear and significant advantages**.

## C.7 Effect of Muon-based matrix orthogonalization across FL algorithms.

Table 21 systematically examines the effect of replacing the local optimizer with `Muon` under a variety of federated optimization frameworks. Across all baselines—including plain local training (Local SGD), control-variate methods (SCAFFOLD), proximal or dynamic regularization methods (FedProx, FedDyn), aggregation-corrected methods (FedCM), and our momentum-aggregated variant—the incorporation of `Muon` consistently yields substantial accuracy improvements.

In particular, `Muon` provides 6.03% and 13.39% absolute gains over Local SGD on ResNet-18 and ViT-Tiny, respectively, demonstrating that `Muon` can significantly accelerate client-side adaptation even without any server-side correction. Similar improvements are observed when `Muon` is combined with stronger FL algorithms:

- `SCAFFOLD + Muon` gains 5.67% / 14.95%,
- `FedCM + Muon` gains 2.04% / 21.05%,
- `FedDyn + Muon` gains 4.14% / 15.90%,
- `FedProx + Muon` gains 7.59% / 16.27%

on ResNet-18 / ViT-Tiny, respectively.

These improvements are consistent and often large, indicating that the benefit of `Muon` is largely orthogonal to the benefit of the federated optimization algorithms themselves: regardless of whether the baseline relies on variance reduction, bias correction, or proximal regularization, `Muon` enables faster local convergence, mitigates drift accumulation, and enhances cross-round stability. The effect is most pronounced when `Muon` is combined with our momentum-aggregation strategy ($\bar{m} + \Delta_G$), which achieves the highest accuracy among all variants. Overall, Table 21 shows that `Muon` acts as a universal performance amplifier for federated learning, producing significant acceleration across diverse FL methodologies and model architectures.

Table 22: Effect of different local optimizers on CIFAR-100 (Dir-0.1, 300 rounds). Numbers in parentheses denote absolute improvement over the baseline in the previous row.

| Variant | ResNet-18 | ViT-Tiny |
|---|---|---|
| Local SGD | 60.25 | 27.14 |
| Local SGD + $\bar{m}$+$\Delta_G$ | 67.56 (↑ 7.31) | 33.23 (↑ 6.09) |
| Local AdamW | 58.97 | 37.86 |
| Local AdamW + $\bar{m}$+$\Delta_G$ | 66.25 (↑ 7.28) | 41.26 (↑ 3.40) |
| Local Shampoo | 59.62 | 37.56 |
| Local Shampoo + $\bar{m}$+$\Delta_G$ | 66.52 (↑ 6.90) | 42.25 (↑ 4.69) |
| Local Adafactor | 58.23 | 36.52 |
| Local Adafactor + $\bar{m}$+$\Delta_G$ | 65.52 (↑ 7.29) | 40.11 (↑ 3.59) |
| Local LAMB | 59.62 | 36.55 |
| Local LAMB + $\bar{m}$+$\Delta_G$ | 64.35 (↑ 4.73) | 38.65 (↑ 2.10) |
| FedLAMB | 62.35 | 36.28 |
| Local Muon | 66.28 | 32.56 |
| Local Muon + $\bar{m}$+$\Delta_G$ | **73.05** (↑ 6.77) | **48.18** (↑ 15.62) |

## C.8    IMPACT OF $\Delta_G$ AND $\bar{m}$ ON OTHER OPTIMIZERS

Table 22 evaluates the effect of applying our correction mechanism, $\bar{m}$+$\Delta_G$, on a variety of local optimizers. Across all optimizers—including AdamW, Shampoo, Adafactor, and LAMB—adding our global momentum correction consistently improves performance on both ResNet-18 and ViT-Tiny. These gains show that our correction effectively accelerates local training and alleviates client drift regardless of the underlying preconditioner.

Notably, the improvement is **substantially larger for Muon** than for any other optimizer. While AdamW, Shampoo, Adafactor, and LAMB obtain moderate gains (typically 3–7% on ResNet-18 and 2–6% on ViT-Tiny), the combination of Local Muon + $\bar{m}$+$\Delta_G$ yields the **largest boost**: **+6.77%** on ResNet-18 and a striking **+15.62%** on ViT-Tiny.

This pronounced improvement highlights a strong synergy between Muon and our correction mechanism. Muon's orthogonalized updates produce well-conditioned local steps, and our global calibration further aligns these steps with the global descent direction. Together, they enhance both optimization geometry and cross-client consistency, resulting in the fastest convergence and highest accuracy among all tested optimizers.

Overall, the results demonstrate that while our correction mechanism consistently accelerates all matrix-aware optimizers, **Muon benefits the most**, underscoring its unique suitability for federated learning with structured parameters.

As shown in Table 22, our framework consistently improves both convolutional and transformer backbones on CIFAR-100 under Dir-0.1 heterogeneity. Starting from standard Local SGD, incorporating our correction terms $\bar{m}$ and $\Delta_G$ yields gains of +7.31 and +6.09 absolute accuracy for ResNet-18 and ViT-Tiny, respectively. A similar trend holds for other first-order optimizers: for Local AdamW, the proposed framework improves accuracy by +7.28 (ResNet-18) and +3.40 (ViT-Tiny); for Local Adafactor, by +7.29 and +3.59.

Beyond first-order methods, our framework also accelerates a range of matrix-based adaptive optimizers. When applied to Local Shampoo, adding $\bar{m}$ and $\Delta_G$ leads to improvements of +6.90 and +4.69 for ResNet-18 and ViT-Tiny, respectively. For Local LAMB, we observe consistent boosts of +4.73 and +2.10, and the resulting models substantially outperform FedLAMB on ResNet-18 (64.35 vs. 62.35). The effect is most pronounced for Muon: Local Muon already performs strongly, but our framework further lifts performance by +6.77 on ResNet-18 and a remarkable +15.62 on ViT-Tiny, achieving the best overall accuracy among all variants. These results demonstrate that our framework is not limited to Muon itself; it provides a generic correction and acceleration mecha-

Table 23: Test accuracy, training loss on CIFAR-100 using **ResNet-18** and **ViT-Tiny** over 300 communication rounds under Dir-0.6 and Dir-0.1 settings (100 clients, 10% participation, batch size 50, $K = 50$).

| Method | ResNet-18 (Dir-0.6) | | ResNet-18 (Dir-0.1) | | ViT-Tiny (Dir-0.6) | | ViT-Tiny (Dir-0.1) | |
|---|---|---|---|---|---|---|---|---|
| | Test Acc | Train Loss | Test Acc | Train Loss | Test Acc | Train Loss | Test Acc | Train Loss |
| Local Muon | 71.66 | 0.395 | 66.71 | 1.504 | 46.69 | 0.201 | 40.53 | 1.432 |
| FedMuon(Algorithm 3) | 74.12 | 0.001 | 73.05 | 0.246 | 50.22 | 0.162 | 48.18 | 0.556 |
| FedMuon(Algorithm 2) | 73.16 | 0.005 | 72.85 | 0.254 | 49.85 | 0.178 | 48.02 | 0.562 |

nism that benefits both classical first-order optimizers and a broad family of matrix-based adaptive methods.

### C.9 COMPARE FEDMUON (ALGORITHM 2) AND FEDMUON (ALGORITHM 3)

Table 23 reports the test accuracy and final training loss on the CIFAR-100 dataset using two backbone architectures, **ResNet-18** and **ViT-Tiny**, under two non-IID data partitions generated by the Dirichlet distribution with concentration parameters $\alpha = 0.6$ and $\alpha = 0.1$. The experiments are conducted over 300 communication rounds with 100 clients, 10% client participation per round, a local batch size of 50, and $K = 50$ local optimization steps.

