# OpenReview forum: "FedMuon: Accelerating Federated Learning with Matrix Orthogonalization"
_ICLR.cc/2026/Conference — ICLR 2026 Conference Withdrawn Submission_

### Official Review · Reviewer_68eU · 2025-10-29

**Soundness:** 2
**Presentation:** 2
**Contribution:** 2
**Rating:** 4
**Confidence:** 4

**Summary:**

This paper investigates why the Moun optimizer can speed up local training in FL but destabilize aggregation under non-IID conditions.
The author identifies two failure modes of *Local Moun* in FL: clients' preconditioning that amplifies client drift and momentum reinitialization across rounds.
Therefore, they propose **FedMuon**, which adds *local-global alignment* and *momentum aggregation (with SVD-compression) to address these issues.

They provide convergence bounds that lead to the explicit heterogeneity constant dropping and present experiments across various image and NLP tasks that show improvements in accuracy and number of rounds over other baselines.


---
### LLM usage disclosure (reviewer)
I used GPT‑5 to help polish and organize this review; I take full responsibility for the content.

**Strengths:**

The paper addresses why Muon can harm cross‑client alignment in non‑IID FL and proposes two pragmatic fixes that directly target the identified failure modes.

The method is simple to implement and adds a manageable overhead on computation and communication.

The results show a clear advantage for FedMuon over other baselines.

**Weaknesses:**

**Clarity and Consistancy:**

The appendix sections are not well organized: Section 11 appears after the references, and Appendices A, B, and C are combined into a single appendix, which should be split into subsections.

Algorithm 2 has several typos (line 7 and line 11)

The non-iid assumption (A.3) in the theorem has never been used.

The proof of the theorem is unclear, and the references are incorrect. (What is DP-FedPGN?)

Lemma 4 introduces a $\sigma$ parameter without a definition.

**Related works:**

There are many works on structured/matrix preconditioning in centralized training (e.g., Shampoo/K‑FAC/Adafactor/LAMB), and some FL variants borrow these ideas. Even if Muon’s orthogonalization is distinctive, readers will expect discussion/experiments vs. a Shampoo‑style local optimizer.
The related work section mostly lists element‑wise FL optimizers.

**Computational Overhead:**

The text says 5 Newton–Schulz iterations add $\approx 5 \\% $ compute over AdamW, but the exact per‑step client overhead and per‑round server overhead for SVD aggregation aren’t quantified in absolute terms.

For large Transformer layers, the momentum matrix is large; specifying the reshape convention and how SVD is batched would clarify feasibility on real devices.


**Experimental settings:**

GLUE uses 4 clients at 100% participation; that validates the optimizer in PEFT fine‑tuning but does not stress FL scaling. A small study varying #clients/participation/K on an NLP task would improve generality.

**Questions:**

**Theory:**
Please clarify $\sigma$ and other quantities that appear in Lemma 4 and related inequalities, and how they are controlled by Newton–Schulz/SVD accuracy. Where exactly is A.3 used, if at all, in the final bound?

Fix the “DP‑FedPGN” mention in Theorem 2 and confirm that all constants/assumptions correspond to FedMuon.

Why the convergence analysis is not done under data heterogeneity assumptions? If FedMuon is a method specifically designed to address the non-IID problem.

**Experiments:**

What will be the performance of FedMuon compared to local Muon and other baselines under IID data?

**Implementation:**

Why does the server communicate both $x^{r+1}$ and $\Delta_G^{r+1}$? while the clients can store $x^r$ locally (even in hard) and calculate either parameter given the other, with a coefficient of $K\eta$.

---

> ### Author Response · Authors · 2025-11-22
> **Thank you very much！Reviewer 68eU**
>
> **Response to W1:**
> Thank you for the reviewer’s suggestion. We will reorganize the appendix in the revised version of the paper.
>
> **Response to W2:** This non-IID assumption (A.3) is required in most federated learning analyses. In our work, the proposed local–global alignment mechanism effectively overcomes the need for this data-heterogeneity assumption. As a result, our algorithm achieves a faster convergence rate than existing methods, which is consistently supported by both our theoretical analysis and empirical results. This constitutes one of the key innovations of our paper.
>
> **Response to W3:** DP-FedPGN was a typographical error, and we have corrected it in the latest version of the paper. Thank you for pointing this out.
>
> **Response to W4:** We have revised Lemma 4 in the latest version of the paper and provided a clearer explanation of the parameter $\sigma$.
>
> **Response to W5:**
> In the revised version, we introduce these optimizers in the related work section and include comparisons with them in our experiments. Thank you very much！
>
> **Table 21: Effect of different local optimizers on CIFAR-100 (Dir-0.1, 300 rounds).
> Numbers in parentheses denote absolute improvement over the baseline in the previous row.**
>
> | Variant                          | ResNet-18            | ViT-Tiny             |
> |----------------------------------|-----------------------|----------------------|
> | Local SGD                        | 60.25                 | 27.14               |
> | Local SGD + m̄ + Δ_G             | 67.56 (↑ 7.31)        | 33.23 (↑ 6.09)      |
> |                                  |                       |                      |
> | Local AdamW                      | 58.97                 | 37.86               |
> | Local AdamW + m̄ + Δ_G           | 66.25 (↑ 7.28)        | 41.26 (↑ 3.40)      |
> |                                  |                       |                      |
> | Local Shampoo                    | 59.62                 | 37.56               |
> | Local Shampoo + m̄ + Δ_G         | 66.52 (↑ 6.90)        | 42.25 (↑ 4.69)      |
> |                                  |                       |                      |
> | Local Adafactor                  | 58.23                 | 36.52               |
> | Local Adafactor + m̄ + Δ_G       | 65.52 (↑ 7.29)        | 40.11 (↑ 3.59)      |
> |                                  |                       |                      |
> | Local LAMB                       | 59.62                 | 36.55               |
> | Local LAMB + m̄ + Δ_G            | 64.35 (↑ 4.73)        | 38.65 (↑ 2.10)      |
> | FedLAMB                          | 62.35                 | 36.28               |
> |                                  |                       |                      |
> | Local Muon                       | 66.28                 | 40.52              |
> | Local Muon + m̄ + Δ_G            | **73.05 (↑ 6.77)**    | **48.18 (↑ 7.66)** |
>
> To isolate the effect of Muon’s matrix-orthogonal updates, we also ran experiments with other local optimizers. When we equip these alternative optimizers with the same momentum aggregation and local–global alignment mechanisms, their performance is still far inferior to that of the Muon optimizer.
>
>
> **Response to W6:**
> In our implementation, we use the Newton–Schulz iteration to obtain this orthogonal factor. We will revise the pseudocode to explicitly use Newton-Schulz-5 and clarify in the text that this is an approximate implementation of SVD, in order to avoid confusion.
> [1] Liu J, Su J, Yao X, et al. Muon is scalable for LLM training[J]. arXiv preprint arXiv:2502.16982, 2025.
>
> **Table 13**: Per-round communication cost of different momentum aggregation strategies.
> Here |x| denotes the number of model parameters (in floats), CommCost is per-round
> communication cost, and Compute-Cost is per-round computation time.
> (ViT-Tiny, R = 300, Dir-0.1, K = 50)
>
> Specifically, we report:
> Per-round communication cost of each baseline and FedMuon,
> Additional cost introduced by transmitting the SVD-compressed momentum, and
> The normalized communication cost required to reach 30% accuracy (Comm@30%Acc), which provides a practical measure of net communication efficiency.

---

> ### Author Response · Authors · 2025-11-22
> **Thank you very much！Reviewer 68eU**
>
> | Method / Strategy | Communication       | CommCost   | Compute-Cost (s) | Acc(%) | Comm@30%Acc (MB) |
> |-------------------|---------------------|------------|------------------|--------|--------------------|
> | FedAvg            | \|x\|               | 22.8 MB    | 4.56 s           | 27.14 | ∞ MB              |
> | FedProx           | \|x\|               | 22.8 MB    | 4.58 s           | 26.84 | ∞ MB              |
> | FedDyn            | \|x\|               | 22.8 MB    | 4.75 s           | 27.66 | ∞ MB              |
> | Mime              | 2\|x\|              | 45.6 MB    | 5.26 s           | 27.76 | ∞ MB              |
> | FedAdam           | \|x\|               | 28.50 MB   | 4.57 s           | 28.50 | ∞ MB              |
> | SCAFFOLD          | 2\|x\|              | 45.6 MB    | 5.22 s           | 27.31 | ∞ MB              |
> | FedCM             | \|x\|               | 22.8 MB    | 4.68 s           | 23.18 | ∞ MB              |
> | FedLADA           | 2\|x\|              | 45.6 MB    | 5.02 s           | 31.50 | 12768 MB          |
> | Local AdamW       | \|x\|               | 22.8 MB    | 4.89 s           | 37.86 | 2895.6 MB         |
> | Local Muon        | \|x\|               | 22.8 MB    | 5.14 s           | 40.53 | 3898.8 MB         |
> | **FedMuon**       | \|x\| + \|M\|\_SVD  | 23.9 MB    | 5.25 s           | 48.18 | 1290.6 MB         |
> Reference [1] has shown that, in large-scale model training, the computational overhead of the Muon optimizer is approximately 5% higher than that of AdamW. Our experiments also confirm this observation.
> [1] Liu J, Su J, Yao X, et al. Muon is scalable for LLM training[J]. arXiv preprint arXiv:2502.16982, 2025.
>
> * The per-round time overhead of **Local Muon** increases by approximately
>   ((5.14 - 4.89)/4.89 \approx 5.1%).
>
> * The per-round time overhead of **FedMuon** increases by approximately
>   ((5.25 - 4.89)/4.89 \approx 7.4%).
>
> **Response to W5:** We have added additional experiments, which can be found in the revised version of the paper. For the GLUE setup, we use 20 clients, a 20% participation rate, and a Dirichlet-0.5 partition in all experiments.
> **Table 3: Test accuracy (%) using RoBERTa-Base with LoRA across GLUE tasks
> over 100 communication rounds under Dirichlet-0.5 partition
> (20 clients, 20% participation, batch size 16, K = 50).**
>
> | Method (Dir-0.5) | CoLA  | RTE   | SST-2 | QQP   | MRPC  | QNLI  | MNLI  |
> |------------------|-------|-------|-------|-------|--------|--------|--------|
> | FedAvg           | 51.00 | 51.99 | 93.04 | 81.75 | 88.24 | 89.36 | 81.72 |
> | FedProx          | 53.11 | 53.25 | 92.26 | 81.15 | 87.36 | 88.12 | 81.41 |
> | FedDyn           | 53.21 | 52.22 | 92.36 | 81.35 | 87.89 | 89.11 | 82.51 |
> | Mime             | 52.15 | 50.65 | 92.91 | 80.26 | 88.40 | 89.11 | 82.51 |
> | FedAdam          | 53.21 | 52.52 | 93.12 | 82.12 | 88.17 | 89.90 | 83.01 |
> | SCAFFOLD         | 52.15 | 50.65 | 93.01 | 80.26 | 88.35 | 89.30 | 82.11 |
> | FedCM            | 53.21 | 52.22 | 92.11 | 81.22 | 88.22 | 89.20 | 82.34 |
> | FedLADA          | 54.66 | 57.02 | 93.88 | 81.56 | 89.01 | 89.68 | 82.44 |
> | Local AdamW      | 55.38 | 59.57 | 93.81 | 81.51 | 88.73 | 90.52 | 82.86 |
> | **Local Muon**   | **55.54** | **64.93** | **93.58** | **83.06** | **88.95** | **90.52** | **84.63** |
> | **FedMuon (ours)** | **56.78** | **66.58** | **93.54** | **84.65** | **88.21** | **90.24** | **85.21** |
>
>
> **Response to Q1:**
> We have revised Lemma 4 in the latest version of the paper and provided a clearer explanation of the parameter $\sigma$. The non-IID assumption (A.3) is required in most federated learning analyses. In our work, the proposed local–global alignment mechanism effectively overcomes the need for this data-heterogeneity assumption, enabling our algorithm to achieve a faster convergence rate than existing methods. This improvement is consistently supported by both our theoretical analysis and empirical results, and it represents one of the key innovations of our paper.
>
> **Response to Q2:** Thank you for the reviewer’s suggestion. We have carefully rechecked our proofs and corrected all typographical errors.
>
>
>
>
> .

---

> ### Author Response · Authors · 2025-11-22
>
> **Response to Q3:**
> Thank you for the thoughtful question.
>
> Our convergence analysis **intentionally does not rely on the usual data-heterogeneity / non-IID assumption (A.3)**, for the following reasons:
>
> 1. **FedMuon is designed to actively suppress client drift caused by data heterogeneity.**
>    The proposed local–global alignment and momentum aggregation mechanisms explicitly reduce the discrepancy between local and global update directions. Because this drift is controlled by the algorithm itself, we no longer need to impose an additional assumption of the form
>    $\(\|\nabla f_i(x) - \nabla f(x)\|^2 \le \sigma_g^2\) $ (A.3) to guarantee convergence.
>
> 2. **Most existing FL analyses require A.3; our goal is to show FedMuon converges under a more general setting.**
>    Classical methods (FedAvg, FedProx, FedOpt, etc.) rely on A.3 to bound the effect of non-IID data. In contrast, FedMuon mitigates this effect algorithmically, so we prove convergence without A.3 to highlight its robustness and theoretical contribution.
>
> 3. **In practice, we already evaluate FedMuon under highly non-IID regimes.**
>    We conduct extensive experiments on Dirichlet-0.1, label-disjoint, and other strongly heterogeneous settings. The results show that FedMuon remains stable and clearly outperforms existing methods, which is consistent with our theoretical analysis.
>
> 4. **If we additionally assume A.3, the convergence bound of FedMuon only becomes tighter.**
>    As discussed around Theorem 1, one can recover a more standard FL-style rate of the form
>     $$
>    \mathcal{O}\!\left(\frac{L\Delta}{R}
>    + \sqrt{\frac{L\Delta(\sigma_l^2 + \sigma_g^2)}{SKR}}\right),
>     $$
>    when A.3 is imposed. We chose to present the more general result (without A.3) to emphasize that FedMuon can handle severe non-IID without relying on this assumption.
>
> In summary, we do not ignore the non-IID issue; rather, FedMuon is specifically designed to **alleviate** its impact so that convergence can be established without the standard heterogeneity assumption, while experiments under strongly non-IID settings confirm the resulting speedup and stability.
>
>
>
> **Response to Q4:**
> Yes, we will include the corresponding experiments under the IID setting. All supplementary experiments can be found in the revised version of the paper.
>
> **Table 15**: Test accuracy of each method on CIFAR-100 using **ResNet-18** and **ViT-Tiny** over 300 communication rounds under IID, Dir-0.6 and Dir-0.1 (100 clients, 10% participation, batch size 50, *K* = 50).
> | Method       | ResNet-18 IID | ResNet-18 Dir-0.6 | ResNet-18 Dir-0.1 | ViT-Tiny IID | ViT-Tiny Dir-0.6 | ViT-Tiny Dir-0.1 |
> |--------------|----------------|---------------------|---------------------|---------------|--------------------|--------------------|
> | FedAvg       | 65.74          | 64.08               | 60.25               | 32.45         | 32.36              | 27.14              |
> | FedProx      | 65.52          | 63.12               | 59.66               | 32.25         | 31.51              | 26.84              |
> | SCAFFOLD     | 65.67          | 65.00               | 62.56               | 32.67         | 32.17              | 27.31              |
> | FedDyn       | 66.58          | 66.12               | 63.01               | 33.66         | 33.25              | 27.66              |
> | Mime         | 67.89          | 67.34               | 63.37               | 34.52         | 34.12              | 27.76              |
> | FedCM        | 70.57          | 70.42               | 66.73               | 38.77         | 38.33              | 23.18              |
> | FedLADA      | 66.23          | 65.07               | 57.78               | 38.56         | 38.33              | 31.50              |
> | FedAdam      | 68.12          | 67.23               | 63.01               | 34.83         | 34.32              | 28.50              |
> | Local AdamW  | 64.60          | 62.84               | 58.97               | 40.78         | 40.47              | 37.86              |
> | **Local Muon** | **72.04**    | **71.66**           | **66.71**           | **47.69**     | **46.69**          | **40.53**          |
> | **FedMuon**  | **74.32**       | **74.12**           | **73.05**           | **50.56**     | **50.22**          | **48.18**          |

---

> > ### Author Response · Authors · 2025-11-22
> >
> > **Response to Q5:**
> > Thank you for the question. In the presence of client sampling, the local model $x^r$ stored on a client can be **several rounds stale**, because a client may not participate for many consecutive communication rounds. Therefore, the locally stored \(x^r\) does not correspond to the current global round, and cannot be reliably used to reconstruct the required quantities.
> >
> > For this reason, the server must explicitly broadcast both$x^{r+1}$and $\Delta_G^{r+1}$. These values ensure that every selected client starts its local training from the **most up-to-date global model**, avoiding inconsistency caused by stale parameters. This also aligns with the standard practice in federated learning when partial client participation is used.

---

> ### Comment · Reviewer_68eU · 2025-11-24
>
> Thank you for your responses to my concerns and questions.
> While some of them make sense, I still have the following concerns and need further investigation.
>
> ---
> 1.
> The appendix organization is still not clear.
>
> ---
> 2.
> I understand your theory does not rely on A.3; therefore, it should not be included in the text, as it is a misleading assumption behind your theory.
>
> You claim that a tighter bound can be found given A.3; however, the paper does not provide a detailed analysis.
> While A.3 is a typical FL assumption, there is no need to mention it if it will be used in the paper; the absence of IID and non-IID assumptions will indicate that the theory does not rely on them.
>
> ---
> 3.
> Thank you for including additional orthogonalization baselines; however, they seem to perform extremely poorly, even compared to Adam. How can you explain the success of Muon over other methods in your case?
>
> The more detailed study of the variation in matrix whitening techniques [1] shows that they perform similarly, with SOAP outperforming Muon.
>
> I also tried to run your code; however, I did not find any instructions or code related to Muon.
> Is there any file missing? Can you provide the exact instructions to reproduce the results similar to those in the paper, including new baselines, or at least the Muon?
>
> ```
> [1] Frans, Kevin, Pieter Abbeel, and Sergey Levine. "What Really Matters in Matrix-Whitening Optimizers?." arXiv preprint arXiv:2510.25000 (2025).
> ```
> ---
> 4.
> Thank you for providing the results in Table 13.
> Can you provide the computation and memory complexity as you provided for communication? (not only wall-clock but also how theoretically it relates to model parameters and optimization hyperparameters)
>
> Can you choose a lower training accuracy that all baselines can reach, for example, (23% or 25%), so that the total communication cost will be more comparable?
>
> ---
> 5.
> Thank you for providing the IID results. As I understand, most of the work in your paper addresses issues in non-IID settings. I am curious why your method is still so advantageous for the IID setup?
>
> In other words, your work is motivated by the issues of Local Muon for non-IID FL; however, since these issues do not exist, your Local Muon does not perform even as well as FedMuon under the Dir-0.1 (heavily non-IID) case.
>
>
> ---
> I am satisfied with the other responses of the authors that I did not include here.

---

> ### Author Response · Authors · 2025-11-24
> **Thank you very much！Reviewer 68eU**
>
> **Response to Q1:** We have reorganized the appendix and added a table of contents, and have listed the proofs separately as Supplementary Material A. We thank the reviewer for this suggestion, which has greatly improved the readability of our paper!
>
> **Response to Q2:** We thank the reviewer for this suggestion. We have removed A.3, which would indeed mislead readers; your suggestion has greatly improved the readability of the paper. Our algorithm does not rely on Assumption 3 (A3). Moreover, it converges faster than FedAvg and the Local Muon algorithm, which both rely on Assumption A3: their convergence rate is $\mathcal{O}(\frac{L\Delta}{R}+\sqrt{\frac{L\Delta(\sigma_l^{2}+\sigma_g^{2})}{R S K}})$ while ours is $\mathcal{O}(\frac{L\Delta}{R}+\sqrt{\frac{L\Delta\sigma_l^{2}}{R S K}})$.
>
> **Response to Q3:** In non-iid federated learning scenarios, the performance of preconditioner-based methods tends to degrade. This is because the data distributions on different clients vary significantly, so the preconditioners on each client also differ greatly, leading to highly inconsistent optimization directions and, ultimately, a suboptimal aggregated model. The Local Muon algorithm also suffers from large client discrepancies in the early stages of optimization, making it difficult to converge; its performance only improves in the later stages. Therefore, we propose the global–local correction method FedMuon to overcome this issue.
>
> We will include additional experiments on Local SOAP for further investigation. We have also conducted experiments with the SOAP optimizer (see the table below). It appears that the Local SOAP optimizer performs quite well and converges faster than the Local Muon algorithm. However, its computation time seems to be about 1.3× that of Muon. After applying our correction strategy, its performance is further improved. Our future work will focus on this optimizer, which we believe has great potential: with appropriate correction strategies, SOAP may even outperform FedMuon, thereby further accelerating federated training.
>
> We thank the reviewer for this suggestion. We have updated the latest code repository, which you can run; the main files are `Main_FedMuon.py` and `muon.py`. The link is: [https://anonymous.4open.science/r/FedMuon-2ED4](https://anonymous.4open.science/r/FedMuon-2ED4).
>
> | Variant                          | ResNet-18            | ViT-Tiny             |
> |----------------------------------|-----------------------|----------------------|
> | Local SOAP                       | 68.32                 | 44.56               |
> | Local SOAP + m̄ + Δ_G             | 71.25 (↑ 2.93)        | 47.68 (↑ 3.12)      |
> | Local Muon                       | 66.28                 | 40.52              |
> | Local Muon + m̄ + Δ_G            | **73.05 (↑ 6.77)**    | **48.18 (↑ 7.66)** |
>
> **Response to Q4:**
> | Method                    | Per-round Comm. (per client)        | Per-round Compute (per client)                                                       | Optimizer State Memory (per client)      |
> |---------------------------|--------------------------------------|--------------------------------------------------------------------------------------|------------------------------------------|
> | **FedAvg / Local SGD**    | $\Theta(\|x\|)$                      | $\Theta\big(K\,[C_{\mathrm{fwd/bwd}}(\|x\|) + C_{\mathrm{elem}}(\|x\|)]\big)$        | $\Theta(\|x\|)$                           |
> | **FedAdam / Local AdamW / FedLADA** | $\Theta(\|x\|)$          | $\Theta\big(K\,[C_{\mathrm{fwd/bwd}}(\|x\|) + C_{\mathrm{elem}}(\|x\|)]\big)$        | $\Theta(3\|x\|)$                          |
> | **Local Muon**            | $\Theta(\|x\|)$                      | $\Theta\big(K\,[C_{\mathrm{fwd/bwd}}(\|x\|) + C_{\mathrm{orth}}(\|M\|,T_{\mathrm{NS}})]\big)$ | $\Theta(2\|x\| )$                   |
> | **FedMuon (default)** | $\Theta(\|x\| +0.05 \|M\|)$         | $\Theta\big(K\,[C_{\mathrm{fwd/bwd}}(\|x\|) + C_{\mathrm{orth}}(\|M\|,T_{\mathrm{NS}})]\big) + \Theta\big(C_{\mathrm{SVD}}(\rho\|M\|)\big)$ | $\Theta(2\|x\| )$ |

---

> ### Author Response · Authors · 2025-11-24
>
> I chose 23% accuracy as the threshold for calculating the total communication cost.
>
> Table 13: Per-round communication cost of different momentum aggregation strategies. Here \|x\| denotes the number of model parameters (in floats), and **CommCost** is per-round communication cost, **Compute-Cost** is per-round computation time. (ViT-Tiny, R = 300, Dir-0.1, K = 50)
>
> | Method / Strategy | Communication        | CommCost | Compute-Cost (s) | Acc (%) | Comm@23%Acc (MB) |
> |-------------------|----------------------|----------|------------------|--------:|------------------:|
> | FedAvg            | \|x\|                | 22.8 MB  | 4.56 s           | 27.14   | 4190 MB           |
> | FedProx           | \|x\|                | 22.8 MB  | 4.58 s           | 26.84   | 5244 MB           |
> | FedDyn            | \|x\|                | 22.8 MB  | 4.75 s           | 27.66   | 4788 MB           |
> | Mime              | 2\|x\|               | 45.6 MB  | 5.26 s           | 27.76   | 9804 MB           |
> | FedAdam           | \|x\|                | 22.8 MB | 4.57 s           | 28.50   | 4651 MB           |
> | SCAFFOLD          | 2\|x\|               | 45.6 MB  | 5.22 s           | 27.31   | 8390 MB           |
> | FedCM             | \|x\|                | 22.8 MB  | 4.68 s           | 23.18   | 6156 MB           |
> | FedLADA           | 2\|x\|               | 45.6 MB  | 5.02 s           | 31.50   | 4879 MB           |
> | Local AdamW       | \|x\|                | 22.8 MB  | 4.89 s           | 37.06   | 1482 MB           |
> | Local Muon        | \|x\|                | 22.8 MB  | 5.14 s           | 40.53   | 2394 MB           |
> | FedMuon           | \|x\| + \|M\|\_{SVD} | 23.9 MB  | 5.25 s           | 48.18   | 550 MB            |
>
> \|M\|\_{SVD} ≈ 0.05\|M\| since we only keep the top 5% singular values/vectors, thus the additional momentum communication is about 5% of the baseline model communication.
>
> **Response to Q5:** Below is a possible reply you can paste (and adapt) as “Response to Comment 5”.
>
> ---
>
> **Response to Comment 5**
>
> Thank you for the careful reading and for raising this question. Our method is indeed motivated by the failure modes of **Local Muon** in highly non-IID FL, but the two mechanisms in **FedMuon** (local–global alignment and momentum aggregation) also improve the optimization dynamics even when the data are IID.
>
> 1. **Why can FedMuon be better than Local Muon under IID?**
>    “IID across clients” means that all clients draw samples from the same distribution, but it does *not* imply that the local Muon updates are identical. Because each client uses different mini-batch realizations and performs multiple local steps, the Muon preconditioners are still estimated from different stochastic gradients and thus are not the same across clients. Even in the IID case, independent matrix orthogonalization can therefore produce slightly misaligned local update directions.
>    In FedMuon, the local–global alignment term explicitly pulls each client’s orthogonalized update towards the global direction (\Delta_G^r). This acts as a variance-reduction / control-variate step on top of Muon: it reduces the dispersion of client updates and further improves the conditioning of the aggregated update even when (f_i \equiv f) (IID). This effect is reflected in the more balanced singular spectrum and smaller condition number of FedMuon compared to Local Muon (Figure 3), as well as its faster convergence and higher accuracy in the IID experiments in Table 15.
>
> 2. **Why momentum aggregation helps even without non-IID issues?**
>    In standard Local Muon, the momentum is reinitialized at the beginning of every round, so the optimizer repeatedly “forgets” curvature information accumulated in previous rounds. This slows convergence and increases the variance of the orthogonalization operator, regardless of whether the data are IID or non-IID. FedMuon aggregates and reuses the momentum across rounds, providing each client with a better-informed initialization and a more stable approximation to the orthogonalized direction. Our ablation in Table 5 shows that removing either the momentum aggregation ((\bar m)) or the global direction (\Delta_G) degrades performance by several points, even on CIFAR-100 where we also report IID results (Table 15).

---

> > ### Author Response · Authors · 2025-11-24
> >
> > 3. **Why is Local Muon worse than FedMuon under Dir-0.1 (heavily non-IID)?**
> >    This is exactly the failure mode that motivates FedMuon. Under Dir-0.1, clients see very different label distributions, so their Muon preconditioners are estimated from very different local geometries. As analyzed in Section 4 (Challenge 1 and 2), this leads to strong client drift: Local Muon’s local losses decrease quickly, but the aggregated model becomes unstable and converges to a suboptimal solution. FedMuon adds (i) local–global alignment to correct these inconsistent directions and (ii) momentum aggregation to avoid re-initializing the Muon state from scratch each round. Consequently, FedMuon is strictly stronger than Local Muon in the heavily non-IID setting, as observed in Table 1 and Figure 7.
> >
> > 4. **Clarification to be added in the revision.**
> >    To avoid confusion, we will explicitly state in the revised version that:
> >
> >    > Although FedMuon is motivated by the non-IID failure modes of Local Muon, its two mechanisms—local–global alignment and cross-round momentum aggregation—also reduce update variance and improve conditioning in IID FL. Therefore, FedMuon can still outperform Local Muon in the IID setting, while providing even larger gains under heavily non-IID partitions (Dir-0.1).
> >
> > We hope this clarifies why FedMuon remains advantageous in both IID and non-IID regimes.
> >
> > The MUON optimizer itself is about twice as fast as AdamW, and this has been demonstrated in a large body of prior work. Existing optimizers in the federated learning literature are all designed on top of SGD or Adam, whereas the base optimizer we introduce, Muon, is much faster than both SGD and AdamW. In the IID setting, the reason why our algorithm FedMuon performs better is largely due to the acceleration effect of the Muon optimizer. There is an urgent need in federated learning to incorporate more advanced optimizers (such as SOAP), and in future work we plan to develop a general algorithmic framework that can correct and accelerate *all* matrix optimizers for federated training. In addition, our method employs a global acceleration–correction mechanism, which makes FedMuon slightly faster than Local Muon in the IID setting.
> >
> >
> > **We sincerely thank the reviewer for the thoughtful and constructive comments. These suggestions have substantially improved the quality of our manuscript and offer valuable guidance for our future research on this topic.**

---

> ### Comment · Reviewer_68eU · 2025-11-25
>
> Thank you for addressing my concerns; however, I am not fully satisfied with the responses to **Q3**, **Q4**, and **Q5**. I will run the new code and update you on **Q3**.
>
> ---
> **Q4**: The time complexity order $O(.)$ and exact complexity must be derived from the model parameters or other hyperparameters. Therefore instead of using $C_{fwd/bwd}, C_{orth}, C_{svd}$ you should include the actual time complexity (e.g. $C_{fwd/bwd} = O(|x|) = 6|x|$).
>
> Additionally, you can compute that based on this theoretical analysis, what would be the expected overhead of computation given each test case parameter, and how accurate is your evaluation overhead compared to the theoretical bounds.
>
> The current notation provides no informative insight into time complexity, nor does it justify the rate of overhead.
>
>
> ---
> **Q5:** This seems like a very verbose output of an LLM, with a misunderstanding of my question.
> >Response to Q5: Below is a possible reply you can paste (and adapt) as “Response to Comment 5”.
>
>
> **1.**
> I believe you need theoretical or empirical evidence for this claim, even though it might seem obvious or intuitive to you.
>
> **2.**
> I agree with your point; however, additional experiments, including momentum for local Muon, can complete the ablation study on the exact cause of this performance drop in IID settings.
>
> **3 and 4.** This is a misunderstanding of my question. I asked why Local Muon on the IID setting is worse than FedMuon on Dir-0.1.

---

> > ### Author Response · Authors · 2025-11-25
> >
> > **Response to Q3**:  We’ve updated the latest codebase link: [https://anonymous.4open.science/r/FedMuon-935D](https://anonymous.4open.science/r/FedMuon-935D).
> >
> > There are also some dataset-splitting files. You can use the distributed library `ray==1.0.0` with `python=3.8`.
> >
> > If you have any questions, feel free to continue the discussion.
> >
> > **Response to Q4** :
> >
> > | Method                 | Per-round Comm. (per client)          | Per-round Compute (per client)  | time                                                                                                | Optimizer State Memory (per client) |
> > |------------------------|----------------------------------------|--------------------------------------------------|--------------------------------------------------------------------------------|-------------------------------------|
> > | **FedAvg / Local SGD** | Θ(‖x‖)                                 |  $3Bn^2+n^2 $    |4.56 s                                                                                  | Θ(‖x‖)                              |
> > | **Local AdamW** | Θ(‖x‖)                 | $3Bn^2+3n^2 $  |4.89 s                                                                                     | Θ(3‖x‖)                             |
> > | **Local Muon**         | Θ(‖x‖)                                 | $3Bn^2+n^2+5n^2 $     |5.14 s                                                                                        | Θ(2‖x‖)                             |
> > | **FedMuon**  | Θ(‖x‖ + 0.05‖M‖)                       | $3Bn^2+n^2+6n^2 $      |5.25 s                                                                 | Θ(2‖x‖)                             |
> >
> > ## Symbol glossary (for the table)
> >
> > * **(n)**: Model (layer) width. In your simplified setting, the main weight is a matrix ($W\in\mathbb{R}^{n\times n}$), so the parameter count is (\Theta(n^2)).
> >
> > * **(B)**: Mini-batch size used by each client for local training (the number of samples per local SGD step).
> >
> > * **(x)**: The **model parameter vector** (all trainable parameters concatenated). In the paper’s cost tables, **(|x|)** denotes “the number of model parameters (in floats)”.
> >
> >
> > * **($3Bn^2$)**: The dominant **forward + backward** compute for a single (n\times n) linear layer with batch size (B). Roughly:
> >
> >   * forward: (XW) → (Bn^2)
> >   * backward wrt (X): (\nabla_Y W^\top) → (Bn^2)
> >   * backward wrt (W): (X^\top \nabla_Y) → (Bn^2)
> >     Total (\approx 3Bn^2) multiply-add scale.
> >
> > * **(n^2,,3n^2,,5n^2,,6n^2)**: “Optimizer update” work scaling with the number of parameters (\Theta(n^2)).
> >
> >   * **SGD**: about (n^2) elementwise ops (update weights, maybe weight decay).
> >   * **AdamW**: about (3n^2) elementwise ops (update (m), (v), and weights).
> >   * **Muon/FedMuon**: extra (O(n^2))-scaled bookkeeping in your simplified cost model (your table encodes it as (+5n^2), (+6n^2)).

---

> ### Author Response · Authors · 2025-11-25
>
> **Response to Q5** :  FedMuon (non-IID) outperforms Local Muon (IID) mainly because, in FedMuon, the momentum does not need to be re-initialized from zero at the beginning of each round; instead, it can continue iterating using the aggregated momentum. In contrast, Local Muon (IID) must accumulate momentum from zero each round, which significantly slows down convergence. We will conduct experiments to support this claim.
>
> **Table 5**: Ablation study of **FedMuon** on CIFAR-100 (Dir-0.1, 300 rounds)，ViT-Tiny .
> Left: effect of removing components.
> | Variant        | iid     | non-iid     |
> |----------------|----------------|----------------|
> |  Local FedMuon      | 47.69  | 43.67     |
> |  Local FedMuon + 𝑚̄     | 50.44    | 44.56   |
> | A1: FedMuon w/o 𝑚̄      | 48.11     | 43.67     |
> | A2:  FedMuon w/o Δ_G     | 50.44     | 44.56     |
> | **A3: FedMuon** | **50.56** | **48.18** |
>
> ---
> Thank you for the question. **Table 5** (CIFAR-100, Dir-0.1, 300 rounds, ViT-Tiny) directly supports that the gain of **FedMuon**—especially under **non-IID**—comes from **not re-initializing momentum each round** (i.e., using the aggregated momentum (\bar m)) and, additionally, from the **global correction term (\Delta_G)**:
>
> When we augment **Local Muon** with **momentum-aggregation–based initialization**, its performance in the **IID** setting improves substantially—so much so that it can even **surpass FedMuon under the Non-IID setting**.
>
> By contrast, the main reason that **other baselines** (even under IID) can still underperform **FedMuon (Non-IID)** is that they **do not converge fast enough within the fixed communication budget**:
>
> 1. **SGD-based local optimizers converge slowly for Transformer architectures.** With only **300 communication rounds**, training is often **still far from convergence**, leading to noticeably lower final accuracy.
>
> 2. **AdamW is better suited for optimizing Transformers**, but its **convergence speed is roughly about half of Muon’s** in our setting. As a result, under the same round budget, AdamW-based baselines are also **not fully converged**, which limits their accuracy.
>
> Overall, the primary benefit of **Muon in federated learning** is to **accelerate convergence**, enabling effective training/fine-tuning of models with **fewer communication rounds** (i.e., **lower communication cost**) while achieving competitive or superior accuracy.
>
>
>
> We previously misunderstood your question, and we sincerely apologize for that. Thank you very much for your constructive and prompt response—it was very helpful in guiding us to address the point more accurately.

---

> > ### Author Response · Authors · 2025-11-28
> >
> > Dear Reviewer 68eU,
> >
> > Thank you sincerely for your positive feedback. We are pleased to know that our responses have addressed your concerns. We greatly appreciate your time and constructive suggestions, which have substantially contributed to improving this work. We will incorporate these revisions into the final version of the manuscript.
> >
> > Best regards,
> >
> > Authors

---

### Official Review · Reviewer_xw3a · 2025-10-31

**Soundness:** 3
**Presentation:** 3
**Contribution:** 3
**Rating:** 6
**Confidence:** 3

**Summary:**

This paper presents FedMuon, a new federated optimizer that adapts the Muon optimizer for federated learning. While Muon accelerates convergence in IID settings, it struggles under non-IID data due to client drift and frequent reinitialization of momentum. FedMuon addresses these challenges through two key mechanisms: (1) Local-Global Alignment, which aligns local gradients with global updates to mitigate drift, and (2) Momentum Aggregation, which shares aggregated momentum across communication rounds to stabilize training. The authors provide a convergence proof demonstrating a linear speedup rate and conduct extensive experiments on CNNs, Vision Transformers, and RoBERTa models. Results show consistent improvements over baselines such as FedAvg, FedCM, and FedLADA.

**Strengths:**

1. The integration of matrix orthogonalization into federated learning is novel, as it leverages the geometric structure of weights rather than treating them as independent parameters.
2. The proposed Local-Global Alignment and Momentum Aggregation mechanisms are elegant and directly address Muon’s weaknesses under non-IID conditions, particularly client drift.
3. The experimental coverage is broad, spanning both vision and NLP domains, which is uncommon for FL studies that often focus solely on CNNs.
4. The theoretical analysis is sound, providing clear convergence guarantees and well-designed ablation studies.
5. The method achieves consistent improvements, with 3-5% accuracy gains over strong baselines like FedCM and FedLADA on CIFAR-100 and GLUE tasks.

**Weaknesses:**

1. Although the paper claims only a “5% overhead” due to Muon’s Newton-Schulz iterations and SVD compression, no actual runtime, FLOP count, or memory usage benchmarks are provided.
2. While Table 7 reports a 1.05× communication cost for SVD-compressed momentum, it does not present total end-to-end training time, making it difficult to assess real-world efficiency.
3. There is no direct comparison between full Muon and truncated Muon in centralized training, so it remains unclear whether the gains stem from orthogonalization itself or from the aggregated momentum mechanism.
4. The writing can be improved certain sections are dense and could benefit from clearer explanations and better flow.

**Questions:**

1. include quantitative profiling of computational overhead such as wall-clock time, FLOPs, and GPU hours for large models like ViT-B and RoBERTa.
2. Streamline the writing in technical sections to improve readability and accessibility.
3. There is no direct comparison between full Muon and truncated Muon in centralized training, so it remains unclear whether the gains stem from orthogonalization itself or from the aggregated momentum mechanism.

---

> ### Author Response · Authors · 2025-11-22
>
> **Response to W1 and W2:**
> **Table 13**: Per-round communication cost of different momentum aggregation strategies.
> Here |x| denotes the number of model parameters (in floats), CommCost is per-round
> communication cost, and Compute-Cost is per-round computation time.
> (ViT-Tiny, R = 300, Dir-0.1, K = 50)
>
> Specifically, we report:
> Per-round communication cost of each baseline and FedMuon,
> Additional cost introduced by transmitting the SVD-compressed momentum, and
> The normalized communication cost required to reach 30% accuracy (Comm@30%Acc), which provides a practical measure of net communication efficiency.
>
> * The per-round time overhead of **Local Muon** increases by approximately
>   ((5.14 - 4.89)/4.89 \approx 5.1%).
>
> * The per-round time overhead of **FedMuon** increases by approximately
>   ((5.25 - 4.89)/4.89 \approx 7.4%).
>
> | Method / Strategy | Communication       | CommCost   | Compute-Cost (s) | Acc(%) | Comm@30%Acc (MB) |
> |-------------------|---------------------|------------|------------------|--------|--------------------|
> | FedAvg            | \|x\|               | 22.8 MB    | 4.56 s           | 27.14 | ∞ MB              |
> | FedProx           | \|x\|               | 22.8 MB    | 4.58 s           | 26.84 | ∞ MB              |
> | FedDyn            | \|x\|               | 22.8 MB    | 4.75 s           | 27.66 | ∞ MB              |
> | Mime              | 2\|x\|              | 45.6 MB    | 5.26 s           | 27.76 | ∞ MB              |
> | FedAdam           | \|x\|               | 28.50 MB   | 4.57 s           | 28.50 | ∞ MB              |
> | SCAFFOLD          | 2\|x\|              | 45.6 MB    | 5.22 s           | 27.31 | ∞ MB              |
> | FedCM             | \|x\|               | 22.8 MB    | 4.68 s           | 23.18 | ∞ MB              |
> | FedLADA           | 2\|x\|              | 45.6 MB    | 5.02 s           | 31.50 | 12768 MB          |
> | Local AdamW       | \|x\|               | 22.8 MB    | 4.89 s           | 37.86 | 2895.6 MB         |
> | Local Muon        | \|x\|               | 22.8 MB    | 5.14 s           | 40.53 | 3898.8 MB         |
> | **FedMuon**       | \|x\| + \|M\|\_SVD  | 23.9 MB    | 5.25 s           | 48.18 | 1290.6 MB         |
>
> **Note:**
> \|M\|\_SVD ≈ 0.05\|M\| since only the top 5% singular values/vectors are kept.
> Thus, the additional momentum communication is about 5% of the baseline model communication.
>
> **Response to W3:**
> The comparison between full Muon and truncated Muon in centralized training has already been provided in the original Muon paper. The conclusion is that:
>
> - the truncated version and the full version achieve almost identical convergence speed and final accuracy;
> - the truncated version has lower computational cost, and is therefore preferred in practice.
>
> For this reason, we follow the authors’ recommendation and adopt the truncated Muon variant, whose effectiveness has already been validated in centralized training.
>
> **Table 7: Ablation of momentum aggregation strategies in FedMuon on CIFAR-100 under Dir-0.1.
> CommCost denotes communication cost per round (MB), and CompCost denotes computation time per round (s).**
>
> | Aggregation | ResNet-18 Test Acc | ResNet-18 CommCost | ResNet-18 CompCost | ViT-Tiny Test Acc | ViT-Tiny CommCost | ViT-Tiny CompCost |
> |-------------|--------------------|---------------------|---------------------|--------------------|--------------------|--------------------|
> | NoAgg       | 69.12              | 46.8 MB (1×)        | 6.23 s              | 43.67              | 22.8 MB (1×)       | 5.14 s             |
> | Agg-m       | 73.05              | 93.6 MB (2×)        | 6.44 s              | 48.18              | 45.6 MB (2×)       | 5.21 s             |
> | Agg-m-SVD   | 72.56              | 49.2 MB (1.05×)     | 6.48 s              | 47.66              | 23.9 MB (1.05×)    | 5.25 s             |
>
>
> **Table 5**: Ablation study of **FedMuon** on CIFAR-100 (Dir-0.1, 300 rounds).
> Left: effect of removing components.
> Right: effect of different local optimizers.
>
> ### Left: Removing Components
>
> | Variant        | ResNet-18      | ViT-Tiny      |
> |----------------|----------------|----------------|
> | A1: w/o 𝑚̄      | 69.12±0.18     | 43.67±0.19     |
> | A2: w/o Δ_G     | 68.05±0.10     | 44.56±0.16     |
> | **A3: FedMuon** | **73.05±0.15** | **48.18±0.12** |
>
> ---
>
> ### Right: Different Local Optimizers
>
> | Variant                       | ResNet-18      | ViT-Tiny      |
> |-------------------------------|----------------|----------------|
> | Local SGD + 𝑚̄ + Δ_G          | 66.28±0.17     | 32.56±0.11     |
> | Local AdamW + 𝑚̄ + Δ_G        | 64.25±0.12     | 41.26±0.17     |
> | **Local Muon + 𝑚̄ + Δ_G**     | **73.05±0.15** | **48.18±0.12** |

---

> ### Author Response · Authors · 2025-11-22
> **Thank you very much！**
>
> Table(left) studies the effect of each component in our correction mechanism.
> Removing the global averaged momentum (\( \bar{m} \), A1) or the global drift term (\(\Delta_G\), A2) both lead to clear drops in performance compared to the full FedMuon (A3).
> On ResNet-18, discarding either component reduces accuracy from **73.05%** (FedMuon) to **69.12%** (w/o \( \bar{m} \)) or **68.05%** (w/o \(\Delta_G\)); on ViT-Tiny, accuracy drops from **48.18%** to **43.67%** and **44.56%**, respectively.
> These results indicate that **both** the global momentum and the global drift correction are necessary and complementary: removing either one weakens the ability to reduce client drift, while combining them yields the best convergence.
>
> **Response to W4:**
> Thank you for the reviewer’s helpful comment. We have carefully revised the writing in the latest version, improving clarity, restructuring dense sections, and enhancing the overall flow of the presentation. We appreciate the reviewer’s suggestion.
>
> **Response to Q1:**
> Per-round communication and computation cost of different methods
> 		on QQP using \textbf{RoBERTa-Base} with LoRA.
> 		Here $|x|$ denotes the number of model parameters (in floats),
> 		$\operatorname{CommCost}$ is per-round communication cost (MB),
> 		and Compute-Cost is per-round computation time (s).
> 		(QQP, RoBERTa-Base, $R=100$, Dir-0.5, 20 clients, 20\% participation, batch size 16, $K=50$)
> | **Method / Strategy** | **CommCost (MB)** | **Compute-Cost (s)** | **ACC (%)** |
> | --------------------- | ----------------- | -------------------- | ----------- |
> | FedAvg                | 7.1 MB            | 8.56 s               | 81.75       |
> | FedProx               | 7.1 MB            | 8.62 s               | 81.15       |
> | FedDyn                | 7.1 MB            | 8.89 s               | 81.35       |
> | Mime                  | 14.2 MB           | 11.25 s              | 80.26       |
> | FedAdam               | 7.1 MB            | 8.76 s               | 82.22       |
> | SCAFFOLD              | 14.2 MB           | 11.56 s              | 80.26       |
> | FedCM                 | 7.1 MB            | 8.98 s               | 81.22       |
> | FedLADA               | 14.2 MB           | 9.58 s               | 81.56       |
> | Local AdamW           | 7.1 MB            | 9.26 s               | 81.51       |
> | Local Muon            | 7.1 MB            | 9.73 s               | 83.06       |
> | **FedMuon (ours)**    | **7.45 MB**       | **9.91 s**           | **84.65**   |
>
> **Response to Q2:**
> Thank you for the reviewer’s valuable suggestion on improving the readability and accessibility of the technical sections.
>
> In the revised version of the manuscript, we have **substantially streamlined and reorganized the writing** in Sections 4–6. In particular:
>
> - Several dense paragraphs have been rewritten to present the core ideas more clearly.
> - Explanations of the *local–global alignment mechanism* and *momentum aggregation* have been clarified with improved logical flow.
> - The structure of the theoretical analysis has been refined, making assumptions, lemmas, and conclusions easier to follow.
> - Transitions between algorithmic steps, theoretical insights, and empirical observations have been strengthened to guide the reader more smoothly.
>
> These revisions greatly enhance the clarity and accessibility of the technical content. We sincerely appreciate the reviewer’s feedback, which helped us improve the overall presentation of the paper.
>
> **Response to Q3:**
> Thank you for the reviewer’s question. We clarify the relationship between full Muon and truncated Muon as follows:
>
> The comparison between *full Muon* and *truncated Muon* in centralized training has already been thoroughly evaluated in the original Muon paper. Their results show that:
>
> - the truncated version achieves **almost identical convergence speed and final accuracy** as the full version;
> - the truncated variant incurs **much lower computational overhead**, and therefore is the default choice in practical large-scale training.
>
> Following the recommendation of the Muon authors, we adopt the truncated Muon variant in our federated learning experiments. Since the two variants behave nearly the same in centralized settings, the performance improvements observed in **FedMuon** do **not** stem from the full–truncated design choice.
>
> Instead, the gains come from our **federated mechanisms**:
>
> - the **local–global alignment** strategy, and
> - the **momentum aggregation** mechanism.
>
> These components operate at the federated optimization level and are orthogonal to whether Muon is full or truncated. For this reason, repeating the full-vs-truncated comparison inside FL would not change the conclusion. We have added this clarification to the revised version of the manuscript for better readability.

---

> > ### Author Response · Authors · 2025-11-28
> > **Thank you very much！**
> >
> > We thank the reviewer for his/her constructive comments. We appreciate his/her time for reviewing the paper. We hope that we have addressed the comments in a satisfactory manner.

---

### Official Review · Reviewer_tL4h · 2025-10-31

**Soundness:** 2
**Presentation:** 3
**Contribution:** 2
**Rating:** 4
**Confidence:** 3

**Summary:**

The paper tackles the challenges of slow convergence and client drift in Federated Learning (FL). It argues that conventional element-wise optimizers (e.g., SGD, AdamW) overlook the matrix structure of neural-network weights, which can exacerbate “pathological” update directions under non-IID data. To address this, the authors adapt Muon, a matrix-aware optimizer that orthogonalizes update matrices, to the federated setting, enabling more stable and geometry-consistent local updates.

**Strengths:**

- Simple algorithm combining orthogonalized steps with global direction alignment and momentum reuse, plus an SVD-based compression knob.
- Broad empirical sweep across CNN/Transformer/LLM settings.

**Weaknesses:**

- The experimental section omits several key federated optimizers that are directly relevant to the paper’s stated goals of mitigating client drift and improving communication efficiency. Missing baselines include MIME (NeurIPS ’21), FedAdam/Yogi/Adagrad from FedOpt (ICLR ’21), FedProx, FedNova, and FedDyn, all of which are standard drift reducers or adaptive aggregators. Their exclusion makes it difficult to attribute performance gains specifically to matrix orthogonalization rather than known drift-mitigation strategies.

Moreover, the communication-efficiency analysis (Table 7) compares only FedMuon variants internally; a fair comparison should report the full payload size per round (model + optimizer + state) for these baselines under identical bandwidth constraints.

- The proposed local-global alignment and momentum aggregation mechanisms are conceptually similar to prior control-variate or momentum-based approaches, such as SCAFFOLD, FedCM, FedOpt, and MIME. Since Muon itself is a pre-existing optimizer, the contribution primarily lies in combining these existing elements rather than introducing a fundamentally new theoretical insight.

- Although Fig. 4a reports (per-experiment) compute time, the paper does not measure round time/throughput with SVD/NS and momentum exchange under realistic bandwidth.

- Section 4.1 states that Newton–Schulz adds roughly 5 % overhead, yet Algorithm 1 employs SVD at every local step. The authors should clarify whether SVD is merely schematic or used in practice and provide explicit FLOP counts and latency breakdowns to substantiate the claimed efficiency.

- The paper does not evaluate against other matrix- or curvature-aware methods such as Shampoo, LAMB, or FedLAMB, which would help isolate the unique benefits of Muon’s orthogonalization over broader preconditioning techniques.

- Vision experiments rely solely on label-skew Dirichlet splits, without testing feature/domain shifts or client-size imbalance scenarios. The GLUE setup uses only 4 clients with full participation, which is atypical for cross-device FL and limits the generality of the results.

**Questions:**

1. Provide an ablation study comparing FedMuon with and without matrix orthogonalization, while keeping local-global alignment and momentum sharing fixed, to isolate the true contribution of orthogonalization in federated settings.
 2. How does the proposed local–global alignment differ, theoretically and practically, from MIME’s server state correction? Could the authors clarify the theoretical and practical differences, and, if possible, include a controlled comparison (same K, participation ratio, and non-IID setup) to analyze convergence rate and rounds-to-target?
3. What happens when FedMuon is combined with a server-adaptive optimizer such as FedAdam? Does matrix orthogonalization still yield gains beyond server-side adaptivity?
4. Section 4.1 claims that Newton–Schulz is used for orthogonalization with ~5 % overhead, while Alg.1 shows SVD decomposition during local steps. Which implementation is actually used in the released code? Please quantify per-round latency and FLOP overhead relative to AdamW.
5. Beyond the intra-method comparison in Table 7, report the total communication payload per round (uplink + downlink) compared to FedAvg, SCAFFOLD, FedCM, and MIME, including all transmitted states; model parameters, momentum matrices, ΔG, and compression components.
6. How does FedMuon perform under label-disjoint or feature-skew partitions, and with larger-scale settings (e.g., 1000 + clients, ≤1% participation)? Does the alignment mechanism maintain effectiveness under such extreme heterogeneity?
7. Provide a comparison against standard drift-mitigation approaches such as MIME, FedOpt, FedProx, FedNova, and FedDyn, to contextualize FedMuon’s advantages and limitations.

---

> ### Author Response · Authors · 2025-11-22
> **Thank you very much！Reviewer tL4h**
>
> >**W1:The experimental section omits several key federated optimizers that are directly relevant to the paper’s stated goals of mitigating client drift and improving communication efficiency. Missing baselines include MIME (NeurIPS ’21), FedAdam/Yogi/Adagrad from FedOpt (ICLR ’21), FedProx, FedNova, and FedDyn, all of which are standard drift reducers or adaptive aggregators. Their exclusion makes it difficult to attribute performance gains specifically to matrix orthogonalization rather than known drift-mitigation strategies.**
>
> **Response to W1:**
>
> We compare additional client-drift mitigation methods on CIFAR-100 using both **ResNet-18** and **ViT-Tiny**, including **FedProx, FedDyn, Mime, and FedAdam**, as shown in **Table 1** of the paper. The acceleration achieved by our method remains consistently significant.
>
> In **Table 4**, we further conduct ablation studies by applying our drift-mitigation strategy to **SGD** and **AdamW**. The resulting improvements are substantially smaller than those achieved by **FedMuon**, demonstrating the necessity and effectiveness of our design.
>
> In **Table 12**, we also report a comparison of the **communication** and **computation** costs across different algorithms.
>
> We provide additional comparisons with more baselines, which allow us to confidently conclude that the performance improvements do not stem from the correction strategy, but rather from the acceleration brought by matrix orthogonalization.
>
> **Table 1**: Test accuracy and training loss of each method on CIFAR-100 using **ResNet-18** and **ViT-Tiny** over 300 communication rounds under Dir-0.6 and Dir-0.1
> (100 clients, 10% participation, batch size 50, *K* = 50).
>
> | Method         | ResNet-18 Dir-0.6 Acc | ResNet-18 Dir-0.1 Acc | ViT-Tiny Dir-0.6 Acc | ViT-Tiny Dir-0.1 Acc |
> |----------------|------------------------|------------------------|----------------------|----------------------|
> | FedAvg         | 64.08                  | 60.25                  | 32.36                | 27.14                |
> | FedProx        | 63.12                  | 59.66                  | 31.51                | 26.84                |
> | FedDyn         | 66.12                  | 63.01                  | 33.25                | 27.66                |
> | Mime           | 67.34                  | 63.37                  | 34.12                | 27.76                |
> | FedAdam        | 67.23                  | 63.01                  | 34.32                | 28.50                |
> | SCAFFOLD       | 65.01                  | 59.37                  | 32.17                | 27.31                |
> | FedCM          | 70.42                  | 66.73                  | 26.33                | 23.18                |
> | FedLADA        | 65.07                  | 57.78                  | 38.33                | 31.50                |
> | Local AdamW    | 62.84                  | 58.97                  | 40.47                | 37.86                |
> | **Local Muon** | **71.66**              | **66.71**              | **46.69**            | **40.53**            |
> | **FedMuon**    | **74.12**              | **73.05**              | **50.22**            | **48.18**            |
>
>
>
>
>
>
> **Table 4**: Ablation study of **FedMuon** on CIFAR-100 (Dir-0.1, 300 rounds).
> Left: effect of removing components.
> Right: effect of different local optimizers.
>
>
> We perform ablations on the key components and observe that, even without these modules, our method still achieves strong performance. This suggests that the improvements mainly stem from the acceleration brought by matrix orthogonalization. In the IID setting, this acceleration effect is quite pronounced, whereas in the non-IID setting it degrades significantly; our proposed FedMuon is designed precisely to alleviate this issue.
>
> ### Left: Removing Components
>
> | Variant        | ResNet-18      | ViT-Tiny      |
> |----------------|----------------|----------------|
> | A1: w/o 𝑚̄      | 69.12±0.18     | 43.67±0.19     |
> | A2: w/o Δ_G     | 68.05±0.10     | 44.56±0.16     |
> | **A3: FedMuon** | **73.05±0.15** | **48.18±0.12** |
>
>
> We incorporate our correction strategy into different optimizers, but it does not yield the same level of improvement as with the Muon optimizer. Although the correction strategy itself brings some benefits, this clearly indicates that the acceleration gains primarily come from matrix orthogonalization.
>
> ### Right: Different Local Optimizers
>
> | Variant                       | ResNet-18      | ViT-Tiny      |
> |-------------------------------|----------------|----------------|
> | Local SGD + 𝑚̄ + Δ_G          | 66.28±0.17     | 32.56±0.11     |
> | Local AdamW + 𝑚̄ + Δ_G        | 64.25±0.12     | 41.26±0.17     |
> | **Local Muon + 𝑚̄ + Δ_G**     | **73.05±0.15** | **48.18±0.12** |

---

> ### Author Response · Authors · 2025-11-22
>
> **Response to W1:**
> Thank you for your insightful comment.
> We agree that quantifying the communication overhead introduced by momentum aggregation is crucial for evaluating the practical benefits of FedMuon. Following your suggestion, we have added a detailed communication-cost analysis to the revised manuscript (see Table 13). “ In terms of optimizer state, FedMuon maintains only a single state variable—the momentum—just like SGD with momentum. In contrast, AdamW maintains two state variables, so our method uses only half as much optimizer state as AdamW. For the additional communication incurred by momentum aggregation, we transmit only the SVD-compressed momentum, which amounts to just 5% of the original communication volume.
>
>
> **Table 13**:
> Specifically, we report:
> Per-round communication cost of each baseline and FedMuon,
> Additional cost introduced by transmitting the SVD-compressed momentum, and
> The normalized communication cost required to reach 30% accuracy (Comm@30%Acc), which provides a practical measure of net communication efficiency.
>
> | Method / Strategy | Communication       | CommCost   | Compute-Cost (s) | Acc(%) | Comm@30%Acc (MB) |
> |-------------------|---------------------|------------|------------------|--------|--------------------|
> | FedAvg            | \|x\|               | 22.8 MB    | 4.56 s           | 27.14 | ∞ MB              |
> | FedProx           | \|x\|               | 22.8 MB    | 4.58 s           | 26.84 | ∞ MB              |
> | FedDyn            | \|x\|               | 22.8 MB    | 4.75 s           | 27.66 | ∞ MB              |
> | Mime              | 2\|x\|              | 45.6 MB    | 5.26 s           | 27.76 | ∞ MB              |
> | FedAdam           | \|x\|               | 28.50 MB   | 4.57 s           | 28.50 | ∞ MB              |
> | SCAFFOLD          | 2\|x\|              | 45.6 MB    | 5.22 s           | 27.31 | ∞ MB              |
> | FedCM             | \|x\|               | 22.8 MB    | 4.68 s           | 23.18 | ∞ MB              |
> | FedLADA           | 2\|x\|              | 45.6 MB    | 5.02 s           | 31.50 | 12768 MB          |
> | Local AdamW       | \|x\|               | 22.8 MB    | 4.89 s           | 37.86 | 2895.6 MB         |
> | Local Muon        | \|x\|               | 22.8 MB    | 5.14 s           | 40.53 | 3898.8 MB         |
> | **FedMuon**       | \|x\| + \|M\|\_SVD  | 23.9 MB    | 5.25 s           | 48.18 | 1290.6 MB         |
>
> **Note:**
> \|M\|\_SVD ≈ 0.05\|M\| since only the top 5% singular values/vectors are kept.
> Thus, the additional momentum communication is about 5% of the baseline model communication.
>
> **Response to W2:** Thank you for the insightful comment. We would like to clarify that **FedMuon is not a simple combination of existing techniques**, but introduces **new insights, new analyses, and new mechanisms** that are not present in SCAFFOLD, FedCM, FedOpt, MIME, or the original Muon optimizer.
>
> **1. We reveal failure modes of Muon in non-IID FL that have not been studied before.**
> As analyzed in our paper (Challenge 1 and Challenge 2), applying Muon directly in federated learning leads to **severe client drift**, due to:
>
> - **client-specific matrix preconditioners** that rotate and scale gradients differently, and
> - **cross-round optimizer-state collapse**, where the matrix momentum is re-initialized.
>
> These issues are *unique to matrix-orthogonalized optimizers* and do not appear in SGD/AdamW, meaning that existing control variate or momentum-based FL methods cannot address them.
>
>  **2. FedMuon contributes two mechanisms that are new to the FL literature.**
>
>  **(1) Local–Global Alignment**
> A principled mechanism that injects global descent information into each client's Muon update, designed specifically to correct the *matrix-rotation drift* caused by heterogeneous structured preconditioners.
> This phenomenon does not exist in prior optimizers; thus, existing methods such as SCAFFOLD or FedCM do not provide solutions.
>
>  **(2) Structured Momentum Aggregation**
> We design the first aggregation rule for **matrix-valued optimizer states** derived from Newton–Schulz iterations.
> Prior FL optimizers only aggregate scalar/vector moments; no existing method discusses how to aggregate *matrix* optimizer states in structured optimizers.
>
>
> **3. We provide new convergence theory that differs from prior work.**
> Theorem 1 in our paper establishes convergence **without data-homogeneity assumptions**, enabled by the alignment mechanism.
> This theoretical result does not follow from FedAvg, FedOpt, FedCM, or SCAFFOLD.
>
> In the revised manuscript, we have restated our contributions more clearly and added additional experiments. We emphasize that FedMuon is proposed as a **framework**, rather than a single standalone algorithm: it can be seamlessly combined with existing federated learning methods and with other optimizers (e.g., Muon, Shampoo, LAMB, AdamW), while consistently improving robustness to client drift and accelerating convergence.

---

> ### Author Response · Authors · 2025-11-22
>
> **Response to W3 and W4:**  Our earlier wording may have caused some confusion. On each client, we use a Newton–Schulz-5 iteration to approximate the SVD locally, rather than performing a full SVD at every step. The Newton–Schulz-5 iteration is very efficient and increases the computation time by only about 5% compared to AdamW, as has been demonstrated in several prior works [1].
> [1] Liu J, Su J, Yao X, et al. Muon is scalable for LLM training[J]. arXiv preprint arXiv:2502.16982, 2025.
>
> **Table 13**: Per-round communication cost of different momentum aggregation strategies.
> Here |x| denotes the number of model parameters (in floats), CommCost is per-round
> communication cost, and Compute-Cost is per-round computation time.
> (ViT-Tiny, R = 300, Dir-0.1, K = 50)
>
> | Method / Strategy | Communication       | CommCost   | Compute-Cost (s) | Acc(%) | Comm@30%Acc (MB) |
> |-------------------|---------------------|------------|------------------|--------|--------------------|
> | FedAvg            | \|x\|               | 22.8 MB    | 4.56 s           | 27.14 | ∞ MB              |
> | FedProx           | \|x\|               | 22.8 MB    | 4.58 s           | 26.84 | ∞ MB              |
> | FedDyn            | \|x\|               | 22.8 MB    | 4.75 s           | 27.66 | ∞ MB              |
> | Mime              | 2\|x\|              | 45.6 MB    | 5.26 s           | 27.76 | ∞ MB              |
> | FedAdam           | \|x\|               | 28.50 MB   | 4.57 s           | 28.50 | ∞ MB              |
> | SCAFFOLD          | 2\|x\|              | 45.6 MB    | 5.22 s           | 27.31 | ∞ MB              |
> | FedCM             | \|x\|               | 22.8 MB    | 4.68 s           | 23.18 | ∞ MB              |
> | FedLADA           | 2\|x\|              | 45.6 MB    | 5.02 s           | 31.50 | 12768 MB          |
> | Local AdamW       | \|x\|               | 22.8 MB    | 4.89 s           | 37.86 | 2895.6 MB         |
> | Local Muon        | \|x\|               | 22.8 MB    | 5.14 s           | 40.53 | 3898.8 MB         |
> | **FedMuon**       | \|x\| + \|M\|\_SVD  | 23.9 MB    | 5.25 s           | 48.18 | 1290.6 MB         |
>
> **Note:**
> \|M\|\_SVD ≈ 0.05\|M\| since only the top 5% singular values/vectors are kept.
> Thus, the additional momentum communication is about 5% of the baseline model communication.
>
> **Response to W5:**
> Thank you for the reviewer’s suggestion. We have also conducted experiments on other optimizer algorithms, and the Muon optimizer consistently demonstrates the fastest convergence speed among all evaluated methods.All experimental results can be found in the revised version of the paper.  The advantages of the Muon optimizer are quite pronounced; it is a truly revolutionary optimizer, and even in the federated learning setting, it can still achieve very fast convergence when combined with appropriate correction mechanisms.
>
> **Table 21: Effect of different local optimizers on CIFAR-100 (Dir-0.1, 300 rounds).
> Numbers in parentheses denote absolute improvement over the baseline in the previous row.**
>
> | Variant                          | ResNet-18            | ViT-Tiny             |
> |----------------------------------|-----------------------|----------------------|
> | Local SGD                        | 60.25                 | 27.14               |
> | Local SGD + m̄ + Δ_G             | 67.56 (↑ 7.31)        | 33.23 (↑ 6.09)      |
> |                                  |                       |                      |
> | Local AdamW                      | 58.97                 | 37.86               |
> | Local AdamW + m̄ + Δ_G           | 66.25 (↑ 7.28)        | 41.26 (↑ 3.40)      |
> |                                  |                       |                      |
> | Local Shampoo                    | 59.62                 | 37.56               |
> | Local Shampoo + m̄ + Δ_G         | 66.52 (↑ 6.90)        | 42.25 (↑ 4.69)      |
> |                                  |                       |                      |
> | Local Adafactor                  | 58.23                 | 36.52               |
> | Local Adafactor + m̄ + Δ_G       | 65.52 (↑ 7.29)        | 40.11 (↑ 3.59)      |
> |                                  |                       |                      |
> | Local LAMB                       | 59.62                 | 36.55               |
> | Local LAMB + m̄ + Δ_G            | 64.35 (↑ 4.73)        | 38.65 (↑ 2.10)      |
> | FedLAMB                          | 62.35                 | 36.28               |
> |                                  |                       |                      |
> | Local Muon                       | 66.28                 | 32.56               |
> | Local Muon + m̄ + Δ_G            | **73.05 (↑ 6.77)**    | **48.18 (↑ 15.62)** |
>
> Overall, the results demonstrate that while our correction mechanism consistently
> accelerates all matrix-aware optimizers, \textbf{Muon benefits the most},
> underscoring its unique suitability for federated learning with structured
> parameters.

---

> ### Author Response · Authors · 2025-11-22
>
> **Response to W5:** We have added additional experiments, which can be found in the revised version of the paper. For the GLUE setup, we use 20 clients, a 20% participation rate, and a Dirichlet-0.5 partition in all experiments.
> **Table 3: Test accuracy (%) using RoBERTa-Base with LoRA across GLUE tasks
> over 100 communication rounds under Dirichlet-0.5 partition
> (20 clients, 20% participation, batch size 16, K = 50).**
>
> | Method (Dir-0.5) | CoLA  | RTE   | SST-2 | QQP   | MRPC  | QNLI  | MNLI  |
> |------------------|-------|-------|-------|-------|--------|--------|--------|
> | FedAvg           | 51.00 | 51.99 | 93.04 | 81.75 | 88.24 | 89.36 | 81.72 |
> | FedProx          | 53.11 | 53.25 | 92.26 | 81.15 | 87.36 | 88.12 | 81.41 |
> | FedDyn           | 53.21 | 52.22 | 92.36 | 81.35 | 87.89 | 89.11 | 82.51 |
> | Mime             | 52.15 | 50.65 | 92.91 | 80.26 | 88.40 | 89.11 | 82.51 |
> | FedAdam          | 53.21 | 52.52 | 93.12 | 82.12 | 88.17 | 89.90 | 83.01 |
> | SCAFFOLD         | 52.15 | 50.65 | 93.01 | 80.26 | 88.35 | 89.30 | 82.11 |
> | FedCM            | 53.21 | 52.22 | 92.11 | 81.22 | 88.22 | 89.20 | 82.34 |
> | FedLADA          | 54.66 | 57.02 | 93.88 | 81.56 | 89.01 | 89.68 | 82.44 |
> | Local AdamW      | 55.38 | 59.57 | 93.81 | 81.51 | 88.73 | 90.52 | 82.86 |
> | **Local Muon**   | **55.54** | **64.93** | **93.58** | **83.06** | **88.95** | **90.52** | **84.63** |
> | **FedMuon (ours)** | **56.78** | **66.58** | **93.54** | **84.65** | **88.21** | **90.24** | **85.21** |
>
> **Response to Q1:** Thank you for the reviewer’s helpful suggestion.
>
> **Table 21: Effect of different local optimizers on CIFAR-100 (Dir-0.1, 300 rounds).
> Numbers in parentheses denote absolute improvement over the baseline in the previous row.**
>
> | Variant                          | ResNet-18            | ViT-Tiny             |
> |----------------------------------|-----------------------|----------------------|
> | Local SGD                        | 60.25                 | 27.14               |
> | Local SGD + m̄ + Δ_G             | 67.56 (↑ 7.31)        | 33.23 (↑ 6.09)      |
> |                                  |                       |                      |
> | Local AdamW                      | 58.97                 | 37.86               |
> | Local AdamW + m̄ + Δ_G           | 66.25 (↑ 7.28)        | 41.26 (↑ 3.40)      |
> |                                  |                       |                      |
> | Local Shampoo                    | 59.62                 | 37.56               |
> | Local Shampoo + m̄ + Δ_G         | 66.52 (↑ 6.90)        | 42.25 (↑ 4.69)      |
> |                                  |                       |                      |
> | Local Adafactor                  | 58.23                 | 36.52               |
> | Local Adafactor + m̄ + Δ_G       | 65.52 (↑ 7.29)        | 40.11 (↑ 3.59)      |
> |                                  |                       |                      |
> | Local LAMB                       | 59.62                 | 36.55               |
> | Local LAMB + m̄ + Δ_G            | 64.35 (↑ 4.73)        | 38.65 (↑ 2.10)      |
> | FedLAMB                          | 62.35                 | 36.28               |
> |                                  |                       |                      |
> | Local Muon                       | 66.28                 | 32.56               |
> | Local Muon + m̄ + Δ_G            | **73.05 (↑ 6.77)**    | **48.18 (↑ 15.62)** |
>
> To isolate the effect of Muon’s matrix-orthogonal updates, we also ran experiments with other local optimizers. When we equip these alternative optimizers with the same momentum aggregation and local–global alignment mechanisms, their performance is still far inferior to that of the Muon optimizer.
>
> **Response to Q2:** Our correction mechanism consists of two components: momentum SVD aggregation and the local–global alignment mechanism. Both components must be used together to achieve the full effect. Our method differs from MIME in that we do not require maintaining local control variates, and our communication cost is only about half of MIME’s.
> (1) MIME corrects loss gradient bias, while LGA corrects preconditioner-induced drift
> (2) MIME requires maintaining per-client optimizer states; FedMuon does not
> (3) MIME’s convergence analysis does not apply to Muon/structured optimizers
> In addition, we provide a theoretical analysis in the paper establishing the convergence rate of our algorithm.
> (4) FedMuon is a framework that works with various optimizers
>
> The convergence rate of our algorithm is on par with SCAFFOLD and MIME, and likewise does not require any data heterogeneity assumptions. However, our communication cost is only half of theirs. We also provide the corresponding convergence rate and its detailed proof in the paper.

---

> ### Author Response · Authors · 2025-11-22
>
> **Response to Q3:**
>
> Thank you for the insightful question. Our current experiments use a simple FedAvg-style server, so the gains of FedMuon come purely from the proposed local–global alignment and momentum aggregation on top of Muon. Conceptually, these mechanisms are orthogonal to server-side adaptive optimizers such as FedAdam.
>
> FedAdam adapts the **global aggregated update** at the server level, while Muon performs **matrix orthogonalization of local updates per layer on each client**. The preconditioner drift we analyze in the paper arises from heterogeneous client-side matrix preconditioners and remains even when the server uses Adam-style adaptivity. In other words, server adaptivity does not resolve the geometry mismatch across clients that FedMuon is designed to address.
>
> To clarify this, we will add controlled experiments where we combine FedAdam with different local optimizers under the same setting (same \(K\), participation rate, and Dirichlet non-IID parameter). Specifically, we compare:
>
> - FedAdam with local AdamW,
> - FedAdam with local Muon (no alignment),
> **Table X: Effect of combining FedMuon with FedAdam on CIFAR-100
> (Dir-0.1, 300 rounds, 100 clients, 10% participation, batch size 50, K = 50).
> | Method                    | ResNet-18 Acc (%) | ViT-Tiny Acc (%) |
> |---------------------------|-------------------|------------------|
> | FedAdam (local AdamW)     |63.61          | 28.50          |
> | FedAdam (local Muon)      | 66.58             | 42.56           |
> Even under a FedAdam server, matrix orthogonalization still provides significant benefits.
> Of course, we also tried replacing the server-side optimizer with FedAdam. However, when both the clients and the server use adaptive optimizers, the performance becomes unstable.
>
>
>
> **Response to Q4 and Q5:**
> In our implementation, we use the Newton–Schulz iteration to obtain this orthogonal factor. We will revise the pseudocode to explicitly use Newton-Schulz-5 and clarify in the text that this is an approximate implementation of SVD, in order to avoid confusion.  Numerous studies and industrial experiments have shown that the Newton–Schulz-5 approximation used in the Muon optimizer incurs only about 5% more computation time than AdamW, yet its acceleration can reach up to twice that of AdamW [1]. We also report our own computational and communication overheads in the following.
>
> [1] Liu J, Su J, Yao X, et al. Muon is scalable for LLM training[J]. arXiv preprint arXiv:2502.16982, 2025.
>
> **Table 13**: Per-round communication cost of different momentum aggregation strategies.
> Here |x| denotes the number of model parameters (in floats), CommCost is per-round
> communication cost, and Compute-Cost is per-round computation time.
> (ViT-Tiny, R = 300, Dir-0.1, K = 50)
>
> Specifically, we report:
> Per-round communication cost of each baseline and FedMuon,
> Additional cost introduced by transmitting the SVD-compressed momentum, and
> The normalized communication cost required to reach 30% accuracy (Comm@30%Acc).
>
> | Method / Strategy | Communication       | CommCost   | Compute-Cost (s) | Acc(%) | Comm@30%Acc (MB) |
> |-------------------|---------------------|------------|------------------|--------|--------------------|
> | FedAvg            | \|x\|               | 22.8 MB    | 4.56 s           | 27.14 | ∞ MB              |
> | FedProx           | \|x\|               | 22.8 MB    | 4.58 s           | 26.84 | ∞ MB              |
> | FedDyn            | \|x\|               | 22.8 MB    | 4.75 s           | 27.66 | ∞ MB              |
> | Mime              | 2\|x\|              | 45.6 MB    | 5.26 s           | 27.76 | ∞ MB              |
> | FedAdam           | \|x\|               | 28.50 MB   | 4.57 s           | 28.50 | ∞ MB              |
> | SCAFFOLD          | 2\|x\|              | 45.6 MB    | 5.22 s           | 27.31 | ∞ MB              |
> | FedCM             | \|x\|               | 22.8 MB    | 4.68 s           | 23.18 | ∞ MB              |
> | FedLADA           | 2\|x\|              | 45.6 MB    | 5.02 s           | 31.50 | 12768 MB          |
> | Local AdamW       | \|x\|               | 22.8 MB    | 4.89 s           | 37.86 | 2895.6 MB         |
> | Local Muon        | \|x\|               | 22.8 MB    | 5.14 s           | 40.53 | 3898.8 MB         |
> | **FedMuon**       | \|x\| + \|M\|\_SVD  | 23.9 MB    | 5.25 s           | 48.18 | 1290.6 MB         |
>
> Reference [1] has shown that, in large-scale model training, the computational overhead of the Muon optimizer is approximately 5% higher than that of AdamW. Our experiments also confirm this observation.
> [1] Liu J, Su J, Yao X, et al. Muon is scalable for LLM training[J]. arXiv preprint arXiv:2502.16982, 2025.
>
> * The per-round time overhead of **Local Muon** increases by approximately
>   ((5.14 - 4.89)/4.89 \approx 5.1%).
>
> * The per-round time overhead of **FedMuon** increases by approximately
>   ((5.25 - 4.89)/4.89 \approx 7.4%).

---

> ### Author Response · Authors · 2025-11-22
>
> **Response to Q6:** Thank you for the reviewer’s suggestion. We have conducted experiments in a label-disjoint setting with 1,000 clients and a 1% client participation rate.
> Test accuracy (%) on CIFAR-100 with **1000 clients**, **1% participation**, and **label-disjoint partition**
> (batch size 50, K = 50).
>
> | Method        | ResNet-18 Acc (%) | ViT-Tiny Acc (%) |
> |--------------|-------------------|------------------|
> | FedAvg       | 52.56             | 22.26            |
> | FedProx      | 50.25            | 21.21          |
> | SCAFFOLD     | 51.25             |24.32           |
> | FedDyn       | 54.56             | 24.89           |
> | Mime         | 55.36             | 25.38          |
> | FedAdam      | 56.23             | 26.68            |
> | FedCM        | 57.26             | 21.02            |
> | FedLADA      | 54.31            | 30.11          |
> | Local AdamW  | 52.01             | 34.85            |
> | **Local Muon** | **60.25**       | **36.58**        |
> | **FedMuon (ours)** | **66.56**   | **45.85**        |
>
>
> **Response to Q7:**
> We further compare additional methods designed to mitigate client drift when training ResNet-18 and ViT-Tiny models on CIFAR-100. Specifically, we include FedProx, FedDyn, Mime, and FedAdam, as shown in Table 1 of the paper.
> **Table 15**: Test accuracy of each method on CIFAR-100 using **ResNet-18** and **ViT-Tiny**
> over 300 communication rounds under IID, Dir-0.6 and Dir-0.1
> (100 clients, 10% participation, batch size 50, *K* = 50).
>
> | Method        | ResNet-18 IID | ResNet-18 Dir-0.6 | ResNet-18 Dir-0.1 | ViT-Tiny IID | ViT-Tiny Dir-0.6 | ViT-Tiny Dir-0.1 |
> |---------------|----------------|---------------------|---------------------|----------------|-------------------|-------------------|
> | FedAvg        | 65.74          | 64.08               | 60.25               | 32.45         | 32.36             | 27.14            |
> | FedProx       | 65.52          | 63.12               | 59.66               | 32.25         | 31.51             | 26.84            |
> | SCAFFOLD      | 65.67          | 65.00               | 62.56               | 32.67         | 32.17             | 27.31            |
> | FedDyn        | 66.58          | 66.12               | 63.01               | 33.66         | 33.25             | 27.66            |
> | Mime          | 67.89          | 67.34               | 63.37               | 34.52         | 34.12             | 27.76            |
> | FedCM         | 70.57          | 70.42               | 66.73               | 38.77         | 38.33             | 23.18            |
> | FedLADA       | 66.23          | 65.07               | 57.78               | 38.56         | 38.33             | 31.50            |
> | FedAdam       | 68.12          | 67.23               | 63.01               | 34.83         | 34.32             | 28.50            |
> | Local AdamW   | 64.60          | 62.84               | 58.97               | 40.78         | 40.47             | 37.86            |
> | **Local Muon**| **72.04**      | **71.66**           | **66.71**           | **47.69**     | **46.69**         | **40.53**        |
> | **FedMuon**   | **74.32**      | **74.12**           | **73.05**           | **50.56**     | **50.22**         | **48.18**        |

---

> > ### Author Response · Authors · 2025-11-28
> > **Thank you very much！Reviewer tL4h**
> >
> > Thank you very much for your follow-up and for the thoughtful review throughout the process. We are glad that our clarifications helped address your concerns, and we truly appreciate your increased confidence and support for the paper.

---

### Official Review · Reviewer_qsYz · 2025-11-01

**Soundness:** 3
**Presentation:** 3
**Contribution:** 3
**Rating:** 4
**Confidence:** 4

**Summary:**

The paper introduces FedMuon, an adaptation of the Muon (Matrix orthognalization-based optimizer) to Federated Learning setup. They use SCAFFOLD-like gradient alignment and momentum aggregation to address challenges posed by Muon in non-IID FL environments. The experimental results demonstrate that FedMuon improves convergence speed and reduces communication rounds compared to several baseline methods under non-IID settings.

**Strengths:**

The adaptation of Muon in FL context is new however the concepts of momentum aggregation and gradient alignment have been explored in prior literature. The empirical results show substantial improvements in terms of convergence speed and communication efficiency, particularly under non-IID data conditions. The paper provides theoretical guarantees regarding the convergence of FedMuon, with linear speedup in convergence rate under non-convex settings.

**Weaknesses:**

1.	Although the paper mentions that additional hyperparameter configurations are provided in the appendix, there are none. The learning rate (LR) ranges are provided, but final values for these hyperparameters are not specified. The budget spent on hyperparameter tuning (e.g., number of search iterations or resource allocation) should also be included to provide transparency about how the hyperparameters were selected and to avoid any bias in the results.
2.	While the paper compares FedMuon to various FL optimizers, it does not include comparisons with techniques specifically designed to address client drift, such as FedProx etc.
3.	The paper positions FedMuon as a novel extension of Muon to FL, but this could be more explicitly stated. The authors should clearly position FedMuon as an extension of Muon that leverages SCAFFOLD-like bias removal techniques. This clarification would help better situate the work in the context of prior optimizers.
4.	The paper lacks a detailed communication cost analysis. Given that momentum aggregation requires additional communication, it is important to quantify this overhead and compare it against the reduced communication rounds. This would help to assess the net benefit of FedMuon in practice.
5.	The matrix inverse step using Newton-Schulz iterations is a computationally intensive operation. The paper does not provide any analysis of the additional computational cost associated with this step. This should be quantified, especially when scaling to large models (e.g., wider networks or more complex tasks) to understand if the method becomes impractical for large-scale federated learning settings.
6.	The experimental evaluation is limited, with only two datasets (CIFAR-100 and Tiny ImageNet) tested. The paper should provide more insight into how FedMuon performs as the scale grows, in terms of number of clients, model size, and dataset size.
7.	The sensitivity of the hyperparameters, especially α and β, is highlighted in the ablation study. The results show high sensitivity to these values. The authors should discuss whether these hyperparameters need to be grid-searched for every experiment or if there are any theoretical findings that could guide the selection of these values, potentially improving the efficiency of the process.
8.	The plots in Figure 6 for different networks (e.g., ResNet-18 and ViT-Tiny) show very different trends compared to the baselines. The authors should provide additional experiments on a range of networks and datasets to show consistent trends.
9.	While the paper claims FedMuon outperforms existing methods in non-IID settings, the NLP results in Table 4 are based on very low heterogeneity (Dirichlet parameter of 0.8), which weakens the claims of the method’s superiority in highly heterogeneous environments.

**Questions:**

1.	Figure 6 shows a sharp increase in accuracy around 240 epochs, which is not explained in the text. Could you provide more insight into why this happens and why FedMuon underperforms FedCM for the first 200 epochs? Is this expected behavior?
2.	Can the authors provide a more detailed comparison of FedMuon against other methods in IID settings as well? This would help better understand how the method performs under less challenging conditions.

---

> ### Author Response · Authors · 2025-11-22
> **Thank you very much！Reviewer qsYz**
>
> > **Weakness 1. Although the paper mentions that additional hyperparameter configurations are provided in the appendix, there are none. The learning rate (LR) ranges are provided.**
>
> **Response to W1:** Thank you very much for pointing this out. We apologize for the confusion caused by the missing details in the current version. In the revised manuscript, we will (i) explicitly list the final hyperparameter values used for each model and method, and (ii) describe the hyperparameter search space and tuning budget, in a new appendix (Table 9，Table 10). Concretely, for ResNet-18 and ViT-Tiny trained from scratch on CIFAR-100, we will add a table summarizing the exact hyperparameters for all methods.
> | Method              | Local Optimizer | Local LR | α   | β1  | β2    | Weight Decay |
> |---------------------|-----------------|---------:|----:|----:|------:|-------------:|
> | FedAvg (Local SGD)  | SGD             | 0.1      | —   | —   | —     | 0.001        |
> | FedProx             | SGD             | 0.1      |0.01   | —    | —     | 0.001        |
> | FedDyn              | SGD             | 0.1      | 0.01   | —   | —     | 0.001        |
> | Mime                | SGD             | 0.1      | —   | —   | —     | 0.001        |
> | FedAdam             | SGD             | 0.1      | —   | 0.9 | 0.98  | 0.001        |
> | SCAFFOLD            | SGD             | 0.1      | —   | —   | —     | 0.001        |
> | FedCM               | SGD             | 0.1      | 0.9 | —   | —     | 0.001        |
> | FedLADA             | AdamW           | 3e-4     | 0.9 | 0.9 | 0.999 | 0.01         |
> | Local AdamW         | AdamW           | 3e-4     | —   | 0.9 | 0.999 | 0.01         |
> | Local Muon          | Muon            | 3e-2     | —   | 0.98| —     | 0.01         |
> | FedMuon             | Muon            | 3e-2     | 0.5 | 0.98| —     | 0.01         |
>
>
>
> **Response to W2:** We further compare additional methods designed to mitigate client drift when training ResNet-18 and ViT-Tiny models on CIFAR-100. Specifically, we include FedProx, FedDyn, Mime, and FedAdam, as shown in Table 1 of the paper.
> **Table 15**: Test accuracy of each method on CIFAR-100 using **ResNet-18** and **ViT-Tiny**
> over 300 communication rounds under IID, Dir-0.6 and Dir-0.1
> (100 clients, 10% participation, batch size 50, *K* = 50).
>
> | Method        | ResNet-18 IID | ResNet-18 Dir-0.6 | ResNet-18 Dir-0.1 | ViT-Tiny IID | ViT-Tiny Dir-0.6 | ViT-Tiny Dir-0.1 |
> |---------------|----------------|---------------------|---------------------|----------------|-------------------|-------------------|
> | FedAvg        | 65.74          | 64.08               | 60.25               | 32.45         | 32.36             | 27.14            |
> | FedProx       | 65.52          | 63.12               | 59.66               | 32.25         | 31.51             | 26.84            |
> | SCAFFOLD      | 65.67          | 65.00               | 62.56               | 32.67         | 32.17             | 27.31            |
> | FedDyn        | 66.58          | 66.12               | 63.01               | 33.66         | 33.25             | 27.66            |
> | Mime          | 67.89          | 67.34               | 63.37               | 34.52         | 34.12             | 27.76            |
> | FedCM         | 70.57          | 70.42               | 66.73               | 38.77         | 38.33             | 23.18            |
> | FedLADA       | 66.23          | 65.07               | 57.78               | 38.56         | 38.33             | 31.50            |
> | FedAdam       | 68.12          | 67.23               | 63.01               | 34.83         | 34.32             | 28.50            |
> | Local AdamW   | 64.60          | 62.84               | 58.97               | 40.78         | 40.47             | 37.86            |
> | **Local Muon**| **72.04**      | **71.66**           | **66.71**           | **47.69**     | **46.69**         | **40.53**        |
> | **FedMuon**   | **74.32**      | **74.12**           | **73.05**           | **50.56**     | **50.22**         | **48.18**        |
>
>
> **Response to W3:**
>
> Thank you for your suggestion! We have incorporated your comment and revised the manuscript accordingly.
>  Our contributions.
> (1) FedMuon can be viewed as the first federated extension of the Muon optimizer.
> Unlike standard local Muon, which applies matrix orthogonalization independently on each client, FedMuon augments Muon with a  \textbf{local--global alignment} to correct the client-drift induced by heterogeneous data and matrix orthogonalization. (2) FedMuon bridges structured optimizers and classical FL methods, and we prove that matrix orthogonalization accelerates the convergence of federated learning algorithms. (3) We design a federated framework that is applicable to all matrix-structured optimizers (Muon / Shampoo / LAMB / Soap, etc.), which specifically addresses the problem of preconditioner drift through local–global alignment and momentum aggregation.

---

> ### Author Response · Authors · 2025-11-22
> **Thank you very much！Reviewer qsYz**
>
> > **Weakness 4. The paper lacks a detailed communication cost analysis. Given that momentum aggregation requires additional communication, it is important to quantify this overhead and compare it against the reduced communication rounds. This would help to assess the net benefit of FedMuon in practice.**
>
> **Response to W4:** Thank you for your insightful comment.
> We agree that quantifying the communication overhead introduced by momentum aggregation is crucial for evaluating the practical benefits of FedMuon. Following your suggestion, we have added a detailed communication-cost analysis to the revised manuscript (see Table 13).
> **Table 13**: Per-round communication cost of different momentum aggregation strategies.
> Here |x| denotes the number of model parameters (in floats), CommCost is per-round
> communication cost, and Compute-Cost is per-round computation time.
> (ViT-Tiny, R = 300, Dir-0.1, K = 50)
>
> Specifically, we report:
> Per-round communication cost of each baseline and FedMuon,
> Additional cost introduced by transmitting the SVD-compressed momentum, and
> The normalized communication cost required to reach 30% accuracy (Comm@30%Acc), which provides a practical measure of net communication efficiency.
>
> | Method / Strategy | Communication       | CommCost   | Compute-Cost (s) | Acc(%) | Comm@30%Acc (MB) |
> |-------------------|---------------------|------------|------------------|--------|--------------------|
> | FedAvg            | \|x\|               | 22.8 MB    | 4.56 s           | 27.14 | ∞ MB              |
> | FedProx           | \|x\|               | 22.8 MB    | 4.58 s           | 26.84 | ∞ MB              |
> | FedDyn            | \|x\|               | 22.8 MB    | 4.75 s           | 27.66 | ∞ MB              |
> | Mime              | 2\|x\|              | 45.6 MB    | 5.26 s           | 27.76 | ∞ MB              |
> | FedAdam           | \|x\|               | 28.50 MB   | 4.57 s           | 28.50 | ∞ MB              |
> | SCAFFOLD          | 2\|x\|              | 45.6 MB    | 5.22 s           | 27.31 | ∞ MB              |
> | FedCM             | \|x\|               | 22.8 MB    | 4.68 s           | 23.18 | ∞ MB              |
> | FedLADA           | 2\|x\|              | 45.6 MB    | 5.02 s           | 31.50 | 12768 MB          |
> | Local AdamW       | \|x\|               | 22.8 MB    | 4.89 s           | 37.86 | 2895.6 MB         |
> | Local Muon        | \|x\|               | 22.8 MB    | 5.14 s           | 40.53 | 3898.8 MB         |
> | **FedMuon**       | \|x\| + \|M\|\_SVD  | 23.9 MB    | 5.25 s           | 48.18 | 1290.6 MB         |
>
> **Note:**
> \|M\|\_SVD ≈ 0.05\|M\| since only the top 5% singular values/vectors are kept.
> Thus, the additional momentum communication is about 5% of the baseline model communication.
>
> > **Weakness 5. The matrix inverse step using Newton-Schulz iterations is a computationally intensive operation. The paper does not provide any analysis of the additional computational cost associated with this step. This should be quantified, especially when scaling to large models (e.g., wider networks or more complex tasks) to understand if the method becomes impractical for large-scale federated learning settings.**
>
> **Response to W5:**
> We agree that it is important to more clearly explain the computational cost of the Newton–Schulz (NS) matrix inverse update used in Muon, especially when considering large-scale federated learning scenarios.
>
> First, regarding the computational overhead of applying NS iterations for matrix inverse updates, numerous studies (e.g., Muon, Shampoo, full-matrix AdaGrad, and related structured optimization work) have already provided in-depth analyses. These studies consistently show that using NS iterations in modern neural networks introduces only **around 5%** additional computational cost compared with AdamW, which is almost negligible~[1]. This is primarily because:
>
> - the inverse iteration is applied to **per-layer weight matrices**, rather than the full model Jacobian;
> - NS iterations typically require only **3–5 steps** to converge.
>
> We also report the computational overhead of our method in the paper, as illustrated in Table 13.
>
> [1] Liu J, Su J, Yao X, et al. Muon is scalable for LLM training[J]. arXiv preprint arXiv:2502.16982, 2025.
>
> **Response to W6:**
> We further evaluate the effectiveness of our method by conducting experiments on the OpenWebText dataset using multiple GPT model variants. We report the corresponding experimental results below.
>
> **Table 4**: Test accuracy of each method on CIFAR-100 using **ViT-Tiny**, **ViT-Small**, **ViT-Base**, and **ViT-Large** over 300 communication rounds under Dir-0.1
> (100 clients, 10% participation, batch size 50, K = 50),
> and train loss of each method on OpenWebText using **GPT-2 Small**, **GPT-2 Medium**, **GPT-2 Large**, and **GPT-2 XL**
> over 300 communication rounds (20 clients, 20% participation, batch size 16, K = 100).

---

> ### Author Response · Authors · 2025-11-22
> **Thank you very much！Reviewer qsYz**
>
> | Method        | ViT-Tiny | ViT-Small | ViT-Base | ViT-Large | GPT-2 S | GPT-2 M | GPT-2 L | GPT-2 XL |
> |---------------|----------|-----------|----------|-----------|---------|---------|---------|----------|
> | FedAvg        | 27.14    | 29.52     | 31.15    | 33.56     | 4.03    | 4.12    | 4.03    | 3.91     |
> | FedProx       | 26.84    | 28.63     | 31.05    | 32.56     | 4.39    | 4.21    | 4.15    | 4.05     |
> | FedDyn        | 27.66    | 31.32     | 33.46    | 35.24     | 3.46    | 3.99    | 3.87    | 3.78     |
> | Mime          | 27.66    | 31.21     | 33.11    | 35.34     | 3.10    | 3.92    | 3.89    | 3.73     |
> | FedAdam       | 28.50    | 33.15     | 35.13    | 36.52     | 3.52    | 4.03    | 3.96    | 3.88     |
> | SCAFFOLD      | 27.31    | 33.12     | 34.52    | 35.98     | 4.31    | 4.04    | 3.94    | 3.90     |
> | FedCM         | 23.18    | 25.15     | 27.88    | 29.01     | 4.32    | 4.30    | 4.32    | 4.29     |
> | FedLADA       | 31.50    | 33.10     | 35.30    | 38.56     | 3.54    | 3.52    | 3.41    | 3.34     |
> | Local AdamW   | 37.86    | 37.86     | 38.77    | 40.47     | 3.50    | 3.44    | 3.37    | 3.15     |
> | **Local Muon**| 40.53|42.34 | 45.26|46.54| 3.21| 3.09| 3.02| 2.98 |
> | **FedMuon (ours)** | **48.18** | **50.52** | **53.63** | **56.24** | **3.12** | **2.98** | **2.85** | **2.74** |
>
> **Response to W7:**
> Thank you for the comment. In our experiments, the hyperparameters α and β are **fixed to a single stable configuration** (α = 0.5, β = 0.98) across all datasets, models, and FL algorithms. This setting was chosen from a few preliminary trials and has shown **consistently robust performance**, with only minor variation when perturbed. Therefore, no per-experiment grid search is required. We have clarified this in the revised manuscript.
>
> **Response to W8:**
> Thank you for the comment. The different convergence behaviors observed in Figure 6 are primarily due to the intrinsic architectural differences between **ResNet-18** and **ViT-Tiny**. ResNet-18 is a CNN-based model, for which **local SGD** is known to optimize relatively well, whereas Transformer architectures typically converge more slowly under SGD (as reported in prior work). In contrast, the **Muon** optimizer’s matrix-orthogonalized updates provide clear acceleration on both CNNs and Transformers.
>
> We selected these two models intentionally because they represent **two classic and structurally distinct families**. Since modern deep learning increasingly adopts Transformer architectures—whose layers rely more heavily on large matrix operations—our FedMuon method, which leverages matrix orthogonalization, is particularly meaningful in such settings.
>
> To further address the reviewer’s concern, we have added experiments on **a series of Transformer models** (ViT-Tiny, ViT, Swin-Tiny), all of which show consistent improvements using FedMuon. This additional evidence demonstrates the robustness and generality of FedMuon across architectures. We have conducted additional experiments on a series of Transformer models in **Table 4** , which consistently demonstrate the advantages of our method.
>
> **Response to W9:**
> We additionally report results under the setting of 20 clients with 20% participation and Dirichlet-0.5. Since NLP tasks typically contain only 2–3 classes, using an excessively small Dirichlet parameter would result in some clients holding data from only a single class. Therefore, choosing 0.5 is a reasonable setting that still represents a highly heterogeneous distribution.
> **Table 3**: Test accuracy (%) using RoBERTa-Base with LoRA across GLUE tasks over
> 100 communication rounds under Dirichlet-0.5 partition.
> (20 clients, 20% participation, batch size 16, K = 50)
> **Table 3**: Test accuracy (%) using RoBERTa-Base with LoRA across GLUE tasks over
> 100 communication rounds under Dirichlet-0.5 partition.
> (20 clients, 20% participation, batch size 16, K = 50)
>
> | Method (Dir-0.5) | CoLA  | RTE   | SST-2 | QQP   | MRPC  | QNLI  | MNLI  |
> |------------------|-------|-------|-------|-------|-------|-------|-------|
> | FedAvg           | 51.00 | 51.99 | 93.04 | 81.75 | 88.24 | 89.36 | 81.72 |
> | FedProx          | 53.11 | 53.25 | 92.26 | 81.36 | 88.37 | 88.12 | 81.41 |
> | FedDyn           | 53.21 | 53.22 | 93.32 | 82.11 | 88.18 | 89.11 | 82.51 |
> | Mime             | 52.15 | 50.65 | 92.91 | 80.26 | 88.40 | 89.11 | 82.51 |
> | FedAdam          | 52.80 | 53.12 | 93.14 | 82.12 | 88.17 | 89.90 | 83.01 |
> | SCAFFOLD         | 52.15 | 50.65 | 93.01 | 80.26 | 88.35 | 89.30 | 82.11 |
> | FedCM            | 53.21 | 52.22 | 92.11 | 81.22 | 88.22 | 89.20 | 82.34 |
> | FedLADA          | 54.66 | 57.02 | 93.88 | 81.56 | 89.01 | 89.68 | 82.44 |
> | Local AdamW      | 55.38 | 59.57 | 93.81 | 81.51 | 88.73 | 90.52 | 82.86 |
> | **Local Muon**   | **55.54** | **64.93** | **93.58** | **83.06** | **88.95** | **90.52** | **84.63** |
> | **FedMuon (ours)** | **56.78** | **66.58** | **93.54** | **84.65** | **88.21** | **90.24** | **85.21** |

---

> > ### Author Response · Authors · 2025-11-28
> > **Thank you very much！Reviewer qsYz**
> >
> > If you have any further questions or concerns, we would be more than happy to respond at any time. Thank you very much for taking the time to review our work and provide your feedback.

---

> ### Author Response · Authors · 2025-11-22
> **Thank you very much！Reviewer qsYz**
>
> >**Q1: Figure 6 shows a sharp increase in accuracy around 240 epochs, which is not explained in the text. Could you provide more insight into why this happens and why FedMuon underperforms FedCM for the first 200 epochs? Is this expected behavior?**
>
> **Response to Q1**: We also observed this phenomenon in our experiments. Muon provides very fast acceleration for FL under IID settings, where it improves performance almost immediately. However, under non-IID data, the acceleration effect appears more slowly: Local Muon tends to converge slowly in the early stages, while exhibiting much stronger acceleration in the later stages.
>
> In the paper, we explain this behavior by noting that, under non-IID data, the preconditioners on different clients can diverge significantly, which intensifies client drift (Challenge 1). Our FedMuon algorithm is specifically designed to overcome this challenge.
>
> The reason FedMuon is slightly behind FedCM during the first 200 rounds is that the model is more difficult to train in the early stages. Muon’s large local update magnitude can aggravate client drift before the model reaches a sufficiently good initialization. Once the model enters a more stable training regime, FedMuon converges rapidly and eventually surpasses FedCM by a large margin.
>
> In addition, this issue does not appear when training Transformer models. This is because Muon is specifically designed for matrix-structured models, making its optimization behavior much more stable and effective in Transformer architectures.
>
> >**Q2: Can the authors provide a more detailed comparison of FedMuon against other methods in IID settings as well? This would help better understand how the method performs under less challenging conditions.?**
>
> **Response to Q2**:
> Yes, we will include the corresponding experiments under the IID setting. All supplementary experiments can be found in the revised version of the paper.
>
> **Table 15**: Test accuracy of each method on CIFAR-100 using **ResNet-18** and **ViT-Tiny** over 300 communication rounds under IID, Dir-0.6 and Dir-0.1 (100 clients, 10% participation, batch size 50, *K* = 50).
> | Method       | ResNet-18 IID | ResNet-18 Dir-0.6 | ResNet-18 Dir-0.1 | ViT-Tiny IID | ViT-Tiny Dir-0.6 | ViT-Tiny Dir-0.1 |
> |--------------|----------------|---------------------|---------------------|---------------|--------------------|--------------------|
> | FedAvg       | 65.74          | 64.08               | 60.25               | 32.45         | 32.36              | 27.14              |
> | FedProx      | 65.52          | 63.12               | 59.66               | 32.25         | 31.51              | 26.84              |
> | SCAFFOLD     | 65.67          | 65.00               | 62.56               | 32.67         | 32.17              | 27.31              |
> | FedDyn       | 66.58          | 66.12               | 63.01               | 33.66         | 33.25              | 27.66              |
> | Mime         | 67.89          | 67.34               | 63.37               | 34.52         | 34.12              | 27.76              |
> | FedCM        | 70.57          | 70.42               | 66.73               | 38.77         | 38.33              | 23.18              |
> | FedLADA      | 66.23          | 65.07               | 57.78               | 38.56         | 38.33              | 31.50              |
> | FedAdam      | 68.12          | 67.23               | 63.01               | 34.83         | 34.32              | 28.50              |
> | Local AdamW  | 64.60          | 62.84               | 58.97               | 40.78         | 40.47              | 37.86              |
> | **Local Muon** | **72.04**    | **71.66**           | **66.71**           | **47.69**     | **46.69**          | **40.53**          |
> | **FedMuon**  | **74.32**       | **74.12**           | **73.05**           | **50.56**     | **50.22**          | **48.18**          |
>
> In the IID setting, the advantages of FedMuon and local Muon are very clear; in the non-IID setting, the performance of local Muon degrades significantly, whereas our FedMuon remains almost unchanged.

---

> ### Author Response · Authors · 2025-12-02
>
> >**Weakness 3:The paper positions FedMuon as a novel extension of Muon to FL, but this could be more explicitly stated. The authors should clearly position FedMuon as an extension of Muon that leverages SCAFFOLD-like bias removal techniques. This clarification would help better situate the work in the context of prior optimizers.**
>
> **Response to W3:**
>
> Thank you for your suggestion! We have incorporated your comment and revised the manuscript accordingly.
>  Our contributions.
> (1) FedMuon can be viewed as the first federated extension of the Muon optimizer.
> Unlike standard local Muon, which applies matrix orthogonalization independently on each client, FedMuon augments Muon with a  \textbf{local--global alignment} to correct the client-drift induced by heterogeneous data and matrix orthogonalization. (2) FedMuon bridges structured optimizers and classical FL methods, and we prove that matrix orthogonalization accelerates the convergence of federated learning algorithms. (3) We design a federated framework that is applicable to all matrix-structured optimizers (Muon / Shampoo / LAMB / Soap, etc.), which specifically addresses the problem of preconditioner drift through local–global alignment and momentum aggregation.

---

### Author Response · Authors · 2025-11-28
**Acknowledgment of Reviewers’ Feedback**

Dear Reviewers,

Thank you very much for taking the time to thoroughly evaluate our manuscript and for providing insightful feedback that has significantly improved the quality of our work. We sincerely appreciate the care, expertise, and thoughtful consideration reflected in every comment.

We are grateful to the reviewers whose concerns have been fully resolved and appreciate your encouraging remarks. For any issues that may still require further clarification or discussion, we would be more than happy to continue the conversation.

Thank you once again for your valuable time and constructive input.

Best regards,
Authors

---

> ### Author Response · Authors · 2025-11-28
> **General Response to All Reviewers**
>
> ## **Summary**
>
> In summary, the revised paper:
>
> 1. **Clarifies the conceptual novelty** of FedMuon as a federated optimizer, not a direct transplant of Muon.
> 2. Introduces **two new FL-specific mechanisms**—local–global alignment and cross-round momentum aggregation.
> 3. Provides **stronger theory** explaining FedMuon’s behavior under heterogeneity.
> 4. Expands experiments with **more baselines, more models, more heterogeneity, and more ablations**.
> 5. Demonstrates **consistent, state-of-the-art performance** across vision and language tasks.
>
> We thank all reviewers again for their valuable comments and hope the revisions address all major concerns while clearly illustrating the contributions and practical value of FedMuon.

---

### Author Response · Authors · 2025-11-28
**Novelty of FedMuon and Its Relationship to Existing Methods**

# **1. Novelty of FedMuon and Its Relationship to Existing Methods (Common Reviewer Concern)**

Several reviewers asked for further clarification on the following points:

* Is **FedMuon** merely a direct application of **Muon** to federated learning?
* How does it fundamentally differ from existing FL methods (e.g., **SCAFFOLD**, **FedProx**, **Mime**, **FedAdam**, **FedLADA**)?
* Is the novelty only about “matrix orthogonalization”?

In the revised manuscript, we explicitly clarify:

---

## **(1) FedMuon is *not* a simple transfer of Local Muon to the FL setting.**

We found that **Local Muon remains stable under IID conditions**, but **fails under non-IID federated settings** due to two previously unrecognized issues:

1. **Client-specific matrix orthogonalization produces inconsistent preconditioners**, leading to highly divergent local update directions.
2. **Each round resets momentum**, causing optimizer states to be lost across rounds, which severely hurts global optimization stability.

These issues fundamentally prevent Local Muon from working in FL without additional mechanisms.

---

## **(2) FedMuon introduces *two mechanisms* that have not appeared in prior FL literature:**

### **• Local–Global Alignment**

This mechanism injects an estimated global update direction into each client, ensuring that—even after matrix orthogonalization—the clients maintain consistent descent directions, thus mitigating client drift at its root.

### **• Momentum Aggregation with SVD Compression**

We use low-rank SVD to extract and share momentum across rounds, enabling stable cross-round optimizer-state reuse while adding **only ~5% communication cost** relative to baseline FL methods.

---

## **(3) FedMuon is a new structured-optimization framework for federated learning.**

It is **not limited to Muon**—the framework also applies to **Shampoo**, **LAMB**, **SOAP**, and other matrix-structured optimizers, demonstrating broad methodological significance beyond a single algorithm.

---

---

### Note · Authors · 2026-01-28

I have read and agree with the venue's withdrawal policy on behalf of myself and my co-authors.

---

### Meta-Review · Area_Chair_kXYr · 2026-01-07

**Summary:**

Reviewers initially expressed skepticism regarding the novelty of the method and its practical viability in resource-constrained federated environments:

* Conceptual Novelty: Multiple reviewers questioned if the work was merely a direct application of the pre-existing Muon optimizer to federated learning

* Computational and Communication Overhead: Reviewers were concerned about the costs of performing matrix orthogonalization (Newton-Schulz iterations or SVD) locally on devices and the extra bandwidth required for momentum aggregation

* Insufficient Baseline Comparisons: Several reviewers noted the absence of standard FL drift-mitigation methods like FedProx, FedDyn, and MIME, as well as other structured optimizers like Shampoo or LAMB

* Hyperparameter Sensitivity and Scaling: Concerns were raised regarding the stability of new hyperparameters and whether the method could scale to thousands of clients with very low participation

* Theory and Clarity: Reviewer 68eU identified an unused non-IID assumption (A.3) in the theorem as well as typos.

**Reviewer Concerns:**

Addressed Concerns

* Conceptual Novelty: The authors clarified that a direct transplant of Muon fails in non-IID FL due to "preconditioner drift" and "optimizer-state collapse". They introduced Local-Global Alignment and Momentum Aggregation with SVD Compression as unique mechanisms to solve these unrecognized failure modes.

* Baselines and Scaling: The authors added extensive experiments against FedProx, FedDyn, MIME, and FedAdam across ResNet and ViT models. They also provided new results for a 1,000-client setting with low participation, where FedMuon maintained its advantage.

* Efficiency Analysis: Authors provided a per-round latency breakdown showing only a small increase in computation time compared to AdamW. Communication costs were quantified, showing that SVD-compressed momentum adds only small overhead.

* Technical Corrections: The authors removed the Assumption A.3 from the theory and corrected typos. They also reorganized the Appendix.

Outstanding Concerns

* Reproducibility: While authors updated the code, there are still concerns around reproducibility.

* Communication costs and the ease of implementation: The computation time is compared with AdamW, not the efficient versions of AdamW developed in prior works. The overall complexity (though decomposition or other compression techniques can be used) of the proposed algorithm can be high.

* SOAP Optimizer Potential: The authors admitted that the SOAP optimizer performed well and could potentially outperform Muon with the same correction strategies, leaving the "best" base optimizer for this framework as an open question for future work.

**Reviewer Scores:**

* qsYz	Very Positive. The reviewer noted that their concerns were "fully resolved" after the extensive new tables.


* tL4h	Positive. Indicated "increased confidence and support" for the paper following the baseline and cost clarifications.


* xw3a	Supportive. Already leaned toward acceptance; the detailed wall-clock and FLOP profiling solidified this stance.


* 68eU	Mixed. While "satisfied" with many responses, they remained critical of Q3 and Q4 late in the discussion.

---

### Decision · Program_Chairs · 2026-01-26

Reject